# Exact Certification of Neural Networks and Partition Aggregation Ensembles Against Label Poisoning

## Abstract

Label-flipping attacks, which corrupt training labels to induce misclassifications at inference, remain a major threat to supervised learning models. This drives the need for robustness certificates that provide formal guarantees about a model's robustness under adversarially corrupted labels. Existing certification frameworks rely on ensemble techniques such as smoothing or partition-aggregation, but treat the corresponding base classifiers as black boxes—yielding overly conservative guarantees. We introduce **EnsembleCert**, the first certification framework for partition-aggregation ensembles that utilizes white-box knowledge of the base classifiers. Concretely, EnsembleCert yields tighter guarantees than black-box approaches by aggregating per-partition white-box certificates to compute ensemble-level guarantees in *polynomial* time. To extract white-box knowledge from the base classifiers efficiently, we develop **ScaLabelCert**, a method that leverages the equivalence between sufficiently wide neural networks and kernel methods using the neural tangent kernel. ScaLabelCert yields the *first* **exact**, **polynomial-time** calculable certificate for neural networks against label-flipping attacks. Ensemble-Cert is either on par, or significantly outperforms the existing partition-based black box certificates. Exemplary, on CIFAR-10, our method can certify upto $+\mathbf{26.5\%}$ more label flips in median over the test set compared to the existing black-box approach while requiring $\mathbf{100\times}$ fewer partitions, thus, challenging the prevailing notion that heavy partitioning is a necessity for strong certified robustness.

## 1 Introduction

Machine learning models, especially those trained in supervised settings, are critically dependent on the integrity of labeled data. This reliance exposes them to *label-flipping attacks*, where the training labels are corrupted to degrade model performance, or induce targeted misclassifications (Biggio et al., 2011; Xiao et al., 2015). In response, a range of empirical defenses have been proposed, including data sanitization techniques that aim to identify and remove poisoned samples prior to training (Paudice et al., 2018), and adversarial training methods that improve robustness by learning on perturbed examples (Bal et al., 2025). However, these approaches often rely on heuristics and have been shown to fail under adaptive attacks (Carlini & Wagner, 2017; Athalye et al., 2018; Koh et al., 2021). This limitation has led to growing interest in *robustness certificates*, that provide formal guarantees about the robustness of a model's predictions under a given adversarial threat model.

Existing certificates against label-flipping poisoning attacks are predominantly derived using *ensemble* methods. Techniques include randomized smoothing (Rosenfeld et al., 2020), where base classifiers are trained on datasets with randomly perturbed labels, and partition aggregation (Levine & Feizi, 2020), which trains base classifiers on *disjoint* partitions of the training data. Since these certificates rely solely on the base classifier outputs, they are inherently black-box (Ashtiani et al., 2020). Black-box treatment of the base classifiers often leads to overly conservative guarantees and provides limited knowledge about the full extent of the ensemble's robustness. One way to understand the true robustness of the certified model is to utilize *white-box* information of the base classifiers, i.e., white-box certificates, that leverage internal model information to yield tighter and more informative guarantees. This raises the question: *How can we leverage white-box knowledge of the base classifiers to derive a stronger certificate for the ensemble?*

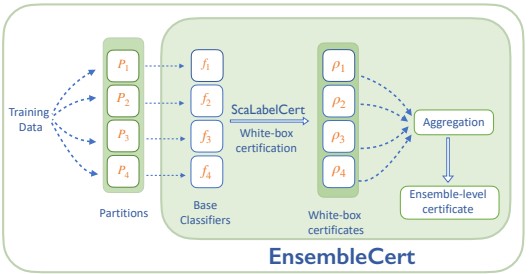
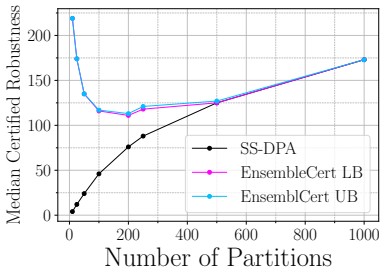

(a) Two-step approach of **EnsembleCert**.

(b) EnsembleCert on CIFAR-10.

Figure 1: (a) Two-step approach of **EnsembleCert** to derive white-box guarantees for partition aggregation ensembles. (b) Evaluation on CIFAR-10 using wide neural networks trained on a regression loss as base classifiers. Using as few as **10** partitions with white-box knowledge enables the ensemble to withstand up to **26.5**% more label flips in median compared to using **1000** partitions. For a definition of the metric *median certified robustness* we refer the reader to Sec. 4.

In this work, we answer this question by proposing **EnsembleCert**, a white-box certification framework for *partition-based aggregation* ensembling techniques (Levine & Feizi, 2020). We focus specifically on the partition-based approach since they are the current state-of-the-art certifiable defense against general data poisoning attacks (Levine & Feizi, 2020; Rezaei et al., 2023; Wang et al., 2022), including label-flipping. Additionally, neural networks can be used as base classifiers in this approach, as opposed to only linear classifiers in the randomized smoothing method (Rosenfeld et al., 2020). EnsembleCert yields tighter white-box guarantees by leveraging the model information of the base classifiers following a simple two-step approach: $(i)$ Extract white-box certificates [1] from the base classifiers for each partition; $(ii)$ Aggregate the white-box certificates to derive an ensemble-wide certificate (see Fig. 1a). The problem of aggregating the partition-wise guarantees to obtain the certificate for the ensemble is formulated as an Integer Program (IP), which we show can be solved efficiently in *polynomial* time.

Thus, given a base model and a certification method for extracting white-box knowledge from the chosen base model, EnsembleCert aggregates the white-box knowledge of base classifiers to achieve ensemble-wide guarantees. In this work, we focus on deriving ensemble-level guarantees when neural networks are chosen as the base model. For certifying neural networks as base models in EnsembleCert, existing white-box approaches face significant challenges as they either rely on computationally intense Mixed Integer Linear Program (MILP) formulation (Sabanayagam et al., 2025) or loose gradient-based parameter bounding approaches (Sosnin et al., 2024). To elaborate, solving the MILP is NP-hard in the worst case, hence LabelCert (Sabanayagam et al., 2025) is practical only for datasets with a few hundred training points and does not even scale to moderately sized datasets like MNIST and CIFAR-10. On the other hand, the parameter-bounding technique (Sosnin et al., 2024) provides overly loose guarantees, leading to vacuous bounds in just few training iterations, especially for multi-class classification tasks. Furthermore, the latter method has so far been evaluated only on small multi-layer perceptrons. These limitations make the existing methods unsuitable for white-box injection into EnsembleCert, naturally raising a broader question: *Can we derive effective and scalable white-box certificates for neural networks against label-flipping attacks?*

We answer this question by developing **ScaLabelCert**, a framework that builds on the *exact* white-box method of LabelCert (Sabanayagam et al., 2025). LabelCert derives the first exact certificate for neural networks against data poisoning by leveraging the equivalence between infinitely wide Neural Networks (NNs) trained with a soft-margin loss and Support Vector Machines (SVM) using the Neural Tangent Kernel (NTK) of the network as their kernel (Chen et al., 2022; Sabanayagam et al., 2023). ScaLabelCert shows that under certain conditions, the computation complexity of LabelCert can be reduced from NP-hard to polynomial time, thus, significantly improving the scalability. Beyond the SVM formulation, ScaLabelCert further extends LabelCert by leveraging the equivalence between infinitely-wide NNs trained with a regression loss and kernel regression under the NTK (Jacot et al., 2018; Arora et al., 2019). With its ability to efficiently compute tight certificates, we adopt ScaLabelCert as our primary mechanism for injecting white-box knowledge into EnsembleCert. The tightness of the resulting partition-wise guarantees reveals the full potential of EnsembleCert and enables a reliable analysis of how partitioning contributes to robustness. Since ScaLabelCert is

---

[1]The specifics of white box information extracted from each base classifier are provided in Sec. 3.1

best suited for certifying infinite-width neural networks (see Sec. 5), we use this instantiation for our primary evaluation. To demonstrate the applicability of EnsembleCert with finite-width networks as base classifiers, we employ the gradient-based parameter bounding approach of (Sosnin et al., 2024) for base-classifier certification. We detail the process of integrating the gradient-based certificate into EnsembleCert and present the evaluation in App. C.4. Finally, to highlight that EnsembleCert is not restricted to neural networks, we also instantiate it with a smoothed linear classifier as the base model and apply randomized smoothing certificates (Rosenfeld et al., 2020) to each base classifier. **Our contributions** are summarized as follows:

**1**. We present **EnsembleCert** in Sec. 3.1, the *first* white-box certification framework for partition aggregation ensembles that leverages the knowledge about base-classifiers to provide white-box informed certificates for the ensemble in **polynomial** time. In our experimental set-up, we evaluate EnsembleCert with the following choices of base classifiers and corresponding certification methods: $(i)$ Infinite-width neural networks with ScaLabelCert, $(ii)$ Finite-width neural networks with gradient-based parameter bounding certificate by Sosnin et al. (2024) and $(iii)$ Smoothed linear classifier with randomized smoothing based certificate by Rosenfeld et al. (2020).

**2**. With **ScaLabelCert** in Sec. 3.2, we derive the *first polynomial-time solvable exact certificate* for infinite-width neural networks against label-flipping attacks and thus, it is the first exact certificate for neural networks against a poisoning threat model that scales to common image benchmarks.

**3**. We show in Sec. 4 that for partition aggregation ensembles with a small number of partitions, the infusion of white-box knowledge results in significant improvement in certified robustness. On analyzing the dependence of certified robustness on the number of partitions, we demonstrate that in certain cases, using as low as 10 partitions with white-box knowledge results in stronger robustness guarantees in comparison to as high as 1000 partitions (see Fig. 1b). These findings call into question the emphasis on using very large numbers of partitions to achieve good certified robustness (Levine & Feizi, 2020), suggesting that excessively deep partitioning, which requires training a prohibitively large number of neural networks, is not a necessity to yield strong guarantees.

## 2 PRELIMINARIES

**Notation.** Matrices are denoted by bold uppercase letters, $\mathbf{M}$, and vectors by bold lowercase letters, $\mathbf{v}$. The $(i, j)$-th entry of a matrix $\mathbf{M}$ is denoted $m_i^j$. For a positive integer $C$, we write $[C] = \{1, \ldots, C\}$. The $\ell_0$ norm is denoted by $\| \cdot \|_0$, and $\mathbf{1}_{\text{condition}}$ represents the indicator function of a given condition. We use $\mathbf{1}^n$ for a vector of all 1s of size $n$. The floor operator is denoted by $\lfloor \cdot \rfloor$.

**Label-flipping and Certification.** In a supervised classification task, the training data $\mathcal{S} = (\mathbf{X}, \mathbf{y})$ consists of feature vectors aggregated in $\mathbf{X} \in \mathbb{R}^{n \times d}$ and labels $\mathbf{y} \in [K]^n$, where $K$ is the number of classes. A learning algorithm $\mathcal{L}_{\text{alg}}$ takes the training set $\mathcal{S}$ and a test sample $\mathbf{t} \in \mathcal{T}$, where $\mathcal{T}$ is the test set, as input to predict the label for $\mathbf{t}$, i.e., $\mathcal{L}_{\text{alg}}(\mathcal{S}, \mathbf{t}) \in [K]$. In a label-flipping attack, we assume that the adversary is allowed to change at most $r \leq n$ training labels. Formally, an adversary can alter the clean labels $\mathbf{y}$ to $\tilde{\mathbf{y}} \in \mathcal{B}_r(\mathbf{y}) := \left\{ \tilde{\mathbf{y}} \in [K]^n \mid \|\tilde{\mathbf{y}} - \mathbf{y}\|_0 \leq r \right\}$ and get a perturbed training set $\tilde{\mathcal{S}} = (\mathbf{X}, \tilde{\mathbf{y}})$. As the certification objective, for every $\mathbf{t} \in \mathcal{T}$, we aim to find the maximum number of label flips $\tilde{r}$ in the clean training data up to which the prediction of $\mathcal{L}_{\text{alg}}$ for $\mathbf{t}$ does not change, i.e.

$$\tilde{r}(\mathbf{t}) = \max_{\tilde{\mathcal{S}}} r \quad s.t. \quad \mathcal{L}_{\text{alg}}(\mathcal{S}, \mathbf{t}) = \mathcal{L}_{\text{alg}}(\tilde{\mathcal{S}}, \mathbf{t}) \ \ \forall \tilde{\mathcal{S}} \in \{\mathcal{S}' \mid \mathbf{y}' \in \mathcal{B}_r(\mathbf{y}))\}.$$

We will refer to $\tilde{r}(\mathbf{t})$ as the certified radius for $\mathbf{t}$. A point-wise certificate then would be a lower bound on the certified radius for a particular sample. The certificate is *exact* if it gives the true certified radius $\tilde{r}(\mathbf{t})$ rather than just a lower bound.

**Semi-Supervised Deep Partition Aggregation (SS-DPA)**. Levine & Feizi (2020) introduce SS-DPA, a framework that builds a certified defense against label-flipping poisoning attacks. The framework certifies a partition aggregation ensemble $g_{\mathcal{S}}$, i.e , an ensemble consisting of $N_p$ base classifiers $f_{\{1, \ldots, N_p\}}$ trained on disjoint partitions $P_{\{1, \ldots, N_p\}}$ of the training data $\mathcal{S}$. The motivation behind training on disjoint partitions is simple: Poisoning one label in the training data affects the prediction of only one of the base-classifiers. The training data $\mathcal{S}$ is first sorted without using the labels and then partitioned based on the sorted order. This ensures that the partitioning is invariant

to any label poisoning attack. As the unlabeled data is trustworthy, we can make use of a self-supervised learning algorithm to extract features from the entire unlabeled training data and train each $f_i$ using the extracted features and labels corresponding to $P_i$. At inference time, each base classifier $f_i$, trained on its corresponding partition $P_i$ of $\mathcal{S}$ predicts the class for a given test sample $\mathbf{t} \in \mathcal{T}$ as $f_i(\mathbf{t}) \in [K]$. The prediction of the ensemble $g_\mathcal{S}(\mathbf{t})$ is then determined by a majority vote: $g_\mathcal{S}(\mathbf{t}) = \arg\max_{c \in [K]} n_c(\mathbf{t})$, where $n_c(\mathbf{t}) := |\{i \in [N_p] \mid f_i(\mathbf{t}) = c\}|$ is the number of votes received by class $c$. Ties are resolved deterministically by choosing the smaller index. If we denote $g_\mathcal{S}(\mathbf{t})$ as $c^*$, the certificate $\tilde{\rho}(\mathbf{t})$ for sample $\mathbf{t}$ is given as:

$$\tilde{\rho}(\mathbf{t}) := \Big\lfloor \frac{n_{c^*}(\mathbf{t}) - \max_{c' \neq c^*}(n_{c'}(\mathbf{t}) + \mathbf{1}_{c' < c^*})}{2} \Big\rfloor.$$

The above guarantee says that for a poisoned dataset $\tilde{\mathcal{S}}$ obtained by changing the labels of at most $\tilde{\rho}$ samples in $\mathcal{S}$, $g_{\tilde{\mathcal{S}}}(\mathbf{t}) = c^*$. As each base classifier is treated as a black-box, the certificate derivation follows from a key worst-case assumption: **The prediction of a base classifier can be changed by a *single* label flip**. The formal description of the worst case scenario is presented in App. A.1. With *white-box* knowledge about the base classifiers, one can *improve* upon this worst-case assumption, leading to a tighter certificate for the ensemble.

# 3 METHODOLOGY: ENSEMBLECERT AND SCALABELCERT

## 3.1 ENSEMBLECERT

The underlying worst-case assumption in existing partition aggregation-based certificates, which says that the prediction of a base classifiers can be changed with a single label flip, can be overcome given that we have the following white-box information: for all base classifiers $f_{\{1,\dots,N_p\}}$ and $\forall c \in [K]$, we have access to $\rho_i^c$, which is the minimum number of label flips in $P_i$ required to change the prediction of the base classifier $f_i$ (trained on $P_i$) to class $c$. Access to the white-box knowledge through $\rho_i^c$ enables verification of the worst-case assumption and provides the necessary information to derive a tighter ensemble-level certificate. Note that the ensemble-level certificate $\tilde{\rho}(\mathbf{t})$ that represents the **maximum number of flips upto which the ensemble prediction for a sample $\mathbf{t}$ remains unchanged**, is simply one less than the minimum number of flips required to change the ensemble prediction. To determine the ensemble certificate $\tilde{\rho}(\mathbf{t})$, we first compute, for each class $c$, the least number of flips needed to make $c$ the majority class, and then take the minimum over all classes.

**Integer Program Formulation for Ensemble-wide Certification**. We denote the problem of finding the minimum number of label flips in the training set required to change the prediction of the ensemble to a particular class $c'$ as $P_1(c')$. The white-box information $\rho_i^c$ is collected in $\boldsymbol{\rho} \in \mathbb{R}^{N_p \times K}$. Given $\boldsymbol{\rho}$, finding the optimal attack for the adversary, which is equivalent to solving $P_1(c')$, poses as a combinatorial optimization problem leading to an Integer Program (IP) formulation of $P_1(c')$. We denote the $i$th base classifier as $f_i$ if trained on the clean data and $\tilde{f}_i$ if trained on the perturbed data. The predictions from $f_{\{1,\dots,N_p\}}$ and $\tilde{f}_{\{1,\dots,N_p\}}$ on the sample $\mathbf{t}$ are collected in the vote configurations $\mathbf{V}$ and $\tilde{\mathbf{V}} \in \mathbb{R}^{N_p \times K}$ respectively: $\forall i \in N_p, c \in [K]: v_i^c = \mathbf{1}\{f_i(\mathbf{t}) = c\}$, $\tilde{v}_i^c = \mathbf{1}\{\tilde{f}_i(\mathbf{t}) = c\}$. Note that $\sum_{c=1}^K v_i^c = 1$ and $\sum_{c=1}^K \tilde{v}_i^c = 1$ for all $i \in [N_p]$. Concretely, to model $P_1(c')$, the number of label flips required to reach the vote configuration $\tilde{\mathbf{V}}$ from $\mathbf{V}$ is $\sum_{i=1}^{N_p} \sum_{c=1}^K \rho_i^c \tilde{v}_i^c$. The constraint that $c'$ should be the majority class after adversarial manipulation of labels can be represented as $\sum_{i=1}^{N_p} \left( \tilde{v}_i^{c'} - \tilde{v}_i^c \right) \geq \mathbf{1}_{c < c'}$, for all $c \neq c'$. Recollect $\sum_{c=1}^K \tilde{v}_i^c = 1, \ \forall i \in [N_p]$ should also be satisfied. Thus, this gives the IP formulation of $P_1(c')$:

$$P_1(c'): \quad \min_{\tilde{\mathbf{V}}} \sum_{i=1}^{N_p} \sum_{c=1}^K \rho_i^c \tilde{v}_i^c \quad \text{s.t.} \quad \forall c \neq c': \ \sum_{i=1}^{N_p} \left( \tilde{v}_i^{c'} - \tilde{v}_i^c \right) \geq \mathbf{1}_{c < c'},$$

$$\forall i \in [N_p], \ \forall c \in [K]: \ \sum_{c=1}^K \tilde{v}_i^c = 1, \quad \tilde{v}_i^c \in \{0, 1\}.$$

The ensemble-level certificate $\tilde{\rho}(\mathbf{t})$ for a test sample $\mathbf{t}$ can then be derived, as mentioned in Sec. 3.1, by simply subtracting one from the minimum over $P_1(c)$, that is, $\tilde{\rho}(\mathbf{t}) = \min_{c \in [K] \setminus c^*} P_1(c) \ - 1$.

**Reduction to Polynomial-time**. Solving $P_1(c)$ in its current form is computationally prohibitive, scaling as $\mathcal{O}(2^{N_p \times K})$ in the worst case. Thus, deriving $\tilde{\boldsymbol{\rho}}$ is even more expensive, with complexity $\mathcal{O}(K \times 2^{N_p \times K})$. The problem becomes intractable even for small values of $N_p$ and $K$, motivating the need for a more tractable alternative. We denote as $P_2(c')$, a relaxation of $P_1(c')$ that finds the minimum number of label flips needed to make $c'$ surpass *only* $c^*$ (the original majority class) in the number of votes, rather than making $c'$ the overall majority class. The formulation of $P_2(c')$ can be obtained from $P_1(c')$ by relaxing the constraint $\sum_{i=1}^{N_p} (\tilde{v}_i^{c'} - \tilde{v}_i^{c}) \geq \mathbf{1}_{c<c'} \ \forall c \neq c'$ to the constraint $(\tilde{v}_i^{c'} - \tilde{v}_i^{c^*}) \geq \mathbf{1}_{c^*<c'}$. Despite this relaxation, we have the following result proved in App. A.2.

**Theorem 1** (Equivalence between problems $P_1$ and $P_2$)**.**

$$\boxed{\min_{c \in [K] \setminus c^*} P_1(c) = \min_{c \in [K] \setminus c^*} P_2(c)}$$

The intuition for the above result is as follows: while trying to make $c'$ surpass $c^*$, if another class $c''$ becomes the majority class, then changing the ensemble prediction to $c''$ should be easier compared to $c'$. This result is particularly important, as we show that $P_2(c)$ can be reduced to an instance of the Multiple Choice Knapsack Problem (MCKP). Since MCKP is solvable in pseudopolynomial-time (Dudzinski & Walukiewicz, 1987), our approach achieves a complexity of $\mathcal{O}(N_p^2)$ for solving $P_2$ per class (App. A.2). Consequently, the ensemble-wide certificate $\tilde{\rho}(\mathbf{t})$ can be computed in *polynomial* time by solving $P_2(c)$ for every class and finding the minimum, that is, $\tilde{\rho}(\mathbf{t}) = \min_{c \in [K] \setminus c^*} P_2(c) \ - 1$. This represents a substantial improvement over the naive ILP formulation with complexity $\mathcal{O}(K \times 2^{N_p \times K})$. We refer to App. A.3 for details on the reduction of $P_2(c')$ to MCKP.

### 3.2 Extraction of White-box Knowledge Through ScaLabelCert

The approach of our white-box certificate *ScaLabelCert* builds on the framework introduced by LabelCert (Sabanayagam et al., 2025). LabelCert provides an exact certificate that determines whether the model prediction remains unchanged when at most $r$ training labels are flipped. This definition of the certificate does not immediately align with the white-box knowledge that EnsembleCert utilizes, which is the minimum number of label flips required to change the prediction of the classifier to a *particular* class. Even more problematic, the computation of the certificate by LabelCert is NP-hard and only scales to a few hundred labeled datapoints. ScaLabelCert makes modifications to the LabelCert approach to address these shortcomings, which result in the computation of an *exact* certificate against label-flipping attacks in *polynomial* time. Next, we provide a brief overview of the approach by LabelCert, and then introduce the developments leading to ScaLabelCert.

**Infinite-Width Neural Networks and The Equivalence to Kernel Methods.** The Neural Tangent Kernel (NTK) of a neural network $f_\theta$ between two inputs $i$ and $j$ with features $x_i$ and $x_j$ is defined as $Q_i^j = \mathbb{E}_\theta[\langle \nabla_\theta f_\theta(x_i), \ \nabla_\theta f_\theta(x_j) \rangle]$, where the expectation is taken over the parameter initialization. When $f_\theta$ is an infinitely wide neural network, the dynamics of training $f_\theta$ for a classification task using a soft-margin loss are the same as those of an SVM with $f_\theta$'s NTK as kernel (Chen et al., 2022). Similarly, if a regression loss (regularized mean-square) is used, the training dynamics are equivalent to those of kernel regression using $f_\theta$'s NTK as kernel (Jacot et al., 2018).

**LabelCert.** For a test sample $\mathbf{t}$, LabelCert computes a point-wise certificate for sufficiently wide neural networks, by deriving a certificate for a kernel SVM with $f_\theta$'s NTK as kernel, which—due to the above equivalence—extends to a certificate for $f_\theta$. Recall that in the dual formulation of an SVM, the parameters are the dual variables $\boldsymbol{\alpha} \in \mathbb{R}^n$ derived by solving the following problem:

$$P_{\mathbf{svm}}(\mathbf{y}) = \min_{\boldsymbol{\alpha}} - \sum_{i=1}^{n} \alpha_i + \frac{1}{2} \sum_{i=1}^{n} \sum_{j=1}^{n} \alpha_i \alpha_j y_i y_j Q_i^j \quad \text{s.t.} \quad 0 \leq \alpha_i \leq C, \ \forall i = [n]$$

where $n$ is the number of training data, $C$ is the regularization parameter that controls the trade-off between maximizing the margin and minimizing classification error, and $Q_i^j$ is the chosen kernel between inputs $i$ and $j$. Let the set of $\boldsymbol{\alpha}$ vectors solving $P_{\mathbf{svm}}(\mathbf{y})$ be $\mathcal{S}(\mathbf{y})$. The prediction for a test

sample $\mathbf{t}$ is given by $p_t = \text{sign}(\sum_{i=1}^{n} \alpha_i \tilde{y}_i Q_t^i)$. Let $\hat{p}_t$ be the prediction of the SVM trained using clean labels. The certificate is computed by converting the following problem $P_{\mathbf{cert}}(\mathbf{y})$ to a MILP:

$$P_{\mathbf{cert}}(\mathbf{y}) := \min_{\tilde{\mathbf{y}}, \boldsymbol{\alpha}} \; \text{sign}(\hat{p}_t) \sum_{i=1}^{n} \alpha_i \tilde{y}_i Q_t^i \quad \text{s.t.} \quad \tilde{y} \in \mathcal{A}_r(\mathbf{y}), \;\; \boldsymbol{\alpha} \in S(\tilde{\mathbf{y}})$$

Whether the model prediction for $\mathbf{t}$ is robust up to $r$ label flips or not is determined by the sign of the solution to $P_{\mathbf{cert}}(\mathbf{y})$, with a positive sign indicating robustness.

**SVM Formulation for Sufficiently Small $C$.** The complexity of solving $P_{\mathbf{cert}}(\mathbf{y})$ comes largely from replacing the inner optimization problem $\boldsymbol{\alpha} \in S(\tilde{\mathbf{y}})$ with the KKT (Karush-Kuhn-Tucker) conditions of $P_{\mathbf{svm}}(\mathbf{y})$, which can be done as $P_{\mathbf{svm}}(\mathbf{y})$ is convex (Dempe & Dutta, 2012; Sabanayagam et al., 2025). We show that on using a sufficiently small $C$, we can entirely forego the inner optimization problem and convert $P_{\mathbf{cert}}(\mathbf{y})$ to a simpler, single-level problem based on Theorem 2.

**Theorem 2.** *Given a soft margin SVM with regularization $C$, kernel entry between training samples $i$, $j$ as $Q_i^j$, and $\boldsymbol{\alpha}$ being the solution to $P_{svm}(\mathbf{y})$, then if*

$$\max_{i \in [n]} \sum_{j \in [n]} |Q_i^j| \leq \frac{1}{C}, \quad \text{it follows that} \quad \forall \mathbf{y} \in \{-1, 1\}^n: \; \boldsymbol{\alpha} = C \cdot \mathbf{1}^n$$

The proof is presented in App. A.4. When $C$ satisfies the condition stated above, the alpha values are equal to $C$ *regardless* of the labels. Thus, choosing $C$ appropriately gives us the liberty to eliminate the inner optimization problem $\boldsymbol{\alpha} \in S(\tilde{\mathbf{y}})$ as $\boldsymbol{\alpha}$ is invariant to different labelings of the data. The SVM prediction in this case simplifies to $p_t = \text{sign}(\sum_{i=1}^{n} C \tilde{y}_i Q_t^i)$. As $C$ is a positive constant, this further simplifies to $p_t = \text{sign}(\sum_{i=1}^{n} \tilde{y}_i Q_t^i)$. Integrating this insight into ScaLabelCert, we develop an efficient computation scheme for exact white-box certificates for infinite-width networks below that calculates the minimum number of label flips needed to change the prediction of the model.

**ScaLabelCert For The Binary Setting.** Our objective is to find the minimum number of label flips required to change the SVM prediction, i.e., to make $\text{sign}(\hat{p}_t) \sum_{i=1}^{n} \alpha_i \tilde{y}_i Q_t^i$ negative. Under sufficiently small $C$, the above objective can be formulated as:

$$O_1(\mathbf{y}): \quad \min_{\tilde{\mathbf{y}} \in \{-1,1\}^n} \frac{1}{2} \sum_{i=1}^{n} (1 - y_i \tilde{y}_i) \quad \text{s.t.} \quad \text{sign}(\hat{p}_t) \sum_{i=1}^{n} \tilde{y}_i Q_t^i < 0, \; \forall i \in [n] : \tilde{y}_i \in \{-1, 1\}.$$

$O_1(y)$ can be solved *in polynomial time* (App. A.5). The intuition is that the labels corresponding to the largest positive contributions in $\text{sign}(\hat{p}_t)(\sum_{i=1}^{n} y_i Q_t^i)$ are the most influential in determining the prediction, so flipping these labels greedily till the prediction changes is the optimal attack from the adversary's point of view. Thus, solving $O(\mathbf{y})$ leads to a polynomial time computable exact certificate for sufficiently-wide neural networks, if $f_\theta$'s NTK is chosen as the SVM's kernel.

**ScaLabelCert For The Multi-Class Setting**. For the multi-class case, we use the one-vs-all strategy by decomposing the problem with $K$ classes into $K$ separate binary classification tasks. For each class $c \in [K]$, a binary classifier is trained to distinguish between samples of class $c$ and samples from all other classes. Assume that $p_c$ is the prediction score of a classifier for the learning problem corresponding to class $c$. Then, the class prediction $c^*$ for a test sample is constructed by $c^* = \arg\max_{c \in [K]} p_c$. The labels are collected in the vector $\mathbf{y}$ where $\mathbf{y}_i^c = 1$ if the class of the $i$th sample is $c$, and 0 otherwise. Recall that for each base classifier, EnsembleCert requires white-box certificates that determine, for *every* class, the minimum number of label flips needed to change the model's prediction to that class. Using a soft-margin kernel SVM with a sufficiently small $C$ as our base model, the certificate **computing minimum number of label flips required to change the prediction of the model to a particular class $\mathbf{c}'$** can be formulated as (derived in App. A.6):

$$O_1(c'): \quad \min_{\tilde{\mathbf{y}}} \sum_{i \in [N]} \left( 1 - \sum_{c \in [K]} y_i^c \tilde{y}_i^c \right) \quad \text{s.t.} \quad \sum_{i \in [N]} \tilde{y}_i^{c'} Q_t^i > \sum_{i \in [N]} \tilde{y}_i^c Q_t^i \quad \forall c \neq c',$$

$$\forall i \in [N], c \in [K]: \sum_{c \in [K]} \tilde{y}_i^c = 1 \;, \tilde{y}_i^c \in \{0, 1\}. \tag{1}$$

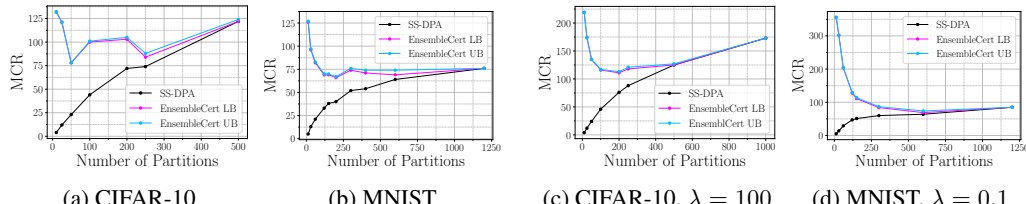

| (a) CIFAR-10 | (b) MNIST | (c) CIFAR-10, $\lambda = 100$ | (d) MNIST, $\lambda = 0.1$ |

Figure 2: EnsembleCert evaluation using the NTK for $(i)$ kernel SVM with a sufficiently small $C$ (a, b); $(ii)$ kernel regression under strong regularization (c, d). Median certified robustness either remains largely invariant across partitions or exhibits a decay until the white-box certificate converges to the black-box certificate. The tightness of our bounds on the *exact* certificate for the ensemble is evident, as the upper (EnsembleCert UB) and lower (EnsembleCert LB) bounds largely coincide across all plots.

While solving $O_1(c')$ is NP-hard, we show that *tight lower and upper bounds* for the solution of $O_1(c')$ can be computed in polynomial time (see App. A.7).

**Certificate for Kernel Regression**. With minor modifications, we can leverage the above formulation to certify a kernel regression model. Specifically, the adjustment is to replace $Q_t^i$ with

$$(Q_{\text{eff}})_t^i = [(Q_{\text{train}} + \lambda I)^{-1} Q_{t,:}]_i$$

where $Q_{\text{train}}$ is the kernel matrix for the training samples; $Q_{t,:}$ is the vector of kernel entries for test sample $\mathbf{t}$ and the training samples; and $\lambda$ is the regularization parameter. Deriving the certificate for kernel regression with the above modifications, we certify a sufficiently wide NN trained on a regularized mean-squared loss by using the network's NTK as the kernel.

**Exact Certificate Given No Partitioning** ($N_p = 1$). When there is no partitioning, we do not need to solve $O_1(c')$ exactly for every $c'$ to get an exact certificate for a stand-alone model. As $O_1(c')$ represents the number of flips required to change the prediction to a particular class $c'$, the *exact* certificate for the stand-alone model can be derived by simply computing the *minimum* over $O_1(c')$ i.e., $\tilde{\rho}(\mathbf{t}) = \min_{c \in [K] \setminus c^*} O_1(c) - 1$. We show that with ScaLabelCert, this can be solved *in polynomial time*, by employing a similar line of argument as Theorem 1. The proof is presented in App. A.6. This results in the **first exact certificate** for neural networks against a poisoning attack that scales to common image benchmark datasets like MNIST or CIFAR-10.

## 4 EXPERIMENTS AND RESULTS

**Implementation Details.** We perform experiments on MNIST, CIFAR-10, and binary MNIST 1-vs-7. Following SS-DPA (Levine & Feizi, 2020), before training the base-classifiers we extract unsupervised features using RotNet (Gidaris et al., 2018) for MNIST and SimCLR (Chen et al., 2020) for CIFAR-10, using pretrained models from Levine & Feizi (2020). For the supervised training of base-classifiers, the extracted RotNet features for MNIST are used as input to an infinitely-wide convolutional network with a one convolutional layer and no pooling for supervised classification. For CIFAR-10, SimCLR features are fed to an infinitely-wide fully-connected network with one hidden layer and no non-linear activation. NTK computations are performed using the Google `neural-tangents` library (Novak et al., 2020). Using the NN-kernel equivalence (Sec. 3.2), the NTK is then used either with a kernel SVM for wide NNs trained on the soft-margin loss or with kernel regression for wide NNs trained on the regularized mean-squared loss. On CIFAR-10, we evaluate EnsembleCert additionally on two different types of base classifiers $(i)$ Finite-width networks and $(ii)$ Smoothed linear classifiers. The relevant implementation details can be found in App. C.4 and App. B respectively. Solving the MCKP for ensemble-level certificates as described in Sec. 3.1 is implemented using standard dynamic programming. The metrics used for evaluation are *certified accuracy*, with certified accuracy at $r$ label flips being the fraction of test samples for which the model prediction is correct and robust up to $r$ label flips; and *median certified robustness* (MCR), which denotes the number of label flips upto which the model prediction for $50\%$ of the correctly classified samples is robust. We provide further implementation details and certification runtimes in App. C.1.

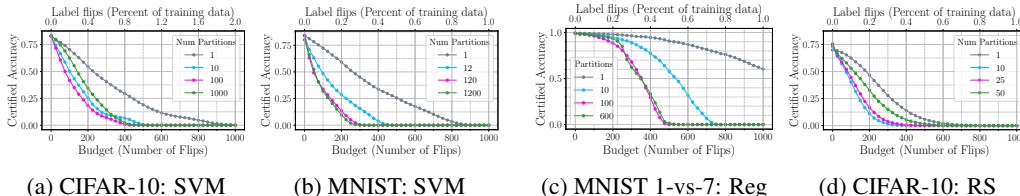

| (a) CIFAR-10: SVM | (b) MNIST: SVM | (c) MNIST 1-vs-7: Reg | (d) CIFAR-10: RS |

Figure 3: Comparing certified accuracies of stand-alone base models and their partition aggregation ensembles. Results with the base model as kernel SVM are in (a) and (b). (c): Using a stand-alone kernel regression model on MNIST 1-vs-7 maintains a certified accuracy of close to **80**% when the certified accuracy for the corresponding best performing ensemble ($N_p = 10$) reaches 0. (d): Smoothed linear regression as base model, comparing the stand-alone case and the ensembling.

**Experiments.** We instantiate EnsembleCert with sufficiently wide NNs trained on the soft-margin loss (equivalent to kernel SVMs with NTK) and the regularized mean square loss (equivalent to kernel ridge regression with NTK). Although regression losses may seem ill-suited for classification, they work well in practice (Mika et al., 1999; Rifkin et al., 2003). Moreover, Arora et al. (2019) showed that kernel ridge regression with the NTK of convolutional NNs achieves competitive performance on image datasets. As noted in our contributions, we additionally perform experiments on CIFAR-10 to evaluate EnsembleCert while using finite-width networks as base classifier. For base classifier certification in this case, we employ the gradient-based parameter bounding technique by Sosnin et al. (2024). The relevant details and plots can be found in App. C.4. We also evaluate EnsembleCert on CIFAR-10 using a smoothed linear regression model as the base classifier, certified via the smoothing-based method of Rosenfeld et al. (2020), which applies to smoothed linear models and yields analytic certificates without requiring the sampling process in randomized smoothing. The details of how we derive the necessary white-box knowledge for EnsembleCert by leveraging the smoothing approach can be found in the App. B. For every choice of base classifier, we observe that *injecting white-box knowledge into the ensemble **substantially increases** certified robustness for low to intermediate numbers of partitions*, highlighting the relative looseness of guarantees obtained using the black-box approach. The substantial improvement in certified robustness achieved by our white-box certificate for kernel methods as base classifiers is evident in Fig. 2. Further results demonstrating the same for every choice of base classifier can be found in App. C. The gap between the white-box and black-box certificates narrows as the number of partitions grows, with the white-box certificate eventually converging to the black-box certificate. This convergence reflects the realization of the worst-case scenario, where a single label flip can alter the prediction of a base classifier. Beyond the point of convergence, our method performs on par with the black-box approach. This behavior is a direct consequence of our method's design and holds consistently across all experiments. In the next sections, we present some crucial insights that can be derived from our evaluation of EnsembleCert and ScaLabelCert.

**Invariance to Number of Partitions with Kernel SVM.** For the instantiation of EnsembleCert with kernel SVM, we use a regularization parameter $C$ that is small enough to satisfy the condition in Theorem 2, as it is the key to computing scalable certificates for kernel SVM. We observe that the MCR of the white-box certificate remains largely **invariant to the number of partitions** for CIFAR-10 (Fig. 2a) and MNIST 1-vs-7 (Fig. 13a) until the point of convergence. On MNIST, there is a **sharp decline initially on increasing the number of partitions, followed by plateauing** (Fig. 2b). These findings indicate that strong guarantees can be achieved without requiring overly large ensembles.

**Robustness Decay with Kernel Regression.** For our instantiation of EnsemblCert with kernel ridge regression, we study the effect of the regularization parameter $\lambda$ on certified robustness of the ensemble. For each dataset, we observe that the trend of certified robustness varies with the regularization parameter $\lambda$. As we increase $\lambda$ from very small values, MCR initially improves with the number of partitions. Beyond a dataset-specific threshold, however, the trend reverses—**increasing the number of partitions leads to lower certified robustness**. For example, on CIFAR-10, when $\lambda = 100$ (which lies beyond the threshold for this dataset), EnsembleCert certifies a median of **219** label flips with just **10** partitions, whereas using **1000** partitions reduces this to **173** (Fig. 2c). Similarly, on MNIST with $\lambda = 0.1$, EnsembleCert certifies **356** label flips using only **12** partitions, whereas using **1200** partitions lowers the certified robustness to **82** (Fig. 2d).We present plots for low values

of $\lambda$ and discuss the behavioral change across the spectrum of $\lambda$ in App. C.5. As robust kernel regression is associated with the use of higher values of $\lambda$ (Hu et al., 2021), the decreasing trend of the MCR observed with high $\lambda$ suggests that **deeper partitioning limits the true robustness potential** of the ensemble when the underlying base classifier has a high degree of robustness.

**To Partition or Not to Partition?** The polynomial-time calculable exact certification method derived by ScaLabelCert allows us to analyze the robustness of sufficiently wide neural networks without employing an ensemble, that is, $N_p = 1$. While the simplification under small $C$ eliminates the need to calculate the training kernel for kernel SVM, making it easily scalable to datasets such as CIFAR-10 and MNIST, it remains an essential component of the pipeline for kernel regression. Thus, performing kernel regression on such datasets without partitioning is computationally challenging due to the need to compute the entire training kernel. Hence, in the no-partition setting, we evaluate ScaLabelCert using the efficient kernel SVMs on all datasets and evaluate using kernel regression only on the relatively small MNIST 1-vs-7 binary dataset. On CIFAR-10, ScaLabelCert achieves non-trivial certified accuracy for up to **1000** label flips, which amounts to **2**% of the training data Fig. 3a. In contrast, the evaluation by Levine & Feizi (2020) fails to certify *any* test sample beyond **500** label flips. Additionally, we compare our method with the gradient-based parameter bounding technique from Sosnin et al. (2024) and show that ScaLabelCert significantly outperforms their method on CIFAR-10 in certified accuracy. Refer to App. C.3 for details. Motivated by the observation that deeper partitioning may limit the ensemble's true robustness potential, we further investigate the role of partitioning by comparing the certified accuracy of a single base model against that of its partition-aggregated ensemble. Our experiments across multiple datasets and base model choices, as shown in Fig. 3, reveal that a **single base model trained on the entire training dataset achieves significantly higher certified accuracy** as compared to its partition-aggregation ensemble. This raises an important question: *Does partition aggregation enhance or diminish the robustness potential of a given base model?*

## 5 DISCUSSION AND CONCLUSION

**Scarcity of relevant white-box certificates.** Our evaluations of EnsembleCert demonstrate significant improvement in certified robustness when the white-box knowledge of the base classifiers is utilised. Notably, EnsembleCert demands white-box certificates deriving minimum number of samples that need to be tampered with to change the prediction to a particular class. The dearth in works exploring certification of this nature pose an imminent challenge in the way of realising the true potential of EnsembleCert.

**Certificate validity for finite-width neural networks.** ScaLabelCert leverages the equivalence of infinite-width kernel methods with kernel methods induced by the NTK. This equivalence in training dynamics and model outputs is exact only in the infinite-width case. For a finite-width neural network however, where $w$ denotes the smallest layer width of the network, the output difference of the network to the SVM is bounded by $\mathcal{O}\left(\frac{\ln w}{\sqrt{w}}\right)$ with probability $p = 1 - \exp(-\Omega(w))$, as shown in Gosch et al. (2025), Liu et al. (2021). As $w$ approaches infinity, the output difference approaches $0$ and $p$ approaches $1$. Consequently, there must exist some width $w'$ such that the output difference between a network with width larger than $w'$ and the corresponding kernel SVM is small enough for the certificate to remain exact. To concretely compute $w'$, one would have to compute the constants associated with the approximation error $\mathcal{O}\left(\frac{\ln w}{\sqrt{w}}\right)$. Unfortunately, the literature on the NTK so far is mainly concerned with providing convergence statements in big-$\mathcal{O}$ notation and not with calculating the individually involved constants. Hence, for a sufficiently wide network, the exact certificate holds with probability $p$ and does not apply with probability $1 - p$. Thus, our certificates obtained by utilizing the neural network and NTK equivalence based on kernel SVM and regression represent an asymptotically exact certificate as the width $w$ approaches infinity.

**On Using Sufficiently Small $C$ in Kernel SVM.** The choice of the parameter $C$, which controls the penalty for misclassifications, introduces a robustness–accuracy trade-off in soft-margin SVMs. Smaller values of $C$ improve robustness to label noise and adversarial perturbations, as they encourage larger margins and reduce the influence of individual (potentially corrupted) points on the decision boundary. Thus, our choice of $C$ for the SVM simplification in Theorem 2 aligns with building robust base-classifiers. Although this choice may not be optimal for clean accuracy, We

show that performance remains competitive. We ask the reader to refer to App. C.2 for a discussion on the robustness-accuracy trade-off and the corresponding experiments.

**Versatility of ScaLabelCert.** Through the formulation $O_1(c')$ (Eq. (1)), ScaLabelCert derives efficient certificates for sufficiently wide networks that compute the minimum number of label flips needed to change the prediction of the classifier to a *particular* class $c'$. Although these certificates are not exact, we compute sufficiently tight bounds (see Sec. 4). Note that the certificate definition is different from the certified radius, which represents the minimum label flips needed to change the classifier's prediction to *any* class. We remind the reader that our certificate for computing the certified radius for a stand-alone model is *exact* and *polynomial-time* calculable (App. A.6). Moreover, ScaLabelCert provides a general framework for certifying kernel SVMs and kernel regression models against label-flipping. Using the NTK is one instance of this framework, enabling efficient certification for sufficiently wide NNs. Finally, a kernel SVM with a sufficiently small $C$ and kernel regression-based classifiers can be interpreted as **weighted nearest-neighbor models**, where $Q_t^i$ and $(Q_{\text{eff}})_t^i$ denote the weight of the $i$th neighbor of the test sample $\mathbf{t}$ for kernel SVM and kernel regression, respectively. From this perspective, ScaLabelCert can also certify **weighted nearest-neighbor models** against label-flipping attacks in **polynomial time**, demonstrating its broad applicability.

**Potential of EnsembleCert for Certifying Against Clean-Label Attacks.** In this work, we utilize EnsembleCert to certify against label-flipping attacks. However, EnsembleCert can also leverage white-box knowledge of base-classifiers to provide robustness guarantees against **clean-label attacks**. Specifically, consider an adversary capable of corrupting *only* the features of a training sample within an $\ell_p$ ball. Under this threat model, the white-box information $\rho_i^c$ can denote the number of samples that must be corrupted to change the prediction of the $i$th base classifier to class $c$. EnsembleCert can aggregate this white-box information from the base classifiers to compute the number of samples in the *entire* training dataset that need to be corrupted to alter the prediction of the ensemble. In this way, EnsembleCert can be adapted to derive white-box certificates for partition aggregation ensembles under multiple threat models. However, deriving efficient and scalable white-box clean-label certificates for certifying the base classifiers is still an open challenge.

**Related Work.** Current ensemble-based poisoning certificates typically use the following ensembling techniques: $(i)$ randomized smoothing (Rosenfeld et al., 2020; Wang et al., 2020; Zhang et al., 2022; Weber et al., 2023), where the randomization is over the training dataset, $(ii)$ partition-based aggregation (Levine & Feizi, 2020; Wang et al., 2022; Rezaei et al., 2023), and $(iii)$ bootstrap aggregation (Jia et al., 2021), where the base classifiers are trained on independently sampled subsets of the training data. None of these works use white-box knowledge of the base classifiers, making them inherently black-box methods. Apart from the white-box certificates discussed in the introduction (Sabanayagam et al., 2025; Sosnin et al., 2024), Gosch et al. (2025) is the only other white-box certification method that certifies NNs against clean-label attacks, notably using the NTK approach similar to ours and to Sabanayagam et al. (2025). The remaining white-box certificates in the literature do not extend to NNs and apply to only decision trees (Meyer et al., 2021; Drews et al., 2020), nearest neighbor models (Jia et al., 2022) or naive Bayes classifiers (Bian et al., 2024).

**Conclusion.** We introduce **EnsembleCert**, a framework that leverages model information from base-classifiers to yield significantly tighter ensemble-level certificates against label-flipping attacks in *polynomial time*. To efficiently extract the white-box information, we develop **ScaLabelCert**, a framework for the exact certification of sufficiently-wide NNs against label-flipping attacks. ScaLabelCert computes exact certificates against label flipping attacks in polynomial time, making it the *first* **polynomial-time** exact certification method that can certify (wide) NNs against data poisoning attacks. Through our evaluation of EnsembleCert instantiated with sufficiently wide NNs, we observe that with robust base-classifiers, the partition aggregation ensemble can achieve stronger guarantees using notably few partitions, outperforming excessively deep partitioning. This is crucial, as excessively deep partitioning requires training a very large number of base-classifiers, introducing significant computational overhead and limiting scalability. The experiments evaluating ScaLabelCert on stand-alone models indicate that employing partition aggregation ensembles does not always bring out the true robustness potential of the chosen base classifier architecture. Overall, our findings motivate the development of effective white-box certificates for finite-width neural networks to bring out the true robustness of a partition aggregation ensemble and to understand the role of partition-based ensembling itself in achieving strong robustness guarantees.

## 6 ETHICS STATEMENT

Our work introduces EnsembleCert and ScaLabelCert, which, for the first time, leverage white-box information to quantify the worst-case robustness of partition aggregation ensembles of neural networks against label poisoning. Although such capabilities could, in principle, be misapplied by adversaries, we contend that understanding these vulnerabilities is essential for the trustworthy and safe use of neural networks. We therefore hold that the advantages of advancing robustness research outweigh the potential downsides, and we do not anticipate any immediate risks arising from our contributions. Radhakrishnan et al. (2022)

## 7 REPRODUCIBILITY STATEMENT

The full codebase, along with configuration files for every experiment, is available at https://figshare.com/s/f4ff623f9c47e63b8ef9, which will be made public upon acceptance.

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

# A  THEORETICAL DETAILS

## A.1  THE FORMAL DESCRIPTION OF THE WORST CASE SCENARIO

Recall that the prediction of the partition aggregation ensemble $g_{\mathcal{S}}(\mathbf{t})$ is determined by a majority vote over the prediction by the base classifiers $f_{\{1,\dots,N_p\}}$: $g_{\mathcal{S}}(\mathbf{t}) = \arg\max_{c \in [K]} n_c(\mathbf{t})$, where $n_c(\mathbf{t}) := |\{i \in [N_p] \mid f_i(\mathbf{t}) = c\}|$ is the number of votes received by class $c$. Ties are resolved deterministically by choosing the smaller index. If we denote $g_{\mathcal{S}}(\mathbf{t})$ as $c^*$, the certificate $\tilde{\rho}(\mathbf{t})$ for sample $\mathbf{t}$, that computes the number of adversarial label flips upto which the prediction of the ensemble will not change, as derived by black-box treatment of the base classifier is given as:

$$\tilde{\rho}(\mathbf{t}) := \Big\lfloor \frac{n_{c^*}(\mathbf{t}) - \max_{c' \neq c^*} (n_{c'}(\mathbf{t}) + \mathbf{1}_{c' < c^*})}{2} \Big\rfloor.$$

As each base classifier is treated as a black box, the certificate derivation follows from a key worst-case assumption: **The prediction of certain base classifiers can be altered by a *single* label flip**. The formal description of the worst-case scenario is given below.

**Formalising the worst-case scenario**: Let $\mathcal{C}_{\text{sec}} := \arg\max_{c \neq c^*} (n_c(t) + \mathbf{1}_{c < c^*})$. One can think of $\mathcal{C}_{\text{sec}}$ as the set of runner-up classes. We define $\mathcal{P}_{\text{maj}}$ as the set of base classifiers that voted for $c^*$. The worst-case scenario can be represented as:

$\exists\, c' \in \mathcal{C}_{sec}$ *s.t. the prediction of at least $\tilde{\rho} + 1$ base classifiers in $\mathcal{P}_{maj}$ can be changed from $c^*$ to $c'$ with one label flip in their corresponding partitions. In such a scenario, attacking the corresponding base classifiers with one label flip each would change the prediction of the ensemble to $c'$.*

Note that the numerator in $\tilde{\rho}$: $n_{c^*} - \max_{c' \neq c^*}(n_{c'}(t) + \mathbf{1}_{c' < c^*})$, is the difference in the number of the votes received by the majority class $c^*$ and $c'$. Flipping the vote of a base classifier from $c^*$ to $c'$ bridges the gap between $c^*$ and $c'$ by 2 votes, explaining the 2 in the denominator. In light of the worst-case assumption, the reader can now see that the certificate actually calculates the number of base classifiers whose prediction needs to be flipped in order to change the ensemble prediction. With *white-box* knowledge about the base classifiers, we can improve upon the worst-case assumption, leading to a tighter certificate for the ensemble. One could argue that we need the white-box information solely about the base classifiers in $\mathcal{P}_{\text{maj}}$ and classes in $\mathcal{C}_{\text{sec}}$ to challenge the assumption. The point to note is that if information about $\mathcal{P}_{\text{maj}}$ indicates that the worst-case scenario cannot be realized, we cannot assume that the adversary will attack partitions only in $\mathcal{P}_{\text{maj}}$ and change the prediction to a class in $\mathcal{C}_{\text{sec}}$. Hence, to derive a tighter certificate, we would need this information for all base classifiers and classes.

## A.2  THEOREM 1: $\min P_1(c) = \min P_2(c)$

**Intuition.** Recall that $P_1(c')$ denotes the minimum number of label flips needed to make $c'$ the *majority class*, whereas $P_2(c')$ denotes the minimum number of label flips needed to make $c'$ surpass the current majority class $c^*$ in number of votes. Intuitively, if $c'$ is the class that requires the fewest flips to become the new prediction, then making it just beat $c^*$ will already make it the majority class.

**Vote Configuration** Let $\tilde{\mathbf{V}} \in \{0, 1\}^{N_p \times K}$ denote the perturbed vote configuration, where $\tilde{v}_i^c = 1$ if partition $i$ votes for class $c$ after label flips, and 0 otherwise. Let $\mathbf{V}$ denote the clean vote configuration. We define $O(\tilde{\mathbf{V}})$ as the number of label flips required to reach configuration $\tilde{\mathbf{V}}$ starting from the clean configuration $\mathbf{V}$.

**Restatement of $P_1(c')$.** Recall that $P_1(c')$ is defined as the minimum number of label flips needed to make $c'$ the majority class. In Sec. 3.1, the contraint was formulated through the set of inequalities

$$\sum_{i=1}^{N_p} \left( \tilde{v}_i^{c'} - \tilde{v}_i^c \right) \geq \mathbf{1}_{c < c'}, \quad \forall c \neq c',$$

which constraints $c'$ to be the majority class (with deterministic tie-breaking). For brevity, we now re-express this condition using the function $\text{majVote}(\tilde{\mathbf{V}}) := \arg\max_{c \in [K]} \sum_{i=1}^{N_p} \tilde{v}_i^c$, as $c' = \text{majVote}(\tilde{\mathbf{V}})$ where ties are resolved deterministically by choosing the class with the smaller index.

With this shorthand notation, we write

$$P_1(c') = \min_{\tilde{\mathbf{V}}} O(\tilde{\mathbf{V}})$$

$$\text{s.t.} \quad c' = \text{majVote}(\tilde{\mathbf{V}}),$$

$$\sum_{c=1}^{K} \tilde{v}_i^c = 1, \quad \forall i \in [N_p],$$

$$\tilde{v}_i^c \in \{0,1\}, \quad \forall i \in [N_p], \ \forall c \in [K].$$

**Restatement of $P_2(c')$.** Problem $P_2(c')$ relaxes the above by requiring $c'$ to surpass *only* the original majority class $c^*$, instead of all classes:

$$P_2(c') = \min_{\tilde{\mathbf{V}}} O(\tilde{\mathbf{V}})$$

$$\text{s.t.} \quad \sum_{i=1}^{N_p} \left( \tilde{v}_i^{c'} - \tilde{v}_i^{c^*} \right) \geq \mathbf{1}_{c^* < c'},$$

$$\sum_{c=1}^{K} \tilde{v}_i^c = 1, \quad \forall i \in [N_p],$$

$$\tilde{v}_i^c \in \{0,1\}, \quad \forall i \in [N_p], \ \forall c \in [K].$$

**Theorem** (Restating Theorem 1).

$$\boxed{\min_{c' \in [K] \setminus \{c^*\}} P_1(c') = \min_{c' \in [K] \setminus \{c^*\}} P_2(c')}$$

*Proof.* We first state three lemmas and then combine them to prove the theorem.

**Lemma 1.** $\forall c' \in [K] \setminus c^*, \quad P_1(c') \geq P_2(c')$.

*Proof.* The feasible region of $P_1(c')$ is contained within that of $P_2(c')$ since the latter has a weaker constraint. Hence, $P_2(c')$ can only be smaller (or equal) to $P_1(c')$. $\qquad\square$

**Lemma 2.** $\forall c_1^* \in \arg\min_{c' \in [K] \setminus c^*} P_1(c'), \quad P_1(c_1^*) = P_2(c_1^*)$.

*Proof.* By contradiction. Suppose $P_1(c_1^*) > P_2(c_1^*)$. Let $\tilde{\mathbf{S}}$ be the optimal solution for $P_2(c_1^*)$, i.e.,

$$O(\tilde{\mathbf{S}}) = P_2(c_1^*). \tag{2}$$

Since $O(\tilde{\mathbf{S}}) < P_1(c_1^*)$, $\tilde{\mathbf{S}}$ is not feasible for $P_1(c_1^*)$. As the feasibility for $P_1(c_1^*)$ requires $c^*$ to be the majority class, there must exist some $c_s^* \neq c_1^*$ such that $c_s^* = \text{majVote}(\tilde{\mathbf{S}})$. Note that $\tilde{\mathbf{S}}$ is feasible for $P_1(c_s^*)$ as $c_s^*$ is the majority class for the vote configuration $(\tilde{\mathbf{S}})$, so

$$O(\tilde{\mathbf{S}}) \geq P_1(c_s^*). \tag{3}$$

Combining (2) and (3) with the assumption that $P_1(c_1^*) > P_2(c_1^*)$ gives $P_1(c_1^*) > P_1(c_s^*)$, contradicting the assumption that $c_1^*$ minimizes $P_1(c')$. $\qquad\square$

**Lemma 3.** $\exists z^* \in \arg\min_{c' \in [K] \setminus c^*} P_2(c')$ *such that* $P_1(z^*) = P_2(z^*)$.

*Proof.* Let $c_2^* \in \arg\min_{c' \in [K] \setminus c^*} P_2(c')$ and let $\tilde{\mathbf{S}}$ be the vote configuration in the optimal solution for $P_2(c_2^*)$, i.e,

$$O(\tilde{\mathbf{S}}) = P_2(c_2^*). \tag{4}$$

Let $z^* = \text{majVote}(\tilde{\mathbf{S}})$. Then $\tilde{\mathbf{S}}$ is feasible for both $P_1(z^*)$, that requires $z^*$ to be the majority class and $P_2(z^*)$, that requires $z^*$ to have higher votes than $c^*$, implying

$$O(\tilde{\mathbf{S}}) \geq P_2(z^*) \quad \text{and} \tag{5}$$

$$O(\tilde{\mathbf{S}}) \geq P_1(z^*). \tag{6}$$

With (4), (5) and the minimality of $c_2^*$, we have $z^* \in \arg\min_{c' \in [K] \backslash c^*} P_2(c')$ and $O(\tilde{\mathbf{S}}) = P_2(z^*)$. Combining this result with (6) and using Lemma 1.1 we conclude $P_1(z^*) = P_2(z^*)$. $\qquad\square$

**Proof of Theorem 1.** Let $c_1^* \in \arg\min_{c' \in [K] \backslash c^*} P_1(c')$ and $z^* \in \arg\min_{c' \in [K] \backslash c^*} P_2(c')$ $s.t.$ $P_1(z^*) = P_2(z^*)$. . Using Lemmas 1.2 and 1.3, we obtain

$$P_1(z^*) = P_2(z^*) \leq P_2(c_1^*) = P_1(c_1^*),$$

which implies

$$P_1(c_1^*) = P_1(z^*) = P_2(z^*),$$

thus proving that

$$\boxed{\min_{c' \in [K] \backslash \{c^*\}} P_1(c') = \min_{c' \in [K] \backslash \{c^*\}} P_2(c')}.$$

$\qquad\square$

### A.3 REDUCTION TO MCKP AND COMPLEXITY ANALYSIS

In this section, we will use the terms base classifiers and partitions interchangeably, and discuss the optimal attack from an adversary's point of view to make $c'$ surpass $c^*$. Let's denote the set of partitions that voted for $c^*$ originally as $\mathcal{P}_{\text{maj}}$, the ones that voted for $c'$ originally as $\mathcal{P}_{\text{target}}$ and rest of the partitions as $\mathcal{P}_{\text{rest}}$. Formally:

$$\mathcal{P}_{\text{maj}} = \{i \in [N_p] \mid v_i^{c^*} = 1\}, \quad \mathcal{P}_{\text{target}} = \{i \in [N_p] \mid v_i^{c'} = 1\}, \quad \mathcal{P}_{\text{rest}} = [N_p] \backslash (\mathcal{P}_{\text{target}} \cup \mathcal{P}_{\text{maj}}).$$

The adversary will not attack partitions in $\mathcal{P}_{\text{target}}$. If a partition in $\mathcal{P}_{\text{rest}}$ is attacked, the vote can change only to $c'$. Changing the vote to any other class will deem the label perturbation pointless. Let $\mathcal{C}_i$ be the set of classes that partition $i$ could vote for after the optimal attack:

$$\forall i \in \mathcal{P}_{\text{rest}}: \ \mathcal{C}_i = \{c \in K \mid c = c' \ or \ v_i^c = 1\} \ , \ \forall i \in \mathcal{P}_{\text{target}}: \ \mathcal{C}_i = \{c'\}.$$

Note that $\forall i \in [N_p]$, we need the binary variable $\tilde{v}_i^c$ only if $c \in \mathcal{C}_i$. Attacking a partition in $\mathcal{P}_{\text{maj}}$ could change the vote to $c'$ or to the class with the minimal number of flips required for a prediction change to that class. Formalizing the above notion, we define $c_{\min}$ as: $\forall i \in \mathcal{P}_{\text{maj}}: \ c_{\min}(i) = \arg\min_{c \in [K] \backslash c^*} \rho_i^c$. Given this we have:

$$\forall i \in \mathcal{P}_{\text{maj}}: \ \mathcal{C}_i = \{c \in K \mid c = c' \ or \ c = c_{\min}(i) \ or \ c = c^*\}.$$

We model the constraint $C_1 := \sum_{i=1}^{N_p} (\tilde{v}_i^{c'} - \tilde{v}_i^{c^*}) \geq \mathbf{1}_{c^* < c'}$ differently. Let $d$ be the original difference between the number of votes for $c'$ and $c^*$: $d = \sum_{i=1}^{N_p} (v_i^{c^*} - v_i^{c'})$. We define $r_i^c$ to be the reduction in the gap between $c'$ and $c^*$ caused by flipping the vote of partition $i$ to class $c$. For partitions in $\mathcal{P}_{\text{maj}}$, if the vote changes to $c'$, the difference will decrease by 2. If the vote goes to any other class, the reduction is by 1. It is trivial to see that $\forall i \in \mathcal{P}_{\text{target}}, c \in \mathcal{C}_i: \ r_i^c = 0$. We can similarly define these values for partitions in $\mathcal{P}_{\text{rest}}$ and get:

$$\forall i \in \mathcal{P}_{\text{maj}}, c \in \mathcal{C}_i: \ r_i^c = \begin{cases} 2 & \text{if } c = c' \\ 1 & \text{else if } c = c_{\min}(i) \\ 0 & \text{else if } c = c^* \end{cases} , \ \forall i \in \mathcal{P}_{\text{rest}}, c \in \mathcal{C}_i: \ r_i^c = \begin{cases} 1 & \text{if } c = c' \\ 0 & \text{else} \end{cases}$$

For $c'$ to have higher number of votes than $c^*$, the total reduction in the difference should be greater than or equal to $d + \mathbf{1}_{c^* < c'}$. Remodeling $C_1$ with the above idea, we can reformulate $P_2(c')$ as:

$$P_2(c') : \quad \min_{\tilde{\mathbf{V}}} \sum_{i=1}^{N_p} \sum_{c \in \mathcal{C}_i} \rho_i^c \, \tilde{v}_i^c \quad \text{s.t.} \quad \sum_{i=1}^{N_p} \sum_{c \in \mathcal{C}_i} r_i^c \, \tilde{v}_i^c \geq d + \mathbf{1}_{c^* < c'},$$

$$\forall i \in [N_p], \ \forall c \in \mathcal{C}_i : \quad \sum_{c \in \mathcal{C}_i} \tilde{v}_i^c = 1, \ \ \tilde{v}_i^c \in \{0,1\}.$$

This problem can be easily converted to a MCKP (multiple choice knapsack problem). To arrive at the excact formulation of MCKP, we need to change the $\min$ objective to a $\max$ objective and reverse the sign of the constraint inequality . Note that solving $\min_{\tilde{v}} O(\tilde{\mathbf{V}})$ is same as solving $A - \max_{\tilde{v}}(A - O(\tilde{\mathbf{V}}))$ , where $A$ is a positive constant. We choose the constant $A$ to be $N_p * \rho_{\max}$, where $\rho_{\max} = \max_{i \in N_p, c \in C_i} \rho_i^c$. Lets denote $P_3(c')$ as follows .

$$P_3(c') = \max_{\tilde{v}} (N_p * \rho_{\max} - \sum_{i=1}^{N_p} \sum_{c \in C_i} \rho_i^c \, \tilde{v}_i^c)$$

$$s.t. \quad (\sum_{i=0}^{N_p-1} \sum_{c \in C_i} r_i^c \, \tilde{v}_i^c) \geq d + \mathbf{1}_{c^* < c'},$$

$$\sum_{c \in C_i} \tilde{v}_i^c = 1, \quad \forall i \in [N_p],$$

$$\tilde{v}_i^c \in \{0,1\}, \quad \forall i \in [N_p], \forall c \in C_i$$

As $\sum_{c \in C_i} \tilde{v}_i^c = 1$ , we can rewrite $N_p * \rho_{\max}$ as $(\sum_{i=0}^{N_p-1} \sum_{c \in C_i} \rho_{\max} * \tilde{v}_i^c)$ . Using this trick, we reformulate $P_3(c')$ as :

$$P_3(c') = \max_{\tilde{v}} \sum_{i=1}^{N_p} \sum_{c \in C_i} (\rho_{\max} - \rho_i^c) \, \tilde{v}_i^c$$

$$s.t. \quad (\sum_{i=0}^{N_p-1} \sum_{c \in C_i} r_i^c \, \tilde{v}_i^c) \geq d + \mathbf{1}_{c^* < c'},$$

$$\sum_{c \in C_i} \tilde{v}_i^c = 1, \quad \forall i \in [N_p],$$

$$\tilde{v}_i^c \in \{0,1\}, \quad \forall i \in [N_p], \forall c \in C_i$$

We use the same trick to reverse the sign of the inequality. We will skip through the construction for the trick as it is exactly the same. Reformulating it finally gives us :

$$P_3(c') = \max_{\tilde{v}} \sum_{i=1}^{N_p} \sum_{c \in C_i} (\rho_{\max} - \rho_i^c) \, \tilde{v}_i^c$$

$$s.t. \quad \sum_{i=0}^{N_p-1} \sum_{c \in C_i} (r_{\max} - r_i^c) \, \tilde{v}_i^c \leq N_p * r_{\max} - (d + \mathbf{1}_{c^* < c'}),$$

$$\sum_{c \in C_i} \tilde{v}_i^c = 1, \quad \forall i \in [N_p],$$

$$\tilde{v}_i^c \in \{0,1\}, \quad \forall i \in [N_p], \forall c \in C_i$$

As we have explicitly specified the $r_i^c$ values, we can see that $r_{\max}$ is 2. The value of $d$ is upper bounded by $N_p$ as it is the difference in the number of votes. Thus, just as a sanity check, we can confirm that $N_p * r_{\max}$ - $(d + 1)$ is positive.

We can define $\rho_{eff} = \rho_{\max} - \rho$ and $r_{eff} = r_{\max} - r$ . Note that $r_{eff}$ and $\rho_{eff}$ are non-negative. Hence we have a MCKP with positive weights and profits .

$$
\begin{aligned}
P_3(c') = &\max_{\tilde{v}} \sum_{i=1}^{N_p} \sum_{c \in C_i} (\rho_{eff})_i^c \, \tilde{v}_i^c \\
&s.t. \quad \sum_{i=0}^{N_p-1} \sum_{c \in C_i} (r_{eff})_i^c \, \tilde{v}_i^c \le N_p * r_{\max} - (d + \mathbf{1}_{c^* < c'}), \\
&\sum_{c \in C_i} \tilde{v}_i^c = 1, \quad \forall i \in [N_p], \\
&\tilde{v}_i^c \in \{0, 1\}, \quad \forall i \in [N_p], \ \forall c \in C_i
\end{aligned}
$$

**Complexity.** The worst case complexity for solving the above problem is $\mathcal{O}((N_p * r_{\max} - (d + \mathbf{1}_{c^* < c'})) * \sum_{i=1}^{N_p} |\mathcal{C}_i|)$ (Dudzinski & Walukiewicz, 1987). Note that $\sum_{i=1}^{N_p} |\mathcal{C}| \le 3 * N_p$. Hence, the worst case complexity of solving the MCKP for our use case is $\mathcal{O}(N_p^2)$. $P_2(c')$ can be computed as $N_p * \rho_{\max} - P_3(c')$. We derive the certificate for the ensemble by solving $P_2(c')$ for every class and finding the minimum. Thus, we derive ensemble-level guarantees by aggregating the white-box certificates from the base classifiers in $\mathcal{O}(K * N_p^2)$.

## A.4 SVM SIMPLIFICATION FOR SUFFICIENTLY SMALL $C$

**Theorem** (Restating Theorem 2). *Given a soft-margin SVM with penalty parameter $C$, kernel matrix entries $Q_i^j$, and dual solution $\boldsymbol{\alpha}$ to $P_{svm}(\mathbf{y})$, if*

$$C \left( \max_{i \in [n]} \sum_{j=1}^{n} |Q_i^j| \right) - 1 \leq 0,$$

*then for all label assignments $\mathbf{y} \in \{-1, 1\}^n$ we have:*

$$\boldsymbol{\alpha} = C \cdot \mathbf{1}^n.$$

*That is, all dual variables are equal to $C$, independent of the choice of labels.*

*Proof.* We restate the dual formulation of the soft-margin SVM optimization problem for completeness. Given training labels $\mathbf{y} \in \{-1, 1\}^n$ and kernel matrix entries $Q_i^j$, the dual problem is:

$$P_{\text{svm}}(\mathbf{y}) = \min_{\boldsymbol{\alpha}} \left( -\sum_{i=1}^{n} \alpha_i + \frac{1}{2} \sum_{i=1}^{n} \sum_{j=1}^{n} y_i y_j \alpha_i \alpha_j Q_i^j \right) \quad \text{s.t.} \quad 0 \leq \alpha_i \leq C \ \forall i \in [n].$$

The gradient of the objective $P_{\text{svm}}(\mathbf{y})$ with respect to $\alpha_i$ is:

$$\frac{\partial P_{\text{svm}}(\mathbf{y})}{\partial \alpha_i} = \sum_{j=1}^{n} y_i y_j \alpha_j Q_i^j - 1.$$

Over the feasible domain $0 \leq \alpha_j \leq C \ \forall j \in [n]$, we can bound the derivative as:

$$\frac{\partial P_{\text{svm}}(\mathbf{y})}{\partial \alpha_i} \leq C \sum_{j=1}^{n} |Q_i^j| - 1.$$

Now, if $C \left( \max_{i \in [n]} \sum_{j=1}^{n} |Q_i^j| \right) - 1 \leq 0$, then for every $i \in [n]$:

$$\frac{\partial P_{\text{svm}}(\mathbf{y})}{\partial \alpha_i} = \sum_{j=1}^{n} y_i y_j \alpha_j Q_i^j - 1 \leq 0.$$

This implies that $P_{\text{svm}}(\mathbf{y})$ is monotonically decreasing in each $\alpha_i$ over the feasible set. Hence, the minimum is attained at the boundary:

$$\alpha_i = C \quad \forall i \in [n].$$

Thus, under the stated condition on $C$, the solution is $\boldsymbol{\alpha} = C \cdot \mathbf{1}^n$, *regardless of the choice of labels* $\mathbf{y} \in \{-1, 1\}^n$. $\square$

## A.5 SCALABELCERT FOR THE BINARY SETTING

Recall that for the binary setting, we wish to find the minimum number of label flips required to change the prediction of a soft-margin SVM that uses a sufficient small $C$ (as described in App. A.4). Under sufficiently small $C$, the SVM prediction $\hat{p}_t$ on a test sample $\mathbf{t}$ simplifies to $\hat{p}_t = \sum_{i=1}^{n} y_i Q_{ti}$. We denote the perturbed training labels as $\tilde{\mathbf{y}}$. Then, the number of label flips required to get the perturbed labels $\tilde{\mathbf{y}}$ from the clean labels $\mathbf{y}$ can be formulated as $\frac{1}{2} \sum_{i=1}^{n} (1 - y_i \tilde{y}_i)$. For the prediction $p_t$ to change when the model is trained on the perturbed labels, the sign of the clean prediction $\hat{p}_t$ and the tamperd prediction $p_t = \sum_{i=1}^{n} \tilde{y}_i Q_{ti}$ should be opposite. With this information we can formulate our objective as:

$$O(\mathbf{y}): \quad \min_{\tilde{\mathbf{y}} \in \{-1,1\}^n} \frac{1}{2} \sum_{i=1}^{n} \big(1 - y_i \tilde{y}_i\big) \quad \text{s.t.} \quad \text{sign}(\hat{p}_t) \sum_{i=1}^{n} \tilde{y}_i Q_{ti} < 0.$$

We show that $O(\mathbf{y})$ can be solved in polynomial time. The intuition being: The labels corresponding to the largest positive contributions in $\text{sign}(\hat{p}_t)(\sum_{i=1}^{n} y_i Q_t^i)$ are the most influential in determining the prediction, so flipping these labels greedily till the prediction changes is the optimal attack from the adversary's point of view.

**Proof.** We define the prediction margin to be the sum $S = \sum_{i=1}^{n} \text{sign}(\hat{p}_t) y_i Q_t^i$. Note that $S$ is always positive as we have included the $\text{sign}(\hat{p}_t)$ inside the sum. The prediction for the SVM trained on the perturbed labels $\tilde{\mathbf{y}}$ will change when $\tilde{S} = \sum_{i=1}^{n} \text{sign}(\hat{p}_t) \cdot \tilde{y}_i Q_t^i$ becomes negative. Let $a_i = \text{sign}(\hat{p}_t) \cdot y_i \cdot Q_{ti} \ \forall i \in [n]$. Thus, $S = \sum_{i=1}^{n} a_i$. Flipping a subset of the clean training labels $F \in 2^{[n]}$ to get the perturbed labels $\tilde{\mathbf{y}}$ changes the $i$th term $a_i$ to $-a_i$ for $i \in F$, resulting in

$$\tilde{S} = S - 2 \sum_{i \in F} a_i.$$

The prediction changes when $\tilde{S}$ is negative, i.e, $\sum_{i \in F} a_i > S/2$. Hence, $O(\mathbf{y})$ reduces to finding the smallest subset $F$ such that satisfies the above condition.

**Greedy algorithm.** Construct $W = (a_1, \ldots, a_n)$ and sort it in descending order: $a_{(1)} \geq a_{(2)} \geq \cdots \geq a_{(n)}$. Let $P_k = \sum_{j=1}^{k} a_{(j)}$ be the cumulative sum of the largest $k$ elements. We find the smallest $k'$ such that $P_{k'} > S/2$ and construct the set $F$ by including the labels corresponding to $a_{(1)}, \ldots, a_{(k')}$. Note that $\forall k < k': \ P_k < S/2$. We claim that $F$ is the minimal set that we want and $k$ is the minimum number of flips required to change the prediction of the SVM. By construction we ensure that $\tilde{S}$ corresponding to the label flips in $F$ is negative. We prove that that $F$ is the minimal set by contradiction. Assume there exists a subset $F' \in 2^{[n]}$ with $|F| = m \leq k' - 1$ such that flipping the labels in $F'$ results in changing the prediction of the SVM , i.e, $\sum_{i \in F} a_i > S/2$. Note that $\sum_{i \in F} a_i$ can be only as large as $P_{k'-1}$, which is the sum of the $k'-1$ largest elements in $W$. But $P_{k'-1}$ is less than $S/2$ as $\forall k < k': \ P_k < S/2$. This contradicts the requirement $\sum_{i \in F'} a_i > S/2$. Thus, $F$ is the minimal subset and $k$ is the minimum number of label flips required to change the SVM prediction.

**Complexity.** Sorting $W$ requires $O(n \log n)$, and scanning for $k$ is $O(n)$. Hence $O(\mathbf{y})$ is solvable in $O(n \log n)$ time, i.e., in polynomial time. Thus, ScaLabelCert provides a polynomial-time computable exact certificate for sufficiently-wide neural networks, when their NTK is used as the SVM kernel. The certificate for kernel regression can be derived similarly by replacing $Q_t^i$ by $(Q_{eff})_t^i$, where $(Q_{eff})_t^i$ can be obtained by a minor modification described in Sec. 3.2.

## A.6 EXACT CERTIFICATE FOR MULTICLASS WITHOUT PARTITIONING PROOF

For the multi-class case, we use the one-vs-all strategy by decomposing the problem with $K$ classes into $K$ separate binary classification tasks. For each class $c \in [K]$, a binary classifier is trained to distinguish between samples of class $c$ and samples from all other classes. Assume that $p_c$ is the prediction score of a classifier for the learning problem corresponding to class $c$. Then, the class prediction $c^*$ for a test sample is constructed by $c^* = \arg\max_{c \in [K]} p_c$. The labels are collected in the vector $\mathbf{y} \in \{0, 1\}^{n \times K}$ where $\mathbf{y}_i^c = 1$ if the class of the $i$th sample is $c$ and 0 otherwise. Recall that for a test sample $\mathbf{t}$, the certificate $\tilde{\rho}(\mathbf{t})$ denotes the maximum number of label flips up to which the prediction for the classifier does not change. We derive the certificate by finding for every class $c' \in [K]$, the minimum number of label flips required to change the prediction of the classifier to a particular class $c'$, and then taking the minimum over $c'$. The number of label flips to reach the perturbed label $\tilde{\mathbf{y}}$ from the clean labels $\mathbf{y}$ can be represented as $\sum_{i=1}^{N}(1 - \sum_{c=1}^{K} y_i^c \tilde{y}_i^c)$. For the class $c'$ to be the predicted class, the score $p_{c'}$ for class $c'$ should exceed the score $p_c$ for every other class $c$. Using a soft-margin kernel SVM with a sufficiently small $C$ as our base model, the score $p_c$ can be written as $p_c = \sum_{i=1}^{N} y_i^c Q_{ti}$, . With this information, the minimum number of label flips required to change the prediction of the model to a particular class $c'$ can be formulated as:

$$
O_1(c') : \quad \min_{\tilde{\mathbf{y}}} \sum_{i \in [N]} \left( 1 - \sum_{c \in [K]} y_i^c \tilde{y}_i^c \right) \quad \text{s.t.} \quad \sum_{i \in [n]} \tilde{y}_i^{c'} Q_t^i > \sum_{i \in [N]} \tilde{y}_i^c Q_t^i \quad \forall c \neq c',
$$
$$
\forall i \in [n], c \in [K] : \sum_{c \in [K]} \tilde{y}_i^c = 1 \ , \tilde{y}_i^c \in \{0, 1\}. \tag{7}
$$

The certificate $\tilde{\rho}(\mathbf{t})$ can be calculated as: $\tilde{\rho}(\mathbf{t}) = \min_{c' \in [K] \setminus \{c^*\}} O_1(c') - 1$. $O_1(c')$ is a Integer Linear Program(ILP) with $n \times K$ binary variables. Hence, the complexity for solving $O_1(c')$ the problem is $\mathcal{O}(2^{n \times K})$. Consequently, the complexity for deriving $\tilde{\rho}(\mathbf{t})$, which is calculated by taking the minimum over $O_1(c')$, is $\mathcal{O}(K \times 2^{n \times K})$. This prompts us to solve a simpler alterative $O_2(c')$ instead, that relaxes the constraint in $O_1(c')$ requiring $c'$ to be the predicted class:

$$
O_2(c') : \quad \min_{\tilde{\mathbf{y}}} \sum_{i \in [N]} \left( 1 - \sum_{c \in [K]} y_i^c \tilde{y}_i^c \right) \quad \text{s.t.} \quad \sum_{i \in [n]} \tilde{y}_i^{c'} Q_t^i > \sum_{i \in [N]} \tilde{y}_i^{c^*} Q_t^i \ ,
$$
$$
\forall i \in [n], c \in [K] : \sum_{c \in [K]} \tilde{y}_i^c = 1 \ , \tilde{y}_i^c \in \{0, 1\}. \tag{8}
$$

$O_2(c')$ calculates the minimum number of label flips needed to make the prediction score $p_{c'}$ for class $c'$ exceed the prediction score $p_{c^*}$ for the original predicted class $c^*$. Notably, this relaxation is similar to the relaxation of $P_1(c')$ to $P_2(c')$ in the context of computing the certificate for the ensemble. Despite the relaxation, we show that the minimum over $O_1(c')$ is preserved, i.e.:

**Theorem 3.**
$$
\min_{c' \in [K] \setminus c^*} O_1(c') = \min_{c' \in [K] \setminus c^*} O_2(c')
$$

The intuition being — while trying to make $c'$ surpass $c^*$, if another class $c''$ becomes the predicted class, then changing the classifier prediction to $c''$ should be easier compared to $c'$. As the design of the relaxation and the intuition are similar to the ensemble case, the proof strategy for this result is exactly the same as Theorem 1. The only difference would be that the notation **majVote**$(\tilde{\mathbf{V}})$, that finds the majority class for a vote configuration $\tilde{\mathbf{V}}$ will be replaced by the **majScore** notation that predicts the class when the model is trained on the perturbed labels $\tilde{\mathbf{y}}$, i.e, **majScore**$(\tilde{\mathbf{y}}) = \arg\max_{c \in [K]} \sum_{i \in [n]} \tilde{y}_i^c Q_t^i$. Hence, we direct the reader to the proof for Theorem 1 provided in App. A.2. With the above result, we can derive the certificate $\tilde{\rho}(\mathbf{t})$ as: $\tilde{\rho}(\mathbf{t}) = \min_{c' \in [K] \setminus \{c^*\}} O_2(c') - 1$

**Solving $O_2(c')$.** We focus our attention on solving $O_2(c')$. Recall that $O_2(c')$ denotes the minimum number of label flips needed to make the prediction score $p_{c'}$ for class $c'$ exceed the prediction score $p_{c^*}$ for the original predicted class $c^*$.

We denote the training samples that were labeled $c^*$ originally as $P_{\text{maj}}$ and samples that were labeled $c'$ originally as $P_{\text{target}}$. $P_{\text{rest}}$ denote the set of remaining samples

$$P_{\text{maj}} = \{i \in [N] \mid y_i^{c^*} = 1\}, \quad P_{\text{target}} = \{i \in [N_p] \mid y_i^{c'} = 1\}, \quad P_{\text{rest}} = [n] \setminus P_{\text{maj}} \cup P_{\text{target}}$$

Let $d$ be the original difference between score for $c'$ and $c^*$: $d = \sum_i (y_i^{c^*} - y_i^{c'})Q_{ti}$. We define $r_i^c$ to be the **reduction** caused by flipping label of the $i$ th sample to class $c$ in the difference between score for $c'$ and $c^*$. For the score $p_{c'}$ exceed the score $p_{c^*}$, the total reduction caused by the label flipping attack should exceed $d$. Lets see what these values will be for different $i$ and $c$. **For samples in $P_{\text{maj}}$.** For samples that were originally labeled $c^*$, if the label is flipped to $c'$, reduction will be $2 * Q_{ti}$. If the label is flipped from $c^*$ to any other class, the reduction is only by $Q_{ti}$ as it affects the score only for $c^*$. If $Q_{ti}$ is positive, flipping the label to $c'$ will cause the maximum reduction possible by flipping the $i$th label. Hence, an optimal attack will flip the $i$th label to $c'$ (if it chooses to flip the label). If $Q_{ti}$ is negative, an optimal attack will not flip the label for the $i$th sample as it will further increase the gap between $c'$ and $c^*$.

$$\forall i \in P_{\text{maj}}, c \in [K], \quad r_i^c = \begin{cases} 2 * Q_{ti} & \text{if } c = c' \\ 0 & \text{else if } c = c^* \\ Q_{ti} & \text{else} \end{cases}$$

**For samples in $P_{\text{target}}$.** Following a similar line of argument as above, we can safely say that $\forall i \in P_{\text{target}}$, if $Q_{ti} < 0$, an optimal attack will flip the label for sample $i$ to $c^*$ (if it chooses to flip the label). If $Q_{ti}$ is positive, an optimal attack will not flip the label for the $i$th sample as it will further increase the gap between $c'$ and $c^*$.

$$\forall i \in P_{\text{target}}, c \in [K], \quad r_i^c = \begin{cases} -2 * Q_{ti} & \text{if } c = c^* \\ 0 & \text{else if } c = c' \\ -Q_{ti} & \text{else} \end{cases}$$

**For samples in $P_{\text{rest}}$.** For samples with the true label other than $c'$ or $c^*$, if $Q_{ti} > 0$, an optimal attack will flip the $i$th label to $c'$. If $Q_{ti}$ is negative, the optimal attack will flip the $i$th label to $c^*$, if it chooses to flip the label.

$$\forall i \in P_{\text{rest}}, c \in C_i, \quad r_i^c = \begin{cases} Q_{ti} & \text{if } c = c' \\ -Q_{ti} & \text{if } c = c^* \\ 0 & \text{else} \end{cases}$$

With this, we can define $r(i)$ as the reduction in the difference between scores for $c'$ and $c^*$ in the optimal attack, if the attacker chooses to flip the $i$th label: $\forall i \in [N], r(i) = \max_{c \in [K]} r_i^c$. Note that if $r_i$ is 0, the $i$th label will not be flipped. Hence we define the set of candidates for flipping the labels as $E_f$: $E_f = \{i \in [N] \mid r(i) > 0\}$

We employ a greedy strategy similar to the one used in the computation of the certificate for the binary case (App. A.5). We first sort the candidate flipping labels $E_f$ based on their $r(i)$ values in the descending order, i.e., $r_{(1)} \geq r_{(2)} \geq \dots$. Let $P_k = \sum_{j=1}^k r_{(j)}$ be the cumulative sum of the largest $k$ elements. We find the smallest $k'$ such that $P_{k'} > d$ and construct $G_{k'}$ by including the indices corresponding to the $k'$ largest $r(i)$ values. We claim that $k'$ is the minimum number of label flips required to make the score for $c'$ exceed $c^*$ and $G_{k'}$ is the minimal set of the labels that need to be flipped to make it happen. The greedy algorithm is illustrated in Algorithm 1.

We prove optimality by contradiction. Recall that $G_k = \{(1), \dots, (k)\}$ is the greedy choice of $k$ largest $r(i)$ and $P_k = \sum_{j=1}^k r_{(j)}$. Suppose there exists a set $F \subseteq E_f$ with $|F| = m \leq k' - 1$ such that flipping labels in $F$ achieves the objective, i.e, $\sum_{i \in F} r(i) > d$. But $G_m$ consists of the $m$ largest $r(i)$, so $\sum_{i \in F} r(i) \leq \sum_{i \in G_m} r(i) = P_m \leq P_{k'-1} \leq d$, a contradiction.

---

**Algorithm 1** Greedy Certificate Computation

---

**Input:** Score gap $d$, reductions $r(1), \ldots, r(n)$
**Output:** Minimum number of flips $k$

1: total_red $\leftarrow 0$
2: sorted $\leftarrow$ sort$(r, \text{descending})$
3: $i \leftarrow 0$
4: **while** total_red $< d$ **do**
5:     total_red $\leftarrow$ total_red $+$ sorted$[i]$
6:     $i \leftarrow i + 1$
7: **end while**
8: **return** $i$

---

**Complexity.** Sorting based on the reduction values takes $\mathcal{O}(n \log n)$, and scanning for the minimal $k'$ takes $\mathcal{O}(n)$. Solving $O_2(c')$ for all $c' \in [K] \setminus c^*$ can be done in $\mathcal{O}(Kn \log n)$, resulting in a polynomial-time **exact** certificate for test sample $\mathbf{t}$: $\tilde{\rho}(\mathbf{t}) = \min_{c' \neq c^*} O_2(c') - 1$.

## A.7 EXTRACTING WHITE-BOX KNOWLEDGE FOR THE MULTI-CLASS SETTING

Recollect that EnsembleCert utilizes the white-box knowledge $\rho_i^c$ for all base classifiers $i \in [N_p]$ and all classes $c \in [K]$ where $\rho_i^c$ denotes the minimum number of flips required to change the prediction of the $i$th base classifier to $c$. For each base classifier, using a soft-margin kernel SVM with a sufficiently small $C$ as our base model, we formulate the problem of finding the minimum number flips required to change the prediction of the classifier to $c'$ is formulated as:

$$O_1(c'): \quad \min_{\tilde{\mathbf{y}}} \sum_{i \in [N]} \left( 1 - \sum_{c \in [K]} y_i^c \tilde{y}_i^c \right) \quad \text{s.t.} \quad \sum_{i \in [n]} \tilde{y}_i^{c'} Q_t^i > \sum_{i \in [N]} \tilde{y}_i^c Q_t^i \quad \forall c \neq c',$$
$$\forall i \in [n], c \in [K]: \quad \sum_{c \in [K]} \tilde{y}_i^c = 1 \quad , \tilde{y}_i^c \in \{0,1\}. \tag{9}$$

The above problem is an ILP with $n \times K$ binary variables, resulting in a computational complexity of $\mathcal{O}(2^{nK})$. For each base classifier, we need to solve this problem for all classes. Consider an ensemble with $N_p$ base classifiers. We would need to solve $N_p \times K$ ILPs with the computational complexity of $O_1(c')$, making the problem intractable. Hence, instead of solving the problem $O_1(c')$ exactly, we bound the problem efficiently, and show that our bounds are sufficiently tight on empirical evaluation.

### A.7.1 LOWER BOUND

For the purpose of computing the exact certificate for the multiclass case in the no partition case, we formulated an alternate problem $O_2(c')$ (Eq. (8)), a relaxed version of $O_1(c')$ which only requires the score $p_{c'}$ for class $c'$ to exceed the score $p_{c^*}$ for class $c^*$. We showed in App. A.6 that $O_2(c')$ can be solved in polynomial time. As the constraint set of $O_2(c')$ is a subset of $O_1(c')$, the solution for $O_2(c')$ will always be less than or equal to $O_1(c')$. Hence, $O_2(c')$ represents a valid lower bound on $O_1(c')$, that is polynomial-time calculable. Notably, $O_2(c')$ is also a certificate by definition, as it is a lower bound on the certified radius.

### A.7.2 UPPER BOUND

To compute a valid upper bound on $O_1(c')$, finding an instance of perturbed labels $\tilde{\mathbf{y}}$ that satisfies the constraints of $O_1(c')$, i.e, **majScore**$(\tilde{\mathbf{y}}) = c'$, is sufficient. The number of label flips needed to reach any such $\tilde{\mathbf{y}}$ from the clean labels $\mathbf{y}$ represents a valid upper bound on $O_1(c')$. To compute this upper bound efficiently, we adopt a greedy strategy that iteratively flips training labels to reach a feasible solution of Eq. (1). At the beginning of each iteration, we compute the current majority class $c^* = \arg\max_c S_c$, where $S_c = \sum_{i=1}^{n} \tilde{y}_i^c Q_t^i$ is the score of class $c$. Note that $c^*$ may change after each label flip, and our method accounts for this by re-evaluating $c^*$ and all per-sample damages $d_i$ after every label flip. Let $S^{(2)} = \max_{c \in [K] \setminus c^*} S_c$ denote the score of the runner-up class. For each sample $i$, we define the *per-sample damage* $d_i$ as the maximum possible reduction in the gap between $c'$ and the majority class achievable by flipping $i$:

$$d_i = \begin{cases} \min\left(2Q_t^i, \ Q_t^i + S_{c^*} - S^{(2)}\right), & \text{if } \tilde{y}_i^{c^*} = 1 \text{ and } Q_t^i > 0, \\ \min\left(2|Q_t^i|, \ |Q_t^i| + S_{c^*} - S^{(2)}\right), & \text{else if } \tilde{y}_i^{c'} = 1 \text{ and } Q_t^i < 0, \\ Q_t^i, & \text{if } \tilde{y}_i^{c^*} = 0, \ \tilde{y}_i^{c'} = 0, \text{ and } Q_t^i > 0, \\ 0, & \text{otherwise.} \end{cases}$$

Intuitively, $d_i$ captures the maximal contribution that flipping sample $i$ can make toward satisfying the class-change constraint of Eq. (1).

**Cases explained**

1. **Case 1**($\tilde{y}_i^{c^*} = 1, Q_t^i > 0$): Flipping a positively contributing $c^*$-labeled sample to $c'$ both reduces $S_{c^*}$ and increases $S_{c'}$, leading to a decrease of $2Q_t^i$. The possibility that the runner-up class becomes the majority is considered by the term $Q_t^i + S_{c^*} - S^{(2)}$, which accounts for the score gap to the second-highest class.

2. **Case 2** ($\tilde{y}_i^{c'} = 1, Q_t^i < 0$): Flipping a negatively contributing $c'$-labeled sample to $c'$ helps both by increasing $S_{c'}$ and decreasing $S_{c^*}$, giving a decrease of $2|Q_t^i|$. The possibility of runner-up class becoming the majority class is handled as above.

3. **Case 3 (neutral sample, $Q_t^i > 0$)**: Flipping such a sample to $c'$ only increases $S_{c'}$, so the reduction in the gap is exactly $Q_t^i$.

In each iteration, we select $i^\star = \arg\max_i d_i$, flip the corresponding label to $c'$ or $c^*$, re-evaluate the majority class, and repeat this process until $c'$ becomes the majority class. Hence, by design, the greedy algorithm results in a feasible $\tilde{\mathbf{y}}$ satisfying Eq. (1), and the number of flips performed constitutes a valid upper bound on $O_1(c')$. Our greedy approach is illustrated in Algorithm 2.

**Complexity.** We choose the label to flip by calculating the possible reduction each label flip can cause in the gap between $c^*$ and $c'$, and choosing the one that causes the maximum reduction. This involves scanning the dataset at every iteration. Hence the worst case complexity in computing the upper bound for $O_1(c')$ is $\mathcal{O}(n^2)$.

---

**Algorithm 2** Greedy Upper Bound Computation for $O_1(c')$

---

1: **Input:** Clean labels $\mathbf{y}$, kernel entries corresponding to the $i$th training sample and test sample $t$: $Q_t^i \, \forall i \in [N]$, target class $c'$
2: **Output:** Upper bound on $O_1(c')$ (number of label flips)
3: Initialize $\tilde{\mathbf{y}} \leftarrow \mathbf{y}$
4: Compute $S_c = \sum_{i=1}^N \tilde{y}_i^c Q_t^i$ for all $c \in [K]$
5: $c^* \leftarrow \arg\max_c S_c$
6: **while** $c^* \neq c'$ **do**
7: $\quad S^{(2)} \leftarrow \max_{c \neq c^*} S_c$
8: $\quad$ **for** $i = 1$ to $N$ **do**
9: $\quad\quad$ **if** $\tilde{y}_i^{c^*} = 1$ and $Q_t^i > 0$ **then**
10: $\quad\quad\quad d_i \leftarrow \min\left(2Q_t^i, \, Q_t^i + S_{c^*} - S^{(2)}\right)$
11: $\quad\quad$ **else if** $\tilde{y}_i^{c'} = 1$ and $Q_t^i < 0$ **then**
12: $\quad\quad\quad d_i \leftarrow \min\left(2|Q_t^i|, \, |Q_t^i| + S_{c^*} - S^{(2)}\right)$
13: $\quad\quad$ **else if** $\tilde{y}_i^{c^*} = 0$ and $\tilde{y}_i^{c'} = 0$ and $Q_t^i > 0$ **then**
14: $\quad\quad\quad d_i \leftarrow Q_t^i$
15: $\quad\quad$ **else**
16: $\quad\quad\quad d_i \leftarrow 0$
17: $\quad\quad$ **end if**
18: $\quad$ **end for**
19: $\quad i^\star \leftarrow \arg\max_i d_i$
20: $\quad$ **if** $\tilde{y}_{i^*}^{c'} = 1$ **then**
21: $\quad\quad \tilde{y}_{i^*}^{c^*} \leftarrow 1$
22: $\quad$ **else**
23: $\quad\quad \tilde{y}_{i^*}^{c'} \leftarrow 1$
24: $\quad$ **end if**
25: $\quad$ Update $S_c = \sum_{i=1}^N \tilde{y}_i^c Q_t^i$ for all $c \in [K]$
26: $\quad c^* \leftarrow \arg\max_c S_c$
27: **end while**
28: **return** Number of flips applied to reach $\tilde{\mathbf{y}}$

---

## B    DETAILS OF RANDOMIZED SMOOTHING INTEGRATION

### B.1    SMOOTHED LINEAR CLASSIFIER AS THE BASE MODEL

In addition to ScaLabelcert for sufficiently wide networks, we use the method from Rosenfeld et al. (2020) for certification of base classifiers. Their approach uses a smoothed linear classifier as the base classifier. The smoothing process involves independently flipping each label with probability $q$ and assigning a flipped label uniformly at random among the remaining $K - 1$ classes. To certify robustness, the method bounds the probability that this randomized classifier switches its prediction from one class to another.

### B.2    PREDICTION BY THE SMOOTHED CLASSIFIER

The method by Rosenfeld et al. (2020) computes, for each pair of classes $(c, c')$, a Chernoff bound $p_{c,c'}(q)$ that gives an upper-bound on the probability that the randomized classifier switches its predicted class from $c$ to $c'$ under the randomized label flips.

For each class $c$, the method evaluates

$$\max_{c' \neq c} p_{c,c'}(q)$$

and defines the predicted class as

$$c^* = \arg \min_{c \in [K]} \max_{c' \neq c} p_{c,c'}(q),$$

### B.3    COMPUTING THE CERTIFICATE

The certified radius $r$ is obtained by plugging the worst-case probability bound $\max_{c' \neq c^*} p_{c^*,c'}(q)$ into the robustness guarantee from a result in Rosenfeld et al. (2020), giving

$$r \leq \frac{\log \left( 4p(1-p) \right)}{2(1-2q) \log \left( \frac{q}{1-q} \right)},$$

where $p = \max_{c' \neq c^*} p_{c^*,c'}(q)$. This bound guarantees that if at most $r$ labels were flipped byt the adversary, the smoothed classifier would still predict $c^*$.

### B.4    ADAPTATION FOR OUR USE CASE

We follow the same prediction rule to obtain $c^*$, However, instead of computing the radius that certifies that the prediction will not change to *any* other class, we focus on certifying robustness against a specific target class $c'$. This is because our objective is to compute the minimum number of label flips required to change the prediction of a base classifier to *every* class. This white-box information is then used by EnsembleCert to construct a white-box infused certificate for the ensemble.

Concretely, we use the pairwise Chernoff bound $p_{c^*,c'}(q)$ and compute

$$r_{c'} \leq \frac{\log \left( 4p_{c^*,c'}(1 - p_{c^*,c'}) \right)}{2(1-2q) \log \left( \frac{q}{1-q} \right)},$$

which gives the number of label flips required to change the prediction specifically from $c^*$ to $c'$.

## C    ADDITIONAL PLOTS

### C.1    FURTHER IMPLEMENTATION DETAILS AND CERTIFICATION RUNTIME

**Hardware.**    All the experiments were done on an internal cluster. We used GPUs solely for the NTK kernel computation, which was done using the Google `neural-tangents` library (Novak et al., 2020). As the kernel computation is not the main focus of our work, we refer interested readers to (Novak et al., 2020) for details on latency and memory requirements. All the following steps for certificate derivation were executed on CPUs.

**Certificate derivation.**    We process the test data in parallel batches of 100 samples. Recall that for each test sample, EnsembleCert first computes $\rho_i^c$ for every base classifier $i \in [N_p]$ and class $c \in [K]$ using ScaLabelCert, which provides both upper and lower bounds. This requires computing $N_p \times K$ entries before passing this information to EnsembleCert for aggregation. In the current implementation, white-box information is computed sequentially by iterating over the partitions and classes. However, these computations are inherently parallelizable across both partitions and classes because the certificates are independent. In particular, the lower bound calculation for each $\rho_i^c$ (App. A.7.1) can be fully vectorized, whereas the upper bound calculation (App. A.7.2) must be performed independently for each sample. The aggregation step, which combines white-box information to derive the ensemble-level certificate, is also executed independently per sample. This step involves solving a Multiple-Choice Knapsack Problem (MCKP) for each test sample, which to the best of our knowledge, cannot be vectorized efficiently.

**Average Certification Time per Sample.**    As discussed, both EnsembleCert and ScaLabelCert yield polynomial-time computable certificates. Since lower bound computations are vectorized within each batch, per-sample latency cannot be measured directly. Instead, we report average amortized time, which refers to the total time taken to certify a batch divided by the number of samples in that batch, providing a fair per-sample estimate. We present this average amortized certification time per sample for EnsembleCert in Fig. 4. Latencies presented in the figure above also include the training and prediction latencies, which are negligible owing to the simplification under small $C$ for kernel SVM and the closed form solution for kernel Regression. The total certification time per sample is the sum of the latencies for the upper and lower bound computations. Importantly, the upper bound latency is not amortized since the computation is performed sequentially per sample.

For small $N_p$, the number of samples per partition is high, which leads to a noticeable gap between the latencies of upper and lower bound computations. This is because $(i)$ the lower bound computation is linear in the number of samples whereas the upper bound computation is quadratic and $(ii)$ lower bounds computation is vectorized whereas upper bound computation is done independently. As $N_p$ increases, white-box aggregation latency for both computations becomes dominant due to the quadratic complexity of MCKP in $N_p$, thereby narrowing the gap between the upper and lower bound curves. For CIFAR-10, the average (amortized) certification time per sample goes from as low as 5 seconds for $N_p = 50$ to as high as under 2 minutes for $N_p = 500$. For MNIST, the average amortized latency varies from 8 sec to a max of around 4 min.

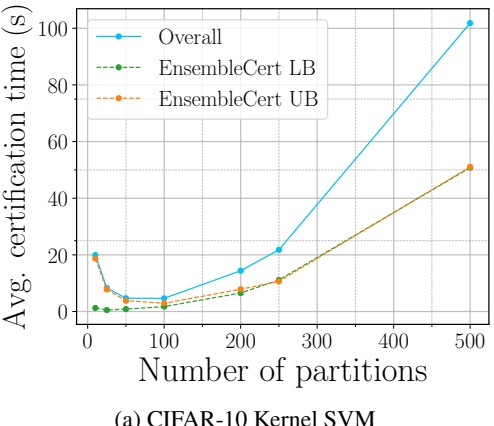

(a) CIFAR-10 Kernel SVM

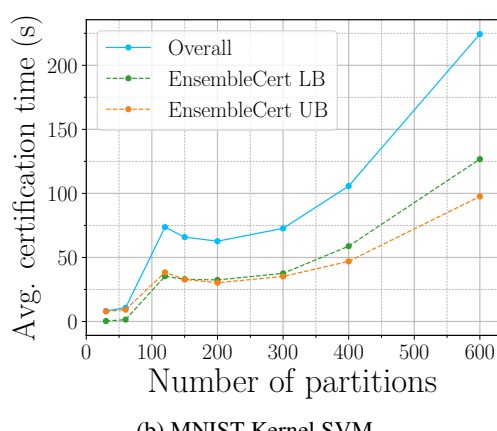

(b) MNIST Kernel SVM

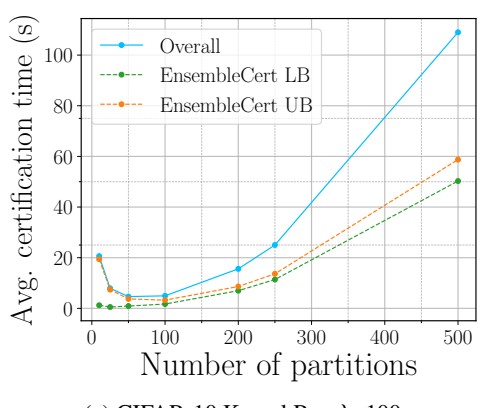 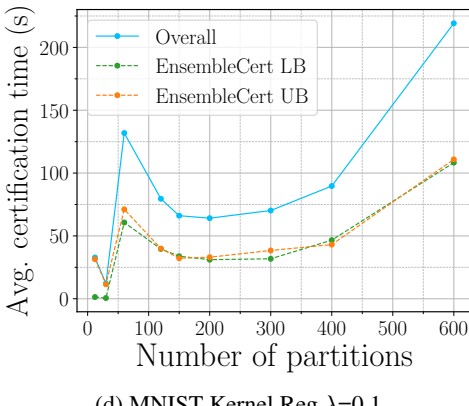

(c) CIFAR-10 Kernel Reg $\lambda$=100       (d) MNIST Kernel Reg $\lambda$=0.1

Figure 4: Figures (a) and (b) show the average latency per sample for kernel SVM on CIFAR-10 and MNIST, respectively. Figures (c) and (d) present the corresponding results for kernel regression. We report results for a single value of $\lambda$ for each dataset, as the trends are consistent across different $\lambda$ values and do not provide any additional qualitative insight.

**Choosing $N_p$.** Sec. 4 discussed the invariance of robustness to partitioning observed for EnsembleCert with kernel SVMs, and the robustness decay observed with sufficiently regularized kernel regression. This raises questions about the practical utility of using heavy partitioning. Moreover, the experiments in Fig. 4 show that larger ensembles are computationally more expensive to certify using EnsembleCert. When robust base classifiers are employed, it becomes evident that using a low to medium number of partitions offers the best trade-off between achieving high certified robustness and minimizing computational costs.

## C.2 ROBUSTNESS-ACCURACY TRADEOFF FOR SMALL C

Recall that the key to obtaining polynomial-time computable certificates by ScaLabelCert for infintely-wide networks trained on the hinge loss is choosing a sufficiently small value of $C$. Hence, this introduces a robustness-accuracy tradeoff as we are constrained to choose $C$ that is sufficiently small to achieve the polynomial-time certificate. To study this tradeoff, we perform 5-fold cross-validation for different values of $C$. For every number of partitions, the accuracy initially remains constant up to a certain threshold, indicating the range of sufficiently small $C$, as SVM performance does not depend on $C$ in this regime. The results for CIFAR-10 are shown in Fig. 6. In addition to studying the tradeoff on an ensemble level, we conduct experiments to study the performance trade-off induced by "sufficiently small C" for stand-alone classifiers as a function of the training set size $N_s$. For each value of $N_s$, we sub-sample the training set for 5 different random seeds while keeping class balance. The models trained on the sub-sampled training data are evaluated on the entire test dataset. The results can be seen in Fig. 5. It is evident from these experiments that performance remains competitive in the small $C$ regime. We do not evaluate on MNIST, as for our chosen kernel and RotNet preprocessing, the threshold for sufficiently small $C$ is significantly larger (order of $10^2$).

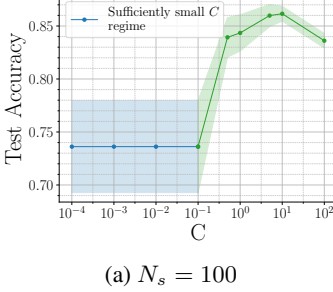 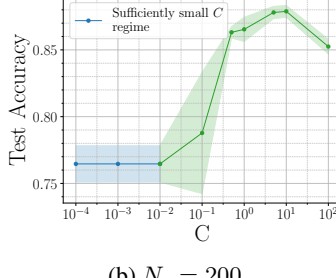 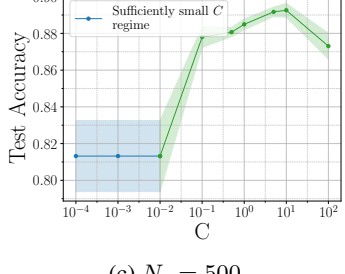

(a) $N_s = 100$       (b) $N_s = 200$       (c) $N_s = 500$

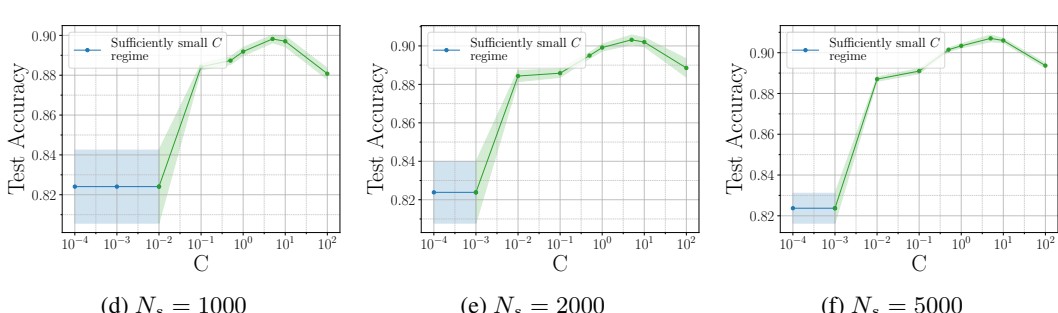

(d) $N_s = 1000$      (e) $N_s = 2000$      (f) $N_s = 5000$

Figure 5: Performance of an infinitely-wide neural network with a single trained on hinge loss across different values of $C$ and the cardinality of the training dataset $N_s$. For each value of $N_s$, we subsample the training set for 5 different random seeds. The accuracy for each value of $N_s$ initially remains constant up to a certain threshold for $C$, indicating the range of sufficiently small $C$, as the SVM performance does not depend on C in this regime. Shading around the central line indicates the standard deviation. In each plot, the region of sufficiently small $C$ is colored blue for distinction. For each plot, the accuracy initially remains constant up to a certain threshold, indicating the range of sufficiently small $C$, as SVM performance does not depend on $C$ in this regime.

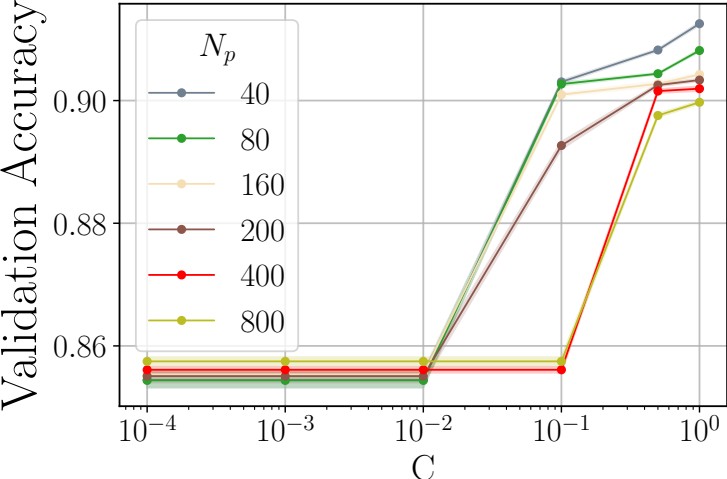

Figure 6: Robustness-accuracy trade-off introduced by sufficiently small C. While Fig. 5 studies empirical performance of stand-alone models, the impact of small $C$ ion the empirical performance of the ensemble is studies here.

## C.3 COMPARISON OF SCALABELCERT WITH THE GRADIENT-BASED BOUNDING CERTIFICATE (SOSNIN ET AL., 2024)

We compare the performance of ScaLabelCert with the gradient-based parameter bounding method proposed by Sosnin et al. (2024) on CIFAR-10. The gradient-based approach uses convex relaxations to over-approximate all possible parameter updates under a given poisoning threat model. The parameter bounds are then propagated to bound the logits for individual classes. The robustness of the prediction for a particular sample can then be certified by checking if the lower bound on the output logit for the predicted class is greater than the upper bounds of all other classes given the perturbation budget. If this condition is satisfied, the prediction is certifiably robust under the given budget. To evaluate this method, we add a linear layer on top of the CIFAR-10 features extracted via SimCLR. We train the network for 2 epochs and keep other parameters consistent with the codebase for Sosnin et al. (2024). For ScaLabelCert, the same SimCLR features are input to an infinitely wide fully-connected network with a single hidden layer and no non-linear activation, as described in Sec. 4. Although the models are not identical, the architectures are structurally aligned as both models rely on fully connected layers without activations, making the comparison meaningful. As shown in App. C.3, ScaLabelCert consistently outperforms the gradient-based method. This improvement highlights the importance of exact certification: while the gradient-based approach is inherently limited by the looseness of its over-approximations, ScaLabelCert provides tight guarantees, resulting in stronger certified robustness.

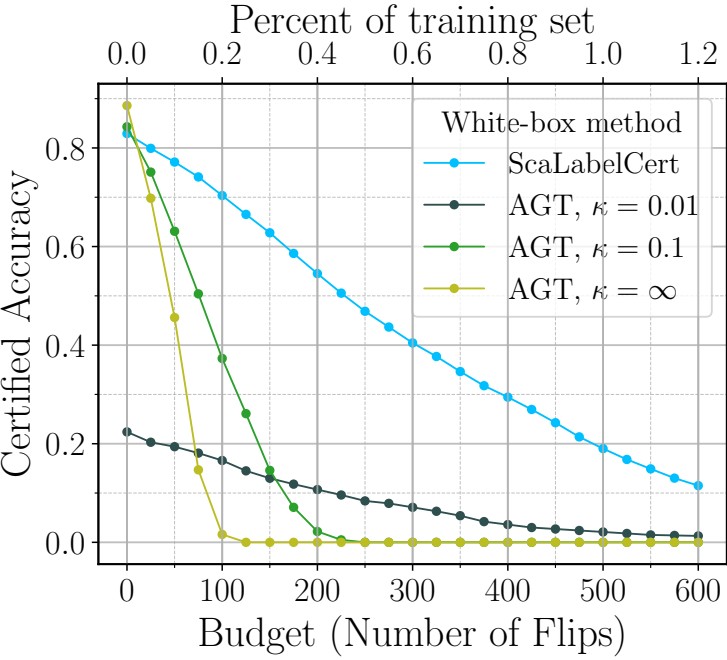

Figure 7: Comparison of ScaLabelCert and gradient-based parameter bounding method. The parameter $\kappa$ represents the gradient clipping parameter. ScaLabelCert significantly outperforms the gradient-based method for all values of $\kappa$.

## C.4 ENSEMBLECERT WITH FINITE-WIDTH NETWORKS AS BASE CLASSIFIERS

The certificates derived by ScaLabelCert are asymptotically exact and deterministic for neural networks as the width of the network goes to infinity. Hence, ScaLabelCert is best suited for certifying infinite-width neural networks. To demonstrate that EnsembleCert can provide deterministic certificates even when finite-width networks are used as base classifiers, we instantiate EnsembleCert with a fully connected linear classifier as the base model and utilize the gradient-based parameter bounding method by Sosnin et al. (2024), introduced in the previous section, to certify the base classifiers. The gradient-based method certifies, for a given sample, whether the model's prediction remains unchanged when at most $r$ training labels are flipped. However, this guarantee does not directly match the white-box quantity required by EnsembleCert, which is the minimum number of label flips needed to change the prediction to a *particular* class. In what follows, we explain how this gradient-based certificate can be incorporated into EnsembleCert..

**Certificate alignment.** For any certificate that verifies whether a model's prediction remains robust under a fixed perturbation budget of $r$ label flips, one can obtain a certificate for the *minimum* number of label flips needed to change the prediction to *some* class by applying the fixed-budget certificate incrementally—starting at $r = 0$ and increasing $r$ until the certificate first indicates non-robustness. Note that EnsembleCert requires white-box information quantifying the minimum number of label flips needed to change the prediction of a base classifier to a *specific* target class. The certificate that computes the minimum flips needed to change the prediction to *some* class then serves as a valid lower bound on $\rho_i^c$, the minimum number of label flips required to force the $i$-th base classifier to predict a particular class $c$. This relationship enables us to incorporate the gradient-based certificate into EnsembleCert.

**Experiments.** We evaluate EnsembleCert on ensembles with finite-width linear networks as base classifiers by applying the gradient-based certificate to each base classifier. As described in the previous section, this certificate can be incorporated into EnsembleCert. Briefly, for each sample we compute the minimum number of label flips required to change the prediction of base classifier $i$ to *some* class by applying the gradient-based certificate over increasing budgets until robustness fails. We then use this value as $\rho_i^c$ for every class $c$. Because this incremental application of the gradient-based method is computationally demanding, our experiments on finite-width models are limited to CIFAR-10 and a small number of partitions. Nevertheless, as shown in Fig. 8, incorporating white-box information through EnsembleCert substantially improves certified accuracy. This demonstrates that EnsembleCert extends naturally to ensembles built from finite-width neural networks. Moreover, the integration procedure highlights a broader utility: any certificate that determines whether a model remains robust up to a given number of label flips can be adapted to certify the base classifiers within EnsembleCert.

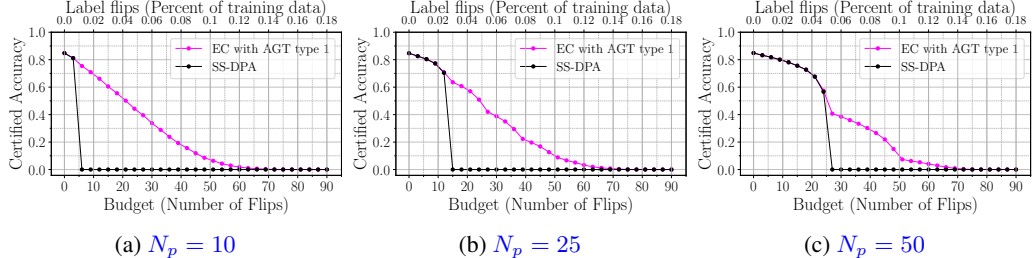

(a) $N_p = 10$      (b) $N_p = 25$      (c) $N_p = 50$

Figure 8: EnsembleCert evaluated on ensembles with finite-width network as base classifiers. The base classifiers are certified by applying the gradient-based bounding technique by Sosnin et al. (2024). Evidently, EnsembleCert significantly outperforms SS-DPA, the black-box approach applied to the ensemble.

## C.5 Robustness trends across $\lambda$ for EnsembleCert with Kernel Regression

We study the effect of the $\mathcal{L}_2$ regularization parameter $\lambda$ on certified robustness. For low $\lambda$, the MCR increases with the number of partitions across all datasets. In contrast, for high $\lambda$, the **certified robustness appears to be negatively affected by increasing the number of partitions**. The contrasting behavior potentially arises due to the varying degree of robustness that the choice of $\lambda$ imparts the base classifier. Low $\lambda$ makes kernel regression unstable, causing base classifiers to be easily influenced by a few label flips, even with large partitions. In this regime, the ensemble is closer to the black-box assumption - that a single label flip can change a base classifier's prediction - even when the number of partitions is small, resulting in white-box certificates that aren't much tighter than the black-box ones. In contrast, using a high degree of regularization exhibits a substantial improvement in the certified accuracy on white-box infusion. Fig. 10 shows how the robustness trend changes with varying $\lambda$ for CIFAR-10. While at $\lambda = 0.01$, the median certified robustness scales almost linearly with the number of partitions, the trend completely changes by the time we reach $\lambda = 100$. Empirical analysis suggests that the nature of the trend changes somewhere between $\lambda = 1$ and $\lambda = 5$. Interestingly, the change in trend also points towards the possibility that for some value between $\lambda = 1$ and $\lambda = 5$, the median certified robustness could exhibit invariance to the number of partitions, a phenomenon we observed with kernel SVMs. Similar behavior is seen for experiments on MNIST as well (Fig. 9). As we mentioned in Sec. 4, the threshold $\lambda$, where the behavior changes is data dependent.

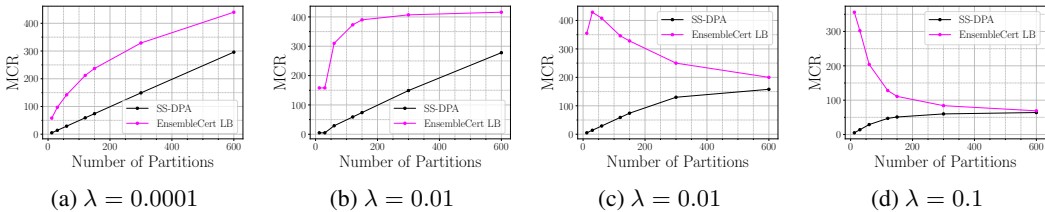

(a) $\lambda = 0.0001$      (b) $\lambda = 0.01$      (c) $\lambda = 0.01$      (d) $\lambda = 0.1$

Figure 9: Robustness trends across different $\lambda$ values on MNIST using kernel regression.

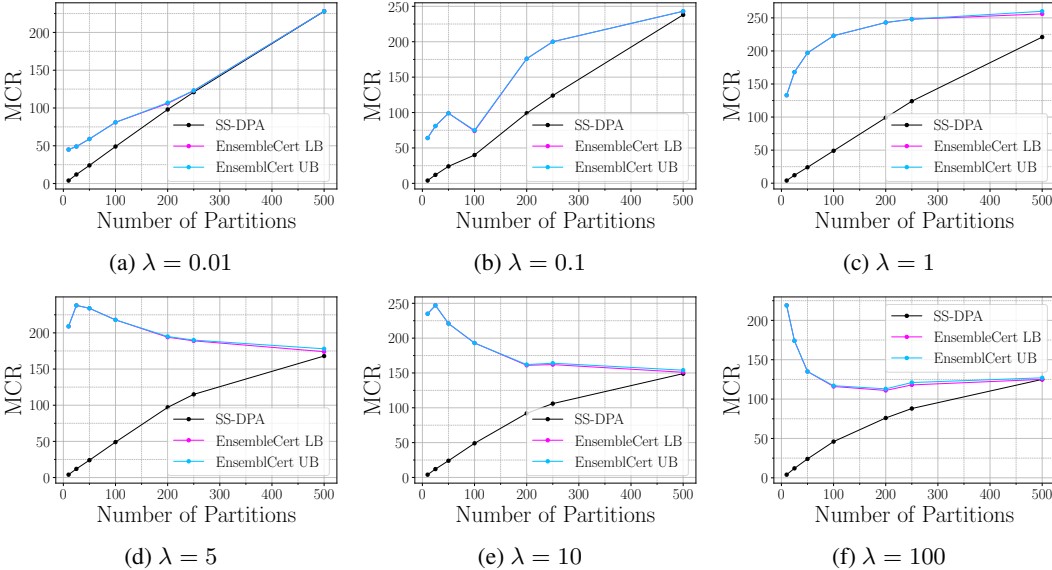

(a) $\lambda = 0.01$      (b) $\lambda = 0.1$      (c) $\lambda = 1$

(d) $\lambda = 5$      (e) $\lambda = 10$      (f) $\lambda = 100$

Figure 10: Robustness trends across different $\lambda$ values on CIFAR-10 using kernel regression.

## C.6    MNIST 1-vs-7

**Kernel Regression.** We present results for evaluation using kernel regression with NTK and the $\ell_2$ regularization parameter $\lambda = 0.1$ in Fig. 11.

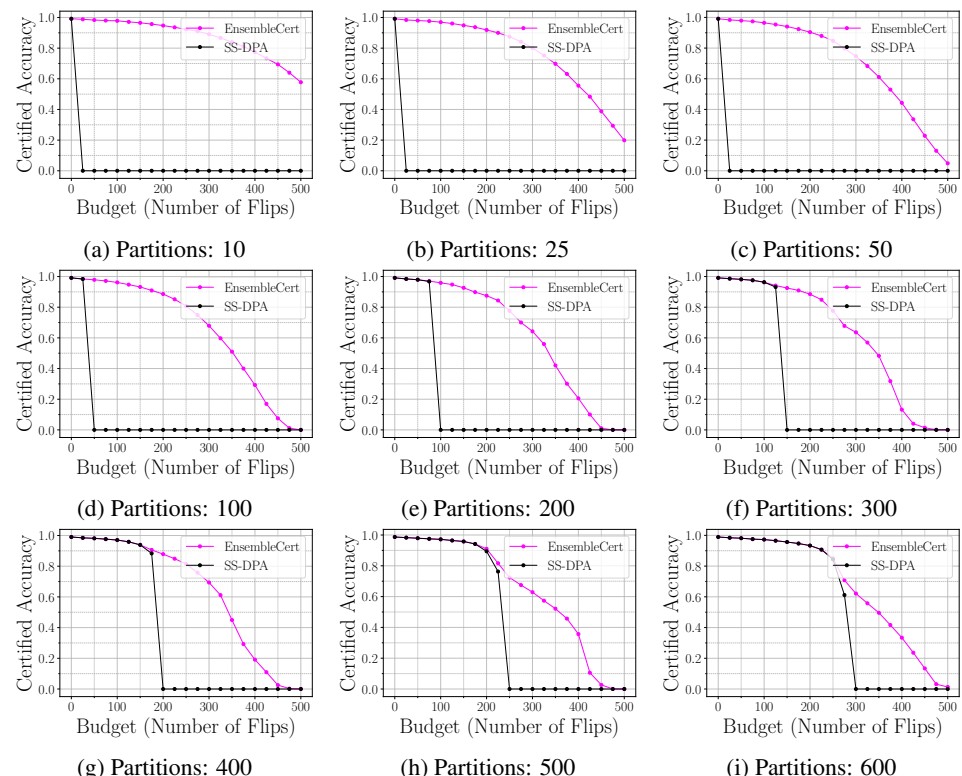

Figure 11: MNIST binary kernel regression with $\lambda = 0.1$ results for different number of partitions.

**Kernel SVM**

We present results for evaluation using kernel SVM with NTK and a sufficiently small $C$ in Fig. 12.

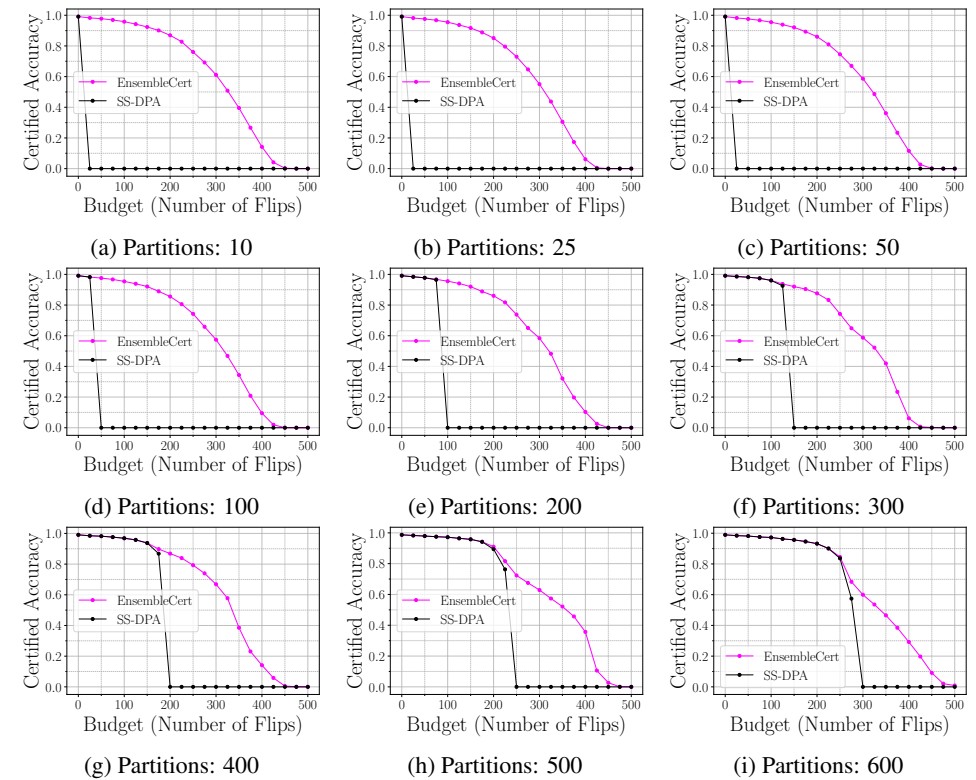

Figure 12: MNIST binary kernel SVM results for different number of partitions.

**MCR results** Consistent with the results on MNIST and CIFAR10, Fig. 13 demonstrates the invariance of median certified robustness to the number of partitions, for both kernel Reg and kernel SVM

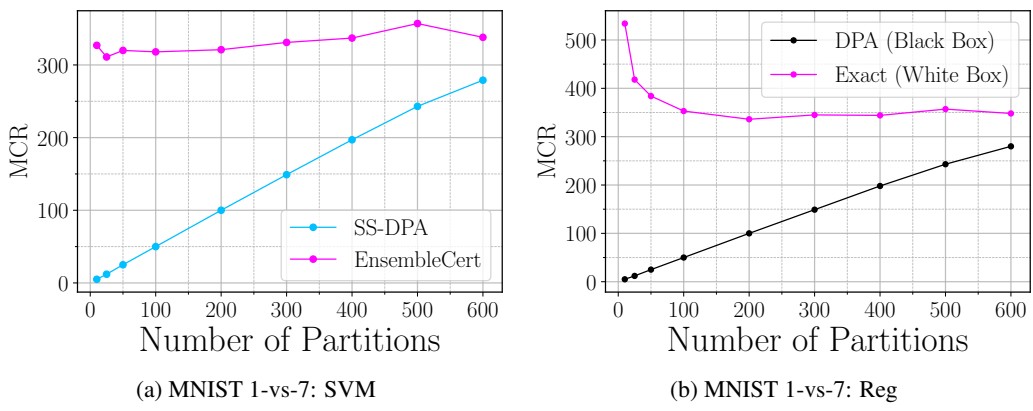

Figure 13: Invariance to number of partitions

### C.7 CIFAR-10

**Kernel Regression**

For low values of $\lambda$, the base classifier by itself is not robust and is closer to the worst-case black box assumption described in App. A.1. Consequently, we do not see a significant improvement on utilizing white-box knowledge of the base classifiers. This is illustrated in Fig. 14 , where we evaluate EnsembleCert using kernel regression as base classifier and a low regularization parameter $\lambda = 0.01$. In contrast, using a high degree of regularization changes the trend as mentioned in the experiments section. In this case, we see a substantial improvement in the certified accuracy on white-box infusion. The results on using a high lambda can be seen in Fig. 15.

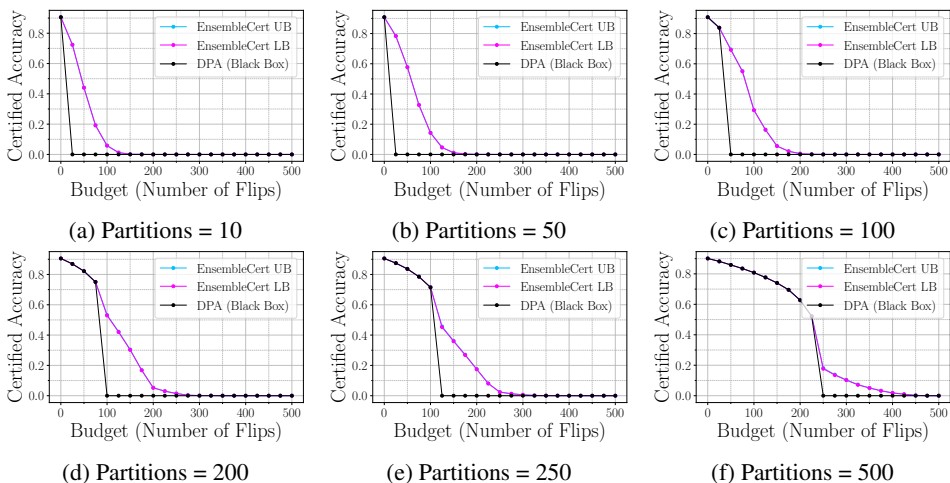

Figure 14: CIFAR-10 kernel regression results for different numbers of partitions ($\lambda = 0.01$).

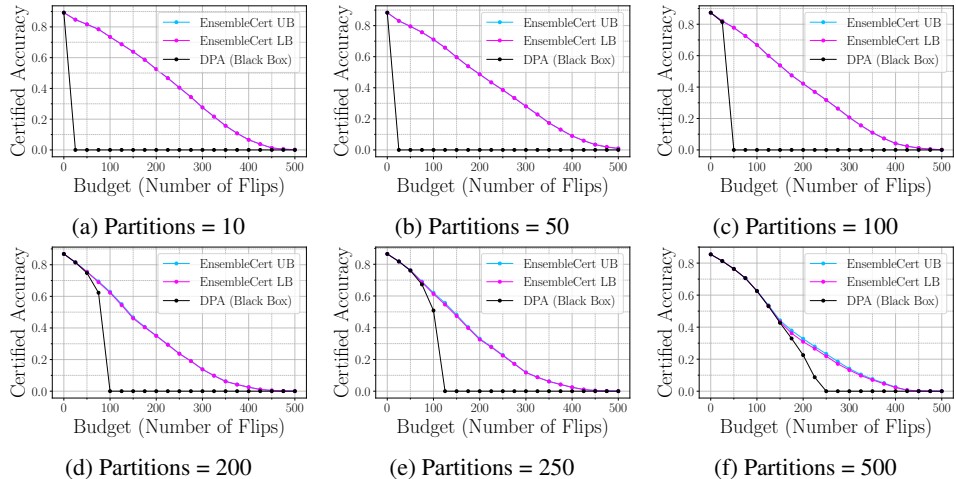

Figure 15: CIFAR-10 kernel regression with high regularization ($\lambda = 10$).

**Kernel SVM**

Substantial improvement can be observed in certified accuracy on white-box infusion, as seen in Fig. 16.

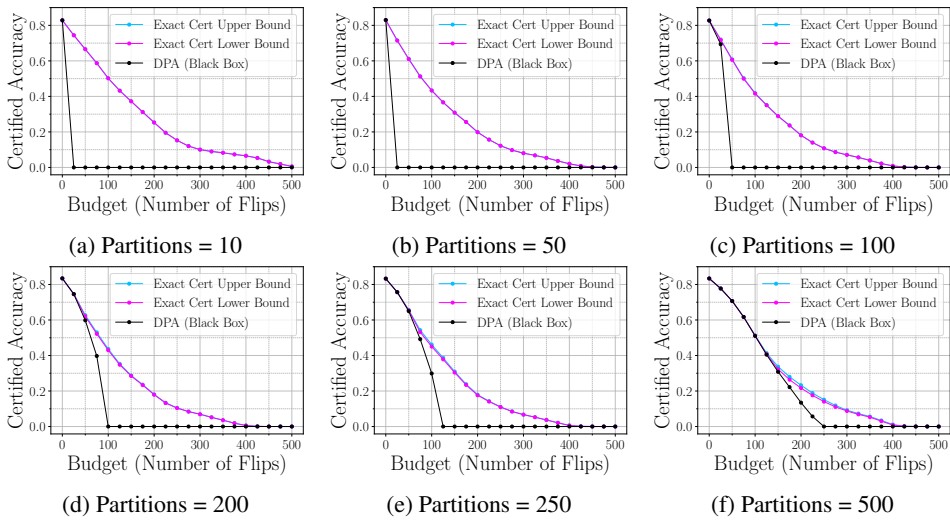

Figure 16: CIFAR-10 kernel SVM results for different numbers of partitions.

**Smoothed linear classifier**

Similar to kernel regression and kernel SVM, substantial improvement is observed in certified accuracy on white-box infusion, as shown in Fig. 17.

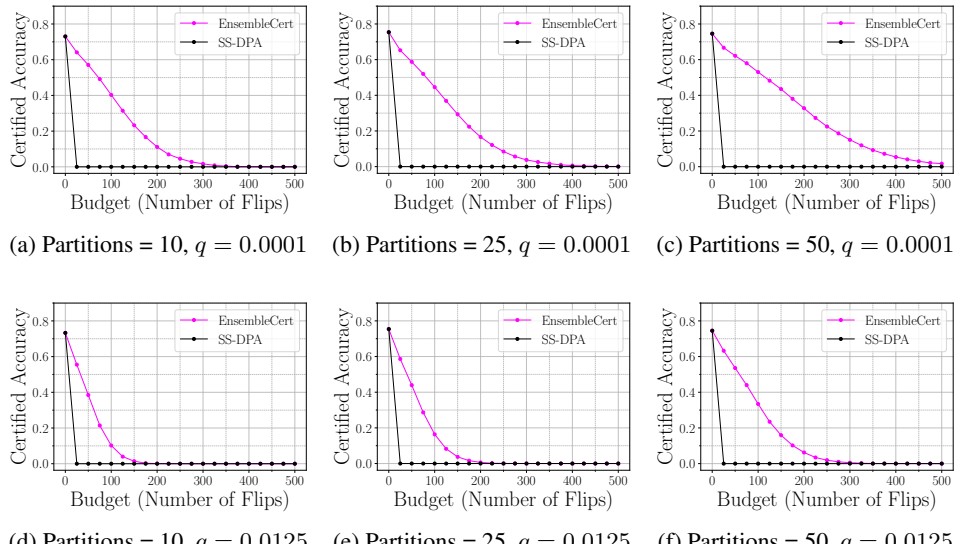

Figure 17: CIFAR-10 Smoothed linear regression as base-classifier

## C.8 MNIST

**Kernel Regression**

We analyze the low and high regularization parameter $\lambda$ on MNIST. Similar to CIFAR-10, we observe that for low values of $\lambda$, the base classifier by itself is not robust and is closer to the worst-case black box assumption described in App. A.1. Consequently, we do not see a significant improvement in utilizing white-box knowledge of the base classifiers. This is illustrated in Fig. 18 using low $\lambda = 0.0001$. In contrast, using a high degree of regularization changes the trend. In this case, we see a substantial improvement in the certified accuracy on white-box infusion. The results on using a high lambda can be seen using high $\lambda = 0.1$ in Fig. 19.

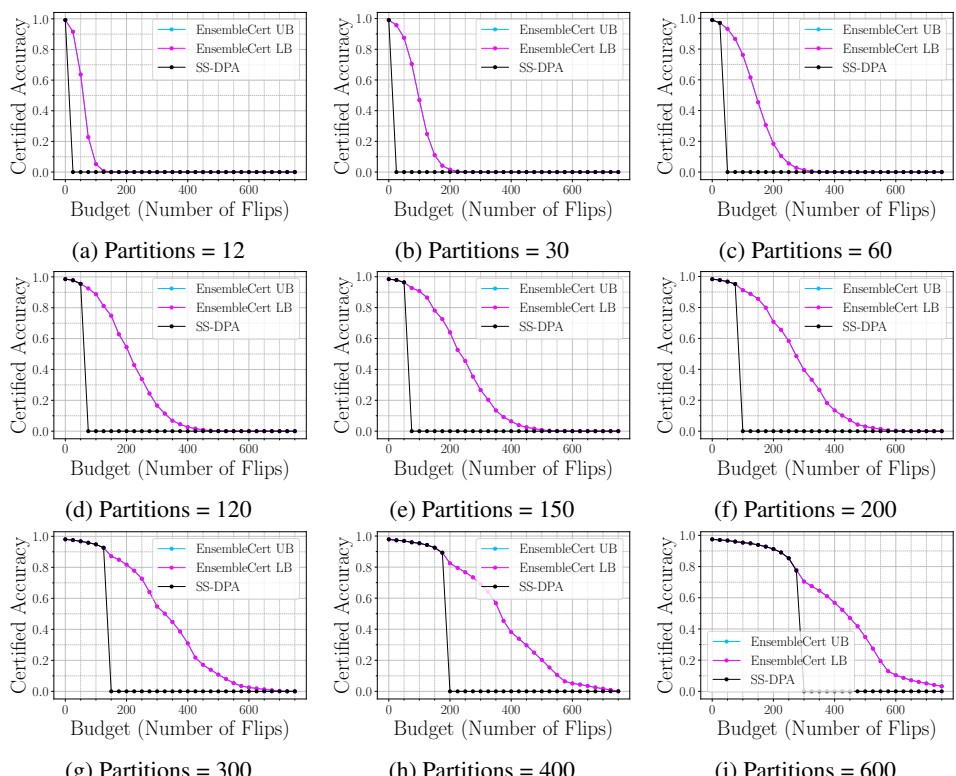

Figure 18: MNIST multi-class kernel regression results for different numbers of partitions ($\lambda = 0.0001$).

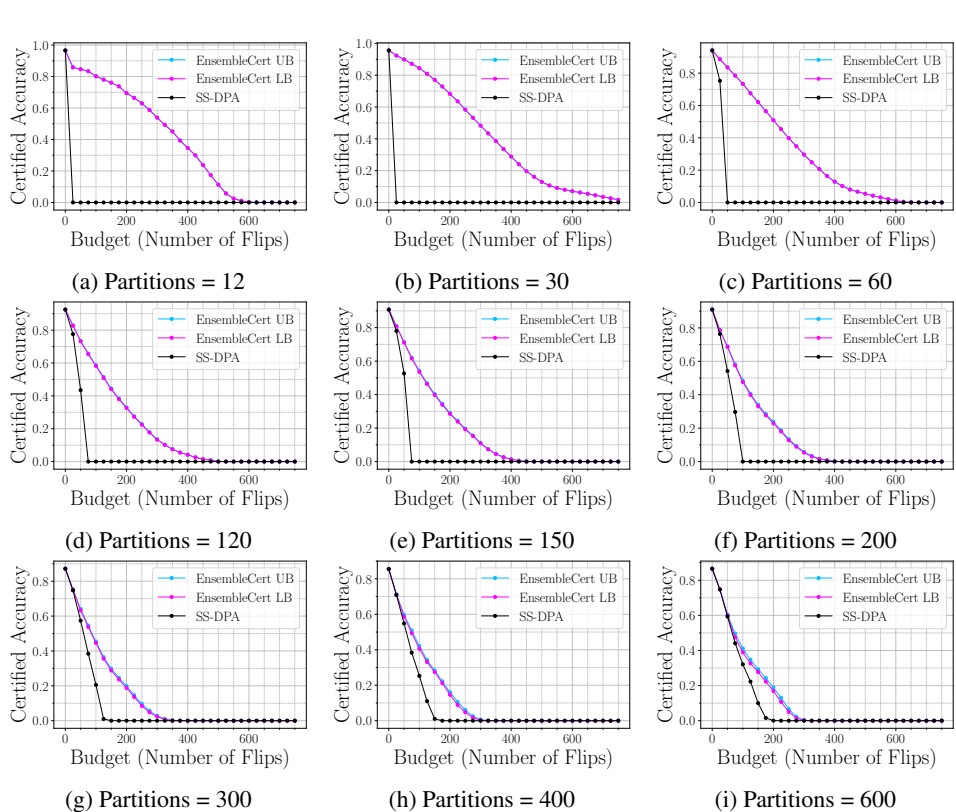

Figure 19: MNIST multi-class kernel regression results for different numbers of partitions ($\lambda = 0.1$).

**Kernel SVM**

Results showing substantial improvement in the certified accuracy on white-box infusion for kernel SVM are observed in Fig. 20.

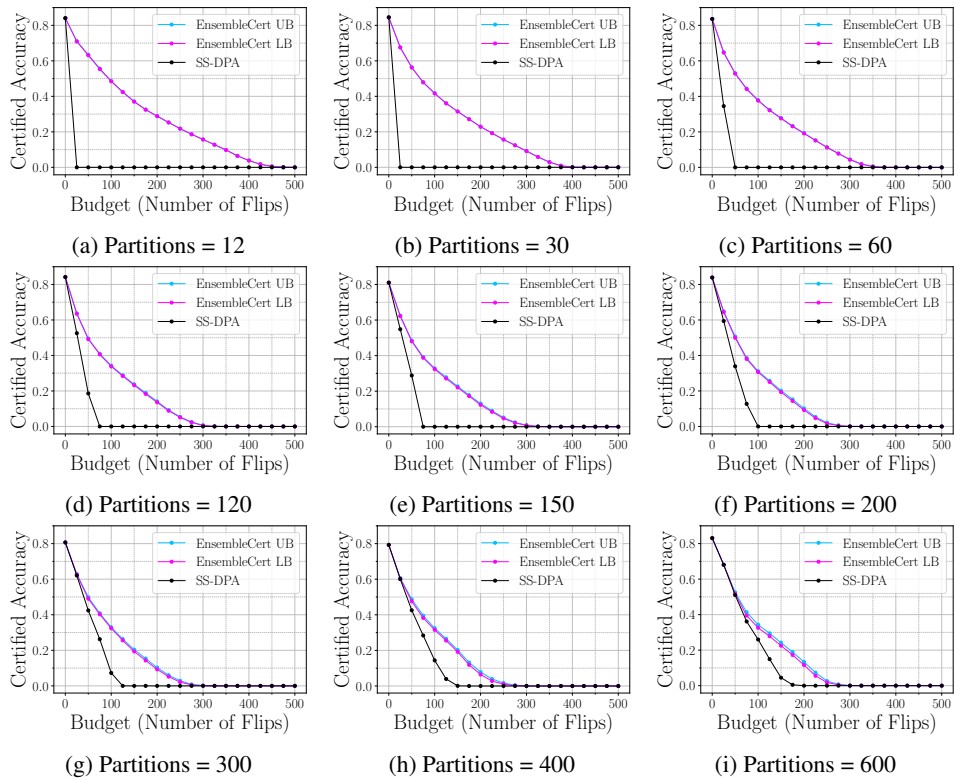

Figure 20: MNIST multi-class kernel SVM results for different numbers of partitions.

