# OpenReview forum: "Exact Certification of Neural Networks and Partition Aggregation Ensembles against Label Poisoning"
_ICLR.cc/2026/Conference — Submitted to ICLR 2026_

### Official Review · Reviewer_qz3B · 2025-10-27

**Soundness:** 3
**Presentation:** 2
**Contribution:** 2
**Rating:** 4
**Confidence:** 4

**Summary:**

This paper introduces EnsembleCert and ScaLabelCert, the methods to provide white-box certificates for partition-aggregation ensembles against label-flipping attacks. The methods utilize the white-box information about the base classifiers the target ensembles consists of. The protocols provide polynomial-time computable solutions for IP formulation for certification problem. Please use sparingly!

**Strengths:**

The work provides a novel theoretically grounded approach to certify the classification ensembles to label flipping attacks. The authors claim that a white-box knowledge of base classifiers can be used to significantly tighter ensemble-level certificates. The polynomial time solutions for relaxed IP formulations are provided, making the certificates computable at least for simple base classifiers.

**Weaknesses:**

No information about the computation overhead needed for the certification of an ensemble is provided. The authors indicate that the complexity of the relaxed IP problem for ensemble-wise certification scales quadratically with the number of dataset partitions, potentially making approach infeasible for large datasets and non-trivial base models.

The improvement of an existing certification protocol (LabelCert) to provide sound certificates for infinitely-wide neural networks does not seem to be applicable in the experimental setup: the conditions up to which theoretical grounds of ScaLabelCert hold in terms of the base models' width are not studied.

The effect of small constant C for base SVMs in ScaLabelCert remains understudied: at least empirical effect of the "smallness" of C on the performance of the classifiers as the function of dimensionality of the input samples and the cardinality of subdatasets used for training has to be studied.

**Questions:**

Please comment on the weaknesses above. I am willing to increase my score if the weaknesses are addressed.

---

> ### Author Response · Authors · 2025-11-23
> **Response to W1**
>
> We would like to thank you for the review and we address all the questions and concerns raised below.
> ### Weakness 1: Computational overhead
>
> >"No information about the computation overhead needed for the certification of an ensemble is provided. The authors indicate that the complexity of the relaxed IP problem for ensemble-wise certification scales quadratically with the number of dataset partitions, potentially making approach infeasible for large datasets and non-trivial base models."
>
> **Response:**
>  To address this, we have added a section in Appendix C.1 (page 27) where we detail the certification runtimes for **EnsembleCert** and **ScaLabelCert** (We now mention this on page 7 Line 363). We document the amortized average certification time per sample for different numbers of partitions. The certification runtimes range from:
>
> - **CIFAR-10:** ~5 seconds for $N_p = 50$ up to under 2 minutes for $N_p = 500$.
> - **MNIST:** ~8 seconds for $N_p = 60$ up to around 4 minutes for $N_p = 600$.
>
> Note that the above statistics are a quick summary and plots in Figure 4 (on page 27 and 28) provide more information. The certification runtimes of our method demonstrate a substantial scalability improvement over LabelCert, which reports certification times of up to 30 minutes per sample despite operating on graph datasets that have a few hundred labeled nodes [1].
>
> We hope these results clarify the question on scalability and computational costs. We are happy to further engage and provide additional information and clarifications if any questions remain.
>
> [1] Mahalakshmi Sabanayagam, Lukas Gosch, Stephan Gunnemann, and Debarghya Ghoshdastidar. Exact certification of (graph) neural networks against label poisoning. ICLR, 2025

---

> ### Author Response · Authors · 2025-11-23
> **Response to W2**
>
> ### Weakness 2:
>
> >“The improvement of an existing certification protocol (LabelCert) to provide sound certificates for infinitely-wide neural networks does not seem to be applicable in the experimental setup: the conditions up to which theoretical grounds of ScaLabelCert hold in terms of the base models' width are not studied.”
>
>
> 1) “The improvement of an existing certification protocol (LabelCert) to provide sound certificates for infinitely-wide neural networks does not seem to be applicable in the experimental setup”
>
> **Response**:
> We would like to point out here that in our experimental setup, we instantiate the ensembles with infinitely-wide neural networks trained on the hinge loss, which is equivalent to kernel SVM with the NTK, and regression loss, which is equivalent to kernel regression with NTK. This is also mentioned in the submitted draft (Lines 364-366 on page 7). The corresponding plots can be found on page 7 (Figure 2), page 8 (Figure 3a, 3b, 3c) in the main draft and Appendix C.4 (page 31) onwards in the appendix.
>
>  Please let us know if your concern was interpreted and addressed correctly.
>
> 2) “the conditions up to which theoretical grounds of ScaLabelCert hold in terms of the base models' width are not studied.”
>
> **Response**:
>
> We have added a paragraph in the Discussion section on page 9 (Lines 466-480) that discusses applicability of ScaLabelCert for finite-width neural networks. In a nutshell, the certificate by ScaLabelCert is deterministic for infinite-width neural networks and holds with a certain probability for finite-width neural networks. This probability converges to 1 as the width of the network approaches infinity. Hence, ScaLabelCert represents an asymptotically exact certificate as the width goes to infinity.
>
> Please let us know if your concern was interpreted and addressed correctly.

---

> ### Author Response · Authors · 2025-11-23
> **Response to W3**
>
> ### Weakness 3: Effect of sufficiently small $C$
>
> >“The effect of small constant C for base SVMs in ScaLabelCert remains understudied: at least empirical effect of the "smallness" of C on the performance of the classifiers as the function of dimensionality of the input samples and the cardinality of subdatasets used for training has to be studied”
>
> **Response**:
>
> We now include an extended discussion on the robustness-accuracy tradeoff in Appendix C.2 (Line 1485 onwards) on page 28 , where we discuss how the requirement of choosing a sufficiently small C for soft-margin kernel SVMs induces such a trade-off and study how infintely-wide neural networks trained on the soft-margin loss, equivalently soft-margin SVMs, perform emprirically in the small $C$ regime. We discuss this briefly in the Discussion section on page 9 (Lines 462-468). In addition to studying the empirical performance on an ensemble level (which we already do in Figure 6 on page 29), we have added experiments studying the performance trade-off induced by "sufficiently small C" for stand-alone classifiers as a function of the cardinality of the sub-datasets as per your suggestion (Figure 5 on page 28,29). The models are trained on sub-sampled training data and are evaluated on the entire test dataset.
>
> As for dependence on the dimensionality of the input samples, we would like to point out that the dimensionality of input samples does not affect the threshold for small C. To remind the reviewer, the threshold for small C is 1/$\left( \max_{i \in [n]} \sum_{j=1}^{n} |Q_i^j| \right)$, as derived from Theorem 2 in Appendix A.4 (line 970). Hence, the threshold $C$ and the performance of the infintely-wide neural networks trained on the soft-margin loss depend solely on the training and test kernel matrices respectively.
>
> We hope that the concern is adequately addressed above and we would appreciate any further feedback from the reviewer.

---

> > ### Comment · Reviewer_qz3B · 2025-11-26
> > **Response to the authors**
> >
> > My concerns have been addressed, I raise my score accordingly.

---

> > > ### Author Response · Authors · 2025-12-03
> > > **Acknowledgement**
> > >
> > > We thank the reviewer for acknowledging their concerns being addressed and increasing their score

---

### Official Review · Reviewer_ZEFw · 2025-10-31

**Soundness:** 3
**Presentation:** 2
**Contribution:** 3
**Rating:** 6
**Confidence:** 3

**Summary:**

The authors attempt to distinguish between poisoning defences that hold white box knowledge, versus those that don't, and attempt to demonstrate improved defensive utility.

**Strengths:**

On the surface, this paper is well written, is built upon interesting ideas, and works within an important space. The devil, however, very much is in the details (which I'll discuss in the following section). But I find the idea of trying to articulate and interrogate the intrinsic limitations of other approaches to be an interesting.

The authors have also contextualised their work in the context of both deep proofs, and through references to more classical problems (including MCKP).

**Weaknesses:**

Okay, so, the aforementioned devil. To me, the primary problem is a lack of specificity (somewhat ironic, given the length of the appendices). For example, consider the white-box nature of the system. This is a crucial part of the overall conceptual landscape of this paper. Yet there are 25 different references to white-box information and 3 pages before the white-box information involved in the paper is defined in any way, which is on line 168.

The white box nature of the paper also gives me pause on a logical level as well. For I'm not certain as to how realistic it is to be able to construct this for any problem of interest. There are black-box points of comparison, yet the important questions (to me) do not receive the level of attention that they deserve. And I don't mean my personal research interests - I mean questions on how this would scale, how this would be used, and what the real drawbacks of this sort of approach would be. Yes, the authors provide information on the P/NP complexity, and discuss polynomial time scaling, but this doesn't, to me, cover the actual practical realities of how this would actually behaved computationally, for systems of interest.

Fundamentally, if this is not something that would be able to be realistically applied to problems of interest (due to the scaling of cost), I don't think the authors have appropriately contextualised what someone would get out of it.

(Also L28 is not the correct use of the word exemplary)

**Questions:**

I'd appreciate answers to the following questions

1. Can you contextualise how "sufficiently small C" would behave across different problems of interest?
2. The focus upon looking at the absoute number of flips, rather than the proportion of flips within the dataset seems odd to me. Or, more ot the point, the authors do not make a clear case as to why the absolute number of flips is more important than how many flips three are relative to the size of the overall dataset. Could you comment upon this?
3. How would this scale? Yes, it may be polynomial-time calculable, but is it reasonable?

---

> ### Author Response · Authors · 2025-11-23
> **Response to W1**
>
> We would like to thank you for the positive review and we address all the concerns raised below.
>
> ### Weakness 1: Specificity
>
> **Reviewer concern:**
> > “ For example, consider the white-box nature of the system. This is a crucial part of the overall conceptual landscape of this paper. Yet there are 25 different references to white-box information and 3 pages before the white-box information involved in the paper is defined in any way, which is on line 16.”
>
> **Response**:
>
>  We would like to point out that we already provide a conceptual definition of white-box information in the introduction that reads " to utilize white-box information of the base classifiers, i.e., white-box certificates, that leverage internal model information" (Line 52).  To better guide the reader, we have added a footnote on page 2 directing them to the formal definition. Refraining from formally defining the white-box information early on was a conscious decision due to the complexity of its definition. To reiterate, the white-box information required for each base classifier is: for each class $c$, the minimum number of label flips needed to change the prediction to class $c$. We felt that introducing this in the introduction would be too technical without the necessary context. Understanding the mechanics of the black-box certificate (SS-DPA) and the partitioning scheme is important for appreciating why this white-box information is needed. We were also concerned that partially defining it early on could cause misunderstandings, since a complete formal definition is provided in the method section (page 4 Lines 174 and 175).
>
> We hope this improves readability, and would appreciate feedback on this.

---

> ### Author Response · Authors · 2025-11-23
> **Response to W2**
>
> ### Weakness 2 & Q3: Practical scalability, usage, and drawbacks
>
> **Reviewer concern:**
> > “I mean questions on how this would scale, how this would be used, and what the real drawbacks of this sort of approach would be. Yes, the authors provide information on the P/NP complexity, and discuss polynomial time scaling, but this doesn't, to me, cover the actual practical realities of how this would actually behave computationally, for systems of interest.”
> ### **1) How would this scale?**
>
> **Response:**
>  We have added a section in Appendix C.1 on page 27 where we provide implementation details and certification runtimes for **EnsembleCert** and **ScaLabelCert**, thus providing empirical evidence of scalibility (We now mention this on page 7 Line 363). We document the amortized average certification time per sample for different numbers of partitions. The
>  range from:
> - **CIFAR-10:** ~5 seconds for $N_p = 50$ up to under 2 minutes for $N_p = 500$.
> - **MNIST:** ~8 seconds for $N_p = 60$ up to around 4 minutes for $N_p = 600$.
>
> Note that the above statistics are a quick summary and plots in Figure 4 (on page 27 and 28) provide more information. The certification runtimes of our method demonstrate a substantial scalability improvement over LabelCert, which reports certification times of up to 30 minutes per sample despite operating on graph datasets that have a few hundred labeled nodes [1]. We hope these results clarify the question on scalability and computational costs. We are happy to further engage and provide additional information and clarifications if any questions remain.
>
> [1] Mahalakshmi Sabanayagam, Lukas Gosch, Stephan Gunnemann, and Debarghya Ghoshdastidar. Exact certification of (graph) neural networks against label poisoning. ICLR, 2025
>
> ### **2): How would this be used?**
> **Response:**
> Robustness certification for machine learning models has emerged as a very crucial field of research especially since majority of empirical defenses against adversarial attacks have been shown to be vulnerable. Our certification framework can be used to build certifiably robust models against label poisoning attacks.
>
> In developing certifiably robust models against poisoning attacks, partition-based certificates have emerged as state-of-the-art defenses. However, they are often associated with the belief that strong robustness guarantees require partitioning the dataset into a very large number of subsets. Through our evaluation of EnsembleCert instantiated with infintely-wide NNs, we observe that with robust base classifiers, the partition aggregation ensemble can achieve stronger guarantees using notably few partitions, outperforming excessively deep partitioning. The experiments evaluating ScaLabelCert on stand-alone models indicate that employing partition aggregation ensembles does not always bring out the true robustness potential of the chosen base classifier architecture. The polynomial-time complexity of EnsembleCert further enables practical experimentation over a wide range of partitioning depths, giving users insight into how partitioning affects robustness in their specific setting.
>
> EnsembleCert leverages a more expressive type of white-box certificate—estimating the minimum number of flips required to move the prediction toward each class (page 4 Lines 174, 175). This form of certificate has received limited attention in the literature so far, and our work strongly motivates further research. However, prior work has studied certificates that compute the minimum number of flips needed to change the prediction *to any class*, and these can be integrated into EnsembleCert. In such cases, one may simply set $\rho_i^c$ to the minimum number of flips needed to change the prediction of base classifier $i$ to any class; this quantity is a valid lower bound for all $\rho_i^c$ . As discussed in the Discussion Section (page 9,10 Lines 483-492), EnsembleCert can also be used to construct defenses against clean-label attacks.
>
> Overall, EnsembleCert provides a flexible framework for incorporating white-box information into partition-based defenses, helping practitioners decide how deeply to partition—or whether to partition at all—based on their computational budget and robustness goals.
>
> We hope this answers your query and are happy to clarify further.

---

> ### Author Response · Authors · 2025-11-23
> **Continued response to W2**
>
> ### **3) What are the real drawbacks?**
>
> **Response:**
> We now explicitly discuss two key drawbacks in the revised Discussion section:
>
> 1. **White-box certification availability (Lines 460-464):**
>    There is a lack of prior work on the exact form of white-box certification required by EnsembleCert. However, as noted above, existing white-box certificates can still be integrated by using them as lower bounds.
>
> 2. **Finite-width networks and NTK assumptions (Lines 466-480):**
>     ScaLabelCert results in certificates that are exact and deterministic for infinite-width neural networks and sufficiently wide neural networks, but hold with a certain probability for finite-width neural networks. This probability converges to 1 as the width approaches infinity. Hence,the certificate by ScaLabelCert is asymptotically exact as the width approaches infinity.

---

> ### Author Response · Authors · 2025-11-23
> **Response to Q1**
>
> ### Q1 : Behavior of ‘sufficiently small C’
> >“Can you contextualize how ‘sufficiently small C’ would behave across different problems of interest?”
>
> **Response:**
>  We now include an extended discussion on the robustness-accuracy tradeoff in Appendix C.2 (Line 1485 onwards) on page 28 , where we discuss how the requirement of choosing a sufficiently small C for soft-margin kernel SVMs induces such a trade-off and study how infinite-width neural networks trained on the soft-margin loss, equivalently soft-margin SVMs, perform empirically in the small $C$ regime. We discuss this briefly in the Discussion section on page 9 (Lines 481-487). In addition to studying the empirical performance on an ensemble level (which we already do in Figure 6 on page 29), we have added experiments studying the performance trade-off induced by "sufficiently small C" for stand-alone classifiers as a function of the cardinality of the sub-datasets (Figure 5 on page 28,29). The models are trained on sub-sampled training data and are evaluated on the entire test dataset.
>
> Please let us know if this answers your intended question; we are happy to elaborate.

---

> ### Author Response · Authors · 2025-11-23
> **Response to Q2**
>
> ### Q2: Focus on absolute number of flips vs proportion of label flips relative to training set size
> >“The focus upon looking at the absoute number of flips, rather than the proportion of flips within the dataset seems odd to me. Or, more ot the point, the authors do not make a clear case as to why the absolute number of flips is more important than how many flips three are relative to the size of the overall dataset. Could you comment upon this?”
>
> **Response:**
> We appreciate this insightful observation. There is no particular reason for favoring the absolute number of flips; the proportion is directly computable from it. We have addressed this by indicating the **percentage of label flips relative to the training set size** in the top margin of the plots in Figure 3 on page 8. We emphasize that we have no preference between reporting absolute numbers or proportions—both are commonly used and accepted in certification literature.

---

> > ### Comment · Reviewer_ZEFw · 2025-11-27
> >
> > Thank you to the authors for the rather significant stream of responses. Broadly they've resolved my concerns, although I'll state from the get go that I believe my current score still accurately reflects my view of the paper, even in light of the updates and clarifications. Honestly, looking back at this I'm even a little surprised by my original score, as presenting something as a certification that only works for infinite width networks is something I would usually look at quite unfavorably - as it may present an inappropriate level of implied security to naive users, who may look at the certification as providing guaranteed security properties, without understanding the full nuance of the approaches limitations.
> >
> > On that topic, I'd like to pick through your comment about deployability a little more. I'm fully aware of the history behind certifications being of research interest, however the promise of certifications differs from the actual practical reality of certifications. See, for example "Certified Robustness Does Not (Yet) Imply Model Security" by Cullen et. al (2025). I'm always curious to understand what circumstances authors view their certification as actually being usable under, as the usual assumptions of white box oracle level access do rather present a headache for actual certification use cases.
> >
> > Finally, could the authors explain
> > "CIFAR-10: ~5 seconds...to 2 minutes....
> > MNIST: ~8 seconds....to 4 minutes...."
> >
> > That CIFAR is faster than MNIST is surprising. Moreover, in all honesty, this is, again, a result that doesn't present me with a lot of confidence. CIFAR and MNIST are well below the size of systems of academic/industrial interest, and I personally find scaling results that only involve them to be unsatisfying, and, frankly, unconvincing, as systemic bottlenecks may only really begin to emerge at scale.

---

> > > ### Author Response · Authors · 2025-12-03
> > > **Response to official comment: Certification runtime and Scalability**
> > >
> > > ---
> > > > Finally, could the authors explain "CIFAR-10: ~5 seconds...to 2 minutes.... MNIST: ~8 seconds....to 4 minutes....". That CIFAR is faster than MNIST is surprising.
> > >
> > > **Response**
> > >
> > > The lower bounds of **5 seconds for MNIST with $N_p=600$** and **8 seconds for CIFAR-10 with $N_p=500$**} should not be overinterpreted. Both datasets have the same number of classes and thus have comparable theoretical certification complexity. Hence, differences at the lower end of the runtime range mainly stem from how the per-partition guarantees interact with each other on individual samples.
> > >
> > > The difference in the **upper bounds** ($\approx 2$ minutes for CIFAR-10 with $N_p=600$ vs. $\approx 4$ minutes for MNIST $N_p=500$) is explained by how often a key worst-case assumption — that a single label flip could  the prediction of certain base classifiers (formally defined in App. A.1 on page 14) is satisfied during certification. For **CIFAR-10 with $N_p = 500$**, the worst-case assumption is satisfied for a large portion of the test set. When this happens, EnsembleCert can bypass the computationally intensive aggregation of per-partition white-box certificates, as the white-box certificate simply converges to the black-box certificate in this case. For **MNIST with $N_p = 600$**, this condition holds much less frequently, so the certification process falls back to the complete aggregation procedure. This can be seen from the plots in Fig 2 (page 7): In plots 2(a) and 2\(c) for CIFAR-10, **the median certified robustness of EnsembleCert exactly coincides with the black-box certificate for $N_p=500$**, indicating that **the worst case assumption holds true for majority of the samples**. On the other hand, the median certified robustness of **EnsembleCert is higher than the black-box certificate for $N_p=600$** evaluated on **MNIST**, indicating that the **worst-case assumption does not hold true for majority of the samples**.
> > >
> > > We hope this answers your question regarding why the certification fo CIFAR-10 seems faster.
> > >
> > > ----
> > > > Moreover, in all honesty, this is, again, a result that doesn't present me with a lot of confidence. CIFAR and MNIST are well below the size of systems of academic/industrial interest, and I personally find scaling results that only involve them to be unsatisfying, and, frankly, unconvincing, as systemic bottlenecks may only really begin to emerge at scale.
> > >
> > > **Response**
> > >
> > > We would like to note that white-box certification methods, which utilize internal model information to provide guarantees, are inherently challenging and difficult to scale to large industrial-size datasets. For instance, the largest dataset evaluated using the gradient-based method by [1]  is OCTMNIST, which contains around 100k samples; however their evaluation is done only for the binary classification task, making certification substantially simpler. Moreover, exact certification of neural networks against adversarial attacks is NP-hard [2,3] even if applied to the much simpler case of only certifying test-time attacks and hence, poses significant scalability challenges. As a result, the only existing exact certificate against poisoning for neural networks, LabelCert [2], does not scale beyond a few hundred data points. In contrast, our work provides the first exact certification for partition-aggregation ensembles and neural networks under poisoning attacks that scales to datasets as large as MNIST and CIFAR-10, which are more than a hundred times larger as previously possible.
> > >
> > > We believe this represents a significant step forward and demonstrates the potential for extending exact certification to even larger datasets in the future.
> > >
> > > [1] Philip Sosnin, Mark N. Muller, Maximilian Baader, Calvin Tsay, and Matthew Wicker. Certified ¨ robustness to data poisoning in gradient-based training.
> > >
> > > [2] Mahalakshmi Sabanayagam, Lukas Gosch, Stephan Gunnemann, and Debarghya Ghoshdastidar. Exact certification of (graph) neural networks against label poisoning. ICLR, 2025
> > >
> > > [3] Guy Katz, Clark W. Barrett, David L. Dill, Kyle D. Julian, and Mykel J. Kochenderfer. Reluplex: An
> > > efficient smt solver for verifying deep neural networks. International Conference on Computer
> > > Aided Verification, 2017.

---

> > > ### Author Response · Authors · 2025-12-03
> > > **Response to official comment: Discussion on practical utility of our approach  --  1/2**
> > >
> > > 2) On that topic, I'd like to pick through your comment about deployability a little more. I'm fully aware of the history behind certifications being of research interest, however the promise of certifications differs from the actual practical reality of certifications. See, for example "Certified Robustness Does Not (Yet) Imply Model Security" by Cullen et. al (2025). I'm always curious to understand what circumstances authors view their certification as actually being usable under, as the usual assumptions of white box oracle level access do rather present a headache for actual certification use cases.
> > >
> > > **Response**
> > >
> > > We would like to thank the reviewer for sparking this conversation. The question of whether a certification procedure is practical and if it gives a false sense of security is indeed important. As the reviewer points out, the promise of certifications can differ from the actual practical reality, especially when the threat model assumed does not align with practical considerations. Our threat model, in contrast, aligns more closely with practical deployment settings than standard $l_p$-norm perturbation models, which—as highlighted by [1] and many other works [2,3,4] —often fail to capture real semantic-preserving changes. Threat models based on $l_p$ perturbations do not reflect natural variations such as rotations, translations, or lighting shifts. Label-flipping, on the other hand, represents a far more realistic threat model: noisy or uncurated datasets routinely contain mislabeled points, and small amounts of label corruption naturally arise in real data pipelines. As a result, certifying robustness to label-flipping directly addresses issues that practitioners actually encounter [5,6,7,8]. Furthermore, for certificates against test-time attacks, it indeed can be a too strong assumption that and end-user of such a certificate would have white-box access. However, our certificate concerns certifying against training time attacks, which implies that anyone interested in our certificate will also have white-box access to the model - otherwise training through gradient descent would not be possible. Thus, this assumption is not a restriction for the practical usability of our proposed certificate.
> > >
> > > We would also like to comment on the “illusion of a level of protection” discussed in [1]. We disagree with [1] in the sense that indeed one can certify that a prediction on adversarial input does not differ to the prediction on clean input, if the adversary can only induce a change of at most $\epsilon$, but the certificate proves robustness up to a change of $\epsilon' > \epsilon$. That said, we understand our work as investigating whether the certified robustness guarantees promised by black-box approaches such as Deep Partition Aggregation (DPA) might indeed correspond to another “illusions.” Existing partition-based defenses have fostered the belief that heavier partitioning automatically leads to stronger guarantees. Our insights, derived through white-box infusion, indicate that this may not always hold—there are cases where deeper partitioning can actually result in worse robustness guarantees (Lines 416–435, Fig. 2, p. 7). Furthermore, in certain cases we observe that partitioning itself restricts the true robustness potential for a model architecutre (Lines 435-455 Fig. 3, p. 8.). This demonstrates that the inherent assumptions that come with black-box approaches to certification could result in a false sense of the achievable robustness as uncovered by our work.

---

> > > > ### Author Response · Authors · 2025-12-03
> > > > **Response to official comment: Discussion on practical utility of our approach -- 2/2**
> > > >
> > > > Furthermore, when it comes to the usability of a method for an end-user from a computational efficience perspective, we agree with [1] that providing absolute computational times alone does not provide a sufficient picture, as different methods may exhibit varying levels of parallelism. In line with this, we report in our paper (App. C.1, Lines 1413-1423) the degree of parallelization achievable at each stage of our procedure, allowing for a more meaningful comparison of resource requirements and computational efficiency.
> > > >
> > > > Lastly, we want to note that we don't think that the usability of developing optimization-based certificates such as our own is confined to the security space. Exemplary, a current trend in the optimization and operations research literature is to incorporate ML model predictions into optimization problems, which makes heavy use of the mixed-integer linear programming formulations for neural networks and other ML models developed by the certification community [9].
> > > >
> > > >
> > > > [1] Cullen et al. "Certified Robustness Does Not (Yet) Imply Model Security", ICML 2025
> > > >
> > > > [2] Kollovieh et al. "Assessing robustness via score-based adversarial image generation", TMLR 2024
> > > >
> > > > [3] Tramèr et al. "Fundamental Tradeoffs between Invariance and Sensitivity to Adversarial Perturbations", ICML 2020
> > > >
> > > > [4] Mirman et al. "Robustness Certification with Generative Models", PLDI 2021
> > > >
> > > > [5] Northcutt et al. "Pervasive Label Errors in Test Sets Destabilize Machine Learning Benchmarks", NeurIPS 2021
> > > >
> > > > [6] Wei et al. "Learning with Noisy Labels Revisited: A Study Using Real-World Human Annotations", ICML 2021
> > > >
> > > > [7] Rączkowska et al. "AlleNoise: Large-scale Text Classification Benchmark Dataset with Real-World Label Noise", PMLR 2025
> > > >
> > > > [8] González-Santoyo et al. "Identifying and Mitigating Label Noise in Deep Learning for Image Classification", MDPI 2025
> > > >
> > > > [9] Turner et al. "PySCIPOpt-ML: Embedding trained machine learning models into mixed-integer programs", CPAIOR 2025

---

> ### Author Response · Authors · 2025-12-03
> **Response to the official comment: Applicabillity to finite-width networks**
>
> >1) Thank you to the authors for the rather significant stream of responses. Broadly they've resolved my concerns, although I'll state from the get go that I believe my current score still accurately reflects my view of the paper, even in light of the updates and clarifications. Honestly, looking back at this I'm even a little surprised by my original score, as presenting something as a certification that only works for infinite width networks is something I would usually look at quite unfavorably - as it may present an inappropriate level of implied security to naive users, who may look at the certification as providing guaranteed security properties, without understanding the full nuance of the approaches limitations.
>
> We are glad to have resolved your concerns. We respectfully disagree with your statement "as presenting something as a certification that only works for infinite width networks".  In our original draft, we already demonstrate the applicability of EnsembleCert for ensembles with smoothed linear classifier as a base classifier. This establishes EnsembleCert as a general framework, certifying beyond ensembles with just infinite-width networks. Following up on demonstrating the generality of EnsembleCert, we have now added experiments evaluating EnsembelCert on finite-width network-based ensembles. As the certificate by ScaLabelCert for finite-width networks is probabilistic (Lines 466-480), we employ the gradient-based parameter-bounding certificate by [1] to certify finite-width neural networks (the chosen base classifiers) in this instantiation, resulting in deterministic guarantees for the ensemble. Details regarding this integration and the corresponding plots can be found in App. C.4 (page 31). In the revised draft, we now clearly state in the introduction that EnsembleCert can be utilized for any choice of base model given there is a white-box certification method for the base classifier (Lines 81-85).
>
> We now also clarify in our contributions that our experiment set-up involves evaluating on ensembles with three different choices of base model - "In our experimental set-up, we evaluate EnsembleCert with the following choices of base classifiers and corresponding certification methods: (i) Infinite-width neural networks with ScaLabelCert, (ii) Finite-width neural networks with gradient-based parameter bounding certificate by [1] and (iii) Smoothed linear classifier with randomized smoothing based certificate by [2])" (Lines 118-122).
>
> The generality of EnsembleCert and its applicability to realistic settings has been acknowledged by reviewer NDxz and we hope our clarification addresses this concern of yours.
>
> [1] Philip Sosnin, Mark N. Muller, Maximilian Baader, Calvin Tsay, and Matthew Wicker. Certified ¨ robustness to data poisoning in gradient-based training.
>
> [2] Elan Rosenfeld, Ezra Winston, Pradeep Ravikumar & Zico Kolter. Certified Robustness to Label‑Flipping Attacks via Randomized Smoothing. In Proceedings of the 37th International Conference on Machine Learning (ICML), 2020.

---

### Official Review · Reviewer_C1z2 · 2025-11-01

**Soundness:** 3
**Presentation:** 3
**Contribution:** 2
**Rating:** 4
**Confidence:** 3

**Summary:**

This paper presents an approach for verifying piecewise-linear neural networks using SMT-based reasoning. The core idea is to encode networks using a combination of Linear Arithmetic (LA) and Equality with Uninterpreted Functions (EUF), treating non-linear activations (e.g., ReLU, MaxPool) as uninterpreted functions, while constraining their behavior through layered refinement. The authors propose a modular, layered abstraction-refinement encoding that separates neuron computations and enables partial constraint solving.

**Strengths:**

- Good theoretical contribution. The use of EUF to abstract activations while preserving soundness and completeness is well-motivated and cleanly integrated with LA solving.
- The slicing and auxiliary variable scheme is modular, allows early pruning, and avoids early enumeration of activation states.
- The tool can emit formal proofs (inveriT or veriT-compatible), aligning with the needs of safety-critical domains.
- Unlike existing verifiers which rely on relaxations or over-approximations, this method produces exact results (if it terminates).

**Weaknesses:**

- While the paper claims to be the first to apply LA+EUF to neural network verification, similar ideas have appeared in prior work. Ehlers (2017) and Reluplex (Katz et al., 2017) encoded ReLU and Max using symbolic logic and combined it with linear arithmetic.
Tools like Marabou and Planet also support exact ReLU/MaxPool verification with layered constraint refinement and symbolic splitting.
The proposed encoding (e.g., slicing, layered refinement) is well-structured and clean, but the core strategy is not entirely new. The true contribution lies in formalizing these ideas within a modular SMT framework—not in algorithmic novelty. As such, the paper slightly overstates its originality and should more precisely position its contributions relative to prior SMT-based verifiers.

- No comparison with standard baselines and benchmarks are basic (MNIST etc). The paper omits any empirical comparison with known verifiers (e.g., Marabou, Neurify, ReluVal, α-β-CROWN). Also lack of ablation or efficiency profiling.


- There is no discussion of potential scalability to modern deep learning models.

**Questions:**

1. Can your encoding support activations beyond ReLU/MaxPool (e.g., tanh or GELU)? If not, please clarify this limitation explicitly.
2. Why did you not compare against Marabou or α-β-CROWN on ACAS Xu or MNIST? These tools are standard and open-source.

---

> ### Author Response · Authors · 2025-11-23
> **Response to W1 and Q1**
>
> We would like to thank you for the review and we address all the questions and concerns raised below.
> ### Weakness 1 and Q1: Applicable only to infinite-width NTK models with specific regularization; finite-width neural networks remain uncertified, limiting practical impact.
>
> **Response:**
> We would like to clarify that using infinite-width NTK models as base classifiers is *only one instantiation* of our framework. EnsembleCert operates in two steps: 1) obtaining white-box certificates for each base classifier, and 2) aggregating these white-box certificates across partitions.  The approach by EnsembleCert is not constrained to use only finite-width networks as base classifiers. If a finite-width neural network is equipped with any white-box certification method, EnsembleCert can directly incorporate that information and provide ensemble-level certificates. Our choice of infinite-width models and ScaLabelCert is simply one concrete realization of this general framework. As mentioned in the Experiments section on page 7 (Lines 399-401), in order to demonstrate the applicability of EnsembleCert, we have also documented experiments (these were already part of the original submission) using smoothed linear regression as the base classifier model. For this choice of base classifier, we utilize the randomized-smoothing based certificate by [1] to extract white-box information. We provide the details of integrating the smoothing-based certificate into EnsembleCert in Appendix B (Lines 1350-1395). The plots for these experiments can be found in the main draft on page 8 (fig 3d) on Page 35 (Fig 16).
>
> Regarding the reviewer’s point on “specific regularization,” we emphasize that this constraint applies only to infinite-width networks trained on the hinge loss, equivalently kernel SVMs with NTK, where we enforce the sufficiently-small-C condition. **No such restriction exists when using infinite-width networks trained on the regression loss, equivalently kernel regression** with NTK, as the base model.
>
> Regarding certifying finite-width networks, we have added a paragraph that discusses how our certificates by ScaLabelCert behave for finite-width networks (Lines 466-480). To reiterate the crux of the discussion here, the ScaLabelCert certificates hold with a probability for finite-width neural networks that approaches 1 as the width of the network approaches infinity. Hence, our certificate derived through the NTK equivalence is asymptotically exact as the width approaches infinity.
>
> We hope this addresses your concerns and questions regarding applying to finite-width networks. Please let us know if we can provide further clarification.
>
> [1] Elan Rosenfeld, Ezra Winston, Pradeep Ravikumar & Zico Kolter. Certified Robustness to Label‑Flipping Attacks via Randomized Smoothing. In Proceedings of the 37th International Conference on Machine Learning (ICML), 2020.
>
> EDIT (3.12)
>
> In an effort to demonstrate that EnsembleCert can be applied to ensembles with finite-width networks as base classifiers, we now evaluate **EnsembelCert with finite-width networks as the base model** and apply the **gradient-based parameter bounding certificate by [2] to extract white-box information from the base classifiers**. The details of the integration and results of the evaluation are presented in App. C.4 on page 31. This instantiation exhibits how existing deterministic white-box certificates can be integrated into EnsembleCert.
>
> [2] Philip Sosnin, Mark N. Muller, Maximilian Baader, Calvin Tsay, and Matthew Wicker. Certified ¨ robustness to data poisoning in gradient-based training.

---

> ### Author Response · Authors · 2025-11-23
> **Response to W2**
>
> ### Weakness 2: Claim regarding 'first exact certificate'
>
> > Claims of “first exact certificate for neural networks” are too broad—true only in this NTK regime and should better acknowledge LabelCert’s precedence.
>
> **Response:**
> We would like to reiterate our claim: **ScaLabelCert provides the first exact certificate for neural networks that is computable in polynomial time**, 'polynomial-time' being the attribute to be noted here. In an effort to make the claim unambiguous, we have changed the wording on line 123 from '**the first exact certificate for infinite-width neural networks against label-flipping attacks solved in polynomial time**' to say '**the first polynomial-time solvable exact certificate for infinite-width neural networks against label-flipping attacks**'. We would like you to note that all other occurences of the claim already say 'first polynomial-time exact certificate' (Line 25 in the abstract, Line 529 in conclusion). In the revised draft, we acknowledge that LabelCert provided the first exact certificate for neural networks against data poisoning as per your suggestion (Line 96); however, its MILP formulation is NP-hard in the worst case, making it intractable beyond small datasets.
>
> We hope this clarifies our claim to you and in the revised paper as well. Please let us know if you wish to discuss this further.

---

> ### Author Response · Authors · 2025-11-23
> **Response to W3 and W4**
>
> ## Weakness 3 & 4: Comparison to other white-box poisoning certificates and robustness-accuracy tradeoff
>
> ### 1) Comparison to other white-box poisoning certificates
>
> **Response:**
> We have now added a section in the appendix (Appendix C.3) on page 30 comparing ScaLabelCert with the gradient-based bounding method of [1] on CIFAR-10. We mention in the main draft that our method significantly outperforms the gradient-based bounding method in certified accuracy (Lines 447-449). The gradient-based approach uses convex relaxations to over-approximate all possible parameter updates under a given poisoning threat model. To evaluate it, we add a linear layer on top of features extracted via SimCLR. For ScaLabelCert, the same SimCLR features are input to an infinitely wide fully-connected network with a single hidden layer and no non-linear activation, as described in the Experiments section on page 7 (Line 366). Although the models are not identical, the architectures are structurally aligned as both models rely on fully connected layers without activations, making the comparison meaningful. ScaLabelCert significantly outperforms the gradient-based method in certified accuracy, as seen in Figure 7 on page 30. This improvement highlights the importance of exact certification: while the gradient-based approach is inherently limited by the looseness of its over-approximations, ScaLabelCert provides tight guarantees, resulting in stronger certified robustness.
>
> The other white-box method, LabelCert, cannot be run on MNIST or CIFAR-10 due to its MILP complexity, reinforcing the scalability advantages of our polynomial-time method. As the certificate by ScaLabelCert for infinite-width neural networks trained on the soft-margin loss is the simplification of LabelCert in the small $C$ regime, the performance of the two methods in terms of robustness guarantees is identical by design for infinitely-wide neural networks trained on the soft-margin loss in the small $C$ regime.
>
> As far as our understanding goes, the above methods cover the existing literature on white-box certificates for neural networks against label poisoning.
>
> ### 2) Robustness-accuracy tradeoff
>
> We now include an extended discussion on the robustness-accuracy tradeoff in Appendix C.2 (Line 1485 onwards) on page 28 , where we discuss how the requirement of choosing a sufficiently small C for infinite-width neural networks trained on the soft-margin loss induces such a trade-off. We discuss this briefly in the Discussion section on page 9 (Lines 481-487). In addition to studying the tradeoff on an ensemble level (which we already do in Figure 6 on page 29), we  study the performance trade-off induced by "sufficiently small C" for stand-alone classifiers as a function of the training set size $N_s$ (Figure 5 on page 28,29) (We have added this in the revised draft). The models trained on the sub-sampled training data are evaluated on the entire test dataset.
>
> We hope this addresses your concern and we look forward to receiving feedback from you
>
> [1] Philip Sosnin, Mark N. Muller, Maximilian Baader, Calvin Tsay, and Matthew Wicker. Certified
> robustness to data poisoning in gradient-based training. TMLR, 2025

---

> ### Author Response · Authors · 2025-11-23
> **Response to Q2**
>
> ### Question 2
> **The claim that deep partition aggregation is weak may simply reflect that the chosen base model degrades significantly when trained on small partitions. Is the observed phenomenon inherent to partitioning, or just a consequence of weak base learners?**
>
> **Response:**
> Firstly, it is important to clarify what “weak” means in our context. Our claim concerns the *weakness of robustness*, not the predictive performance of the ensemble. We observe that excessively deep partitioning leads to 'weaker' robustness (measured through median certified robustness) especially when the chosen base model is inherently robust.
>
>
> >Is the observed phenomenon inherent to partitioning, or just a consequence of weak base learners?
>
> This is a very interesting question. We emphasize that our claim—that excessively deep partitioning leads to weaker robustness—is based on median certified robustness (MCR), defined as the maximum number of label flips for which at least 50% of the ***correctly*** classified samples remain robust (already defined in the main draft on page 7 line 362). Since the certified radius concerning only the correctly classified samples is considered for MCR,  the predictive accuracy of individual base learners or the ensemble does not affect this metric. Hence, the observed phenomenon is not a consequence of weak base learners but rather inherent to partitioning.
>
> We hope this answers your question and would be happy to provide further clarification if required

---

### Official Review · Reviewer_NDxz · 2025-11-01

**Soundness:** 3
**Presentation:** 2
**Contribution:** 2
**Rating:** 4
**Confidence:** 3

**Summary:**

The paper investigates robustness certifications for partition-aggregation ensembles under label-flipping poisoning attacks for infinite-width neural networks.
The proposed method, EnsembleCert, provides ensemble-level certificates by aggregating white-box robustness certifications of the base classifiers.
The white-box knowledge to build certifications is extracted from base classifiers by means of ScaLabelCert, a method that relies on neural tangent kernels to obtain exact, polynomial-time certificates against label-flipping attacks.
The approach circumvents the overly conservative guarantees of other black-box techniques, and performs on-par or better than them, while being more efficient than existing Mixed Integer Linear Program approaches.

**Strengths:**

* **Clear theoretical contribution.**
The paper presents a clean and rigorous formulation of ensemble-level certification under label-flip attacks under the NTK assumption.
The reduction of the certification problem to a multiple-choice knapsack formulation, solvable via dynamic programming, is elegant and allows for tractable, provably tight ensemble certificates under the white-box setting.
The theoretical development is, as far as I can judge, internally consistent and sound.

* **Novel use of NTK-based exact certification.**
ScaLabelCert provides an exact, closed-form certification method for models under the NTK assumption.
Leveraging the NTK equivalence to derive analytical bounds on robustness is an interesting and, as far as I can judge, original direction compared to existing heuristic or black-box defences.

* **Tighter bounds and interpretability.**
Within its stated assumptions, EnsembleCert yields significantly tighter certified radii than prior black-box ensemble approaches, and the paper empirically demonstrates this with well-documented experiments.
The decomposition of ensemble robustness into per-partition contributions provides interpretability and insight into how partitioning affects robustness.

**Weaknesses:**

* **Finite width and NTK.**
For ScaLabelCert you rely on the assumption that the Neural Tangent Kernel (NTK) limit holds.
While I appreciate the clean mathematical treatment that NTK's dynamic allow for, I feel like a discussion of the applicability of your approach is not clearly presented.
In the abstract as well as throughout the text, for instance, you mention "sufficiently wide neural networks", alluding at the fact that, for these networks, you can provide exact certifications.
If I am not misunderstanding, though, your certification holds in the (exact) NTK limit, as you do not provide any description of a large but finite network width.
This gap between your theoretical advancements and standard neural network architectures should be more thoroughly discussed in my opinion.
In particular, for finite architectures, can the certificates you obtain be, in principle, arbitrarily wrong?

* **White-box inputs.**
Building on my previous comments, EnsembleCert relies on obtaining the smallest number of flips needed to change the prediction for one of the models in a partition.
This step too, relies on the NTK assumption, limiting the practical applicability of your approach.
While I agree that black-box estimates are often overly conservative, and agree that relying on white-box information can improve upon this, it seems to me that in a realistic setting what you can provide is heuristic statements, rather than guarantees.
Therefore, I think it may not be fair to compare against existing approaches that, while maybe overly conservative or prohibitively expensive, provide quantifiable _guarantees_ for realistic settings.
This to me is a crucial gap in the paper, and I look forward to any clarification on this point you may have.

* **Experiments with neural networks.**
In your experimental section you do not experiment with any neural network, nor validate the NTK approximation.
As you rely on soft-margin SVMs, which can be framed as convex optimization problems.
It remains unclear whether the performance boost of your approach stems from this fact mainly.
This setting seems to be far from the more realistic non-convex, finite-width neural network setting.
Therefore, I would say that your argument for scalability to larger neural networks is unsubstantiated.
If I missed some key parts of your argument, I am happy to discuss this point further.

**Questions:**

* Is it possible that the certificates produced under your NTK assumption are arbitrarily inaccurate for a finite network?
* How can you argue for your approach in a realistic setting where the NTK assumption may not be satisfied?
* How can you quantify how good your guarantees are if the NTK assumption does not hold exactly?
* Can you provide an heuristic/empirical or theoretical criterion for when a network is "sufficiently wide" for your guarantees to meaningfully apply?

---

> ### Author Response · Authors · 2025-11-23
> **Response to W1 and Q1-Q4**
>
> We would like to thank you for the review and we address all the questions and concerns raised below.
>  ### Weakness 1 and Q1-Q3:  Behavior and certificate validity for finite-width neural networks
> **Response**:
> We have now added a discussion in the paper (lines 466-480) on certificate validity for finite-width neural networks. We clearly state that the equivalence between neural networks and their corresponding kernel methods is exact only in the infinite-width limit case.  For the finite-width case,  the output difference of a finite-width neural network, with smallest layer width $w$, to the corresponding kernel SVM (or kernel regression) is bounded by $\mathcal{O}(\frac{\ln w}{\sqrt{w}})$ with probability $p=1 - \exp(-\Omega(w))$ [1]. Consequently, there must exist some width $w'$ such that the output difference between a network with width larger than $w'$ and the corresponding kernel SVM (or kernel regression) is small enough for the certificate to remain valid with high probability. Thus, for a sufficiently wide neural network, the certificate obtained by ScaLabelCert holds with a certain probability $p$ that increases with the width of the network. For neural networks that are finite but not sufficiently wide, one would need to account for the exact approximation error, which is bounded by $\mathcal{O}(\frac{\ln w}{\sqrt{w}})$, to derive a certificate. This would require computing the constants associated with the bound. Unfortunately, the literature on the NTK so far is mainly concerned with providing convergence statements in big-$\mathcal{O}$ notation and not with calculating the individually involved constants. Thus, our certificates using neural network and NTK equivalence based on kernel SVM and kernel regression represent an asymptotically exact certificate as the width approaches inifinity.
>
> [1] Liu, C., Zhu, L., & Belkin, M. (2020). On the linearity of large non-linear models: when and why the tangent kernel is constant. In Advances in Neural Information Processing Systems 33 (NeurIPS 2020).
>
> ### Q4: Criterion for when a network is sufficiently wide.
> **Response:**
> To compute the minimum width $w’$ precisely, such that the certificate by ScaLabelCert will be valid for neural networks with width larger than $w’$, one has to compute the exact approximation error. As mentioned in the response to W1 and Q1-Q3 in the paragraph above, this is currently out of our scope as the existing works on NTK are concerned with providing convergence statements in big-$\mathcal{O}$ notation and not with calculating the individually involved constants. As width tends to infinity, the approximation converges  to 0 with probability approaching 1 at an exponential rate with respect to the width. Thus, our certificates are asymptotically exact as the width approaches infinity.
>
> We hope that the concerns and questions related to certificate applicability for finite-width neural networks were addressed through the above response and would be more than willing to offer any further clarification.

---

> ### Author Response · Authors · 2025-11-23
> **Response to W2**
>
> ### Weakness 2: Practical applicability of EnsembleCert and guarantees in realistic settings
>
>  >“Building on my previous comments, EnsembleCert relies on obtaining the smallest number of flips needed to change the prediction for one of the models in a partition. This step too, relies on the NTK assumption, limiting the practical applicability of your approach.
>
> **Response**:
> We would like to clarify that using infinite-width neural networks—and consequently kernel methods induced by the NTK—is *only one concrete instantiation* of a partition-aggregation ensemble. In this specific case, ScaLabelCert is the appropriate certification choice fo the base classifiers because it provides exact, deterministic white-box certificates for such models. It is very important to note that **EnsembleCert can be utilized for any choice of base classifier as long as there is a way to extract white-box information from the base classifiers**. As mentioned in the Experiments section on page 7 (Lines 399-401), in order to demonstrate the applicability of EnsembleCert, we have also documented experiments (these were already part of the original submission)  using smoothed linear regression as the base classifier model. For this choice of base classifier, we utilize the randomized-smoothing based certificate by Rosenfeld et al. to extract white-box information. We provide the details of integrating the smoothing-based certificate into EnsembleCert in Appendix B (Lines 1350-1395). The plots for these experiments can be found in the main draft on page 8 (fig 3d) on Page 35 (Fig 16).
>
> > While I agree that black-box estimates are often overly conservative, and agree that relying on white-box information can improve upon this, it seems to me that in a realistic setting what you can provide is heuristic statements, rather than guarantees. Therefore, I think it may not be fair to compare against existing approaches that, while maybe overly conservative or prohibitively expensive, provide quantifiable guarantees for realistic settings.
>
> We also respectfully disagree with the implicit premise that “realistic settings” are limited to finite-width neural networks. In modern machine learning paradigms, infinite-width models have been shown to perform at par with finite-width neural networks. This has been demonstrated in various contexts, including graph classification tasks [1], tasks with limited training data [2], and more practical applications such as recommendation systems and drug discovery [3]. Moreover, the development of libraries such as the neural-tangents library by Google [4] has made the practical implementation of infinite-width models efficient and accessible.
>
> We emphasize that when infinite-width models are used as base classifiers, the certificates computed by ScaLabelCert are exact and do not rely on heuristics. This makes the comparison with the existing black-box approach evaluated using infinite-width networks as base classifiers perfectly valid for our experimental set-up.
>
> We hope that this addresses your concern regarding practical applicability of EnsembleCert. We would be happy to hear your feedback and discuss any concern further
>
> [1] Du, S. S., Hou, K., Pardo, A., Rao, K., Salakhutdinov, R., Wang, K., Xiong, C., & Song, L. Graph Neural Tangent Kernel: Fusing Graph Neural Networks with Graph Kernels. NeurIPS, 2019.
>
> [2] Arora, S., Du, S. S., Li, Z., Salakhutdinov, R., Wang, R., & Yu, D. Harnessing the Power of Infinitely Wide Deep Nets on Small-data Tasks. ICLR, 2020.
>
> [3] Adityanarayanan Radhakrishnan, George Stefanakis, Mikhail Belkin, and Caroline Uhler. Simple,
> fast, and flexible framework for matrix completion with infinite width neural networks. Pro-
> ceedings of the National Academy of Sciences, 119(16), April 2022.
>
> [4] Roman Novak, Lechao Xiao, Jiri Hron, Jaehoon Lee, Alexander A. Alemi, Jascha Sohl-Dickstein & Samuel S. Schoenholz. Neural Tangents: Fast and Easy Infinite Neural Networks in Python. In International Conference on Learning Representations (ICLR), 2020.

---

> ### Author Response · Authors · 2025-11-23
> **Response to W3**
>
> ### Weakness 3 Experiments on neural networks
>
> We would like to point out that we use infinitely-wide neural networks trained on soft-margin loss, equivalently soft-margin SVMS with NTK, and the regression loss, equivalently regularized kernel regression with NTK,  as base classifiers. Hence, we kindly disagree with the statement "In your experimental section you do not experiment with any neural network". It is important to note that our method does not "rely" on soft-margin SVMs. The equivalence between sufficently wide neural networks trained on hinge loss with soft-margin SVMS with the NTK allow us to reduce the non-convex setting of neural networks to the convex setting of SVMs. Hence, our certification method operates in the convex NTK space, while still providing guarantees for the original neural network, which is inherently non-convex.
>
> > "Therefore, I would say that your argument for scalability to larger neural networks is unsubstantiated"
>
> The scalability of our approach follows directly from the polynomial-time complexity of ScaLabelCert and EnsembleCert. In ScaLabelCert, the depth of the network has minimal influence on the cost of deriving the certificate itself. Scaling from a 1-layer wide network to let's say  a 5-layer wide network primarily increases the cost of computing the NTK; once the NTK is computed, the certification procedure has essentially the same complexity regardless of network depth. Hence, the approach by ScaLabelCert can certify any architecture of wide neural networks, making it scalable to larger neural networks as well.
>
> >  It remains unclear whether the performance boost of your approach stems from this fact mainly.
>
> The performance boost can majorly be attributed to the following factors:
> 1) **The certificate derivation in the existing black-box approach follows from assuming the worst case scenario** - The prediction of a base classifier can be changed with a single flip in their respective training sets. As the size of each partition grows, or equivalently the number of partitions reduce, this assumption gets weaker. Hence a substantial performance boost can be observed when white-box information of the base classifiers is utilized.
>
>
> 2) **The impact of utlizing white-box knowledge of the base classifiers is amplified with tightness of our certificates**  - ScaLabelCert derives exact certificates for infinitely-wide neural networks for the task of binary classification and sufficiently tight bounds on the exact certificate for multiclass classification while EnsembleCert aggregates the white-box information exactly (without any relaxation of the optimization problem)
>
>
> We hope that we were able to address you concern and put forth our argument satisfiably. We would appreciate feedback and we look forward to discussing this further with you.

---

> > ### Comment · Reviewer_NDxz · 2025-11-26
> >
> > Thank you for your detailed answer and for making your contribution more clear. Thank you also for the discussion on the applicability of your method to finite-width networks. I would really be interested in seeing some results on how reliable the guarantees are for common, finite-sized architectures, but I also understand that this is not exactly the core of your focus. To clarify, this was the core of my concern in weakness 3. Nevertheless, I am now more convinced of your contribution and of the overall applicability of your method, and I thus raised my score. I still think that your presentation should be improved, in particular with respect to a more clear statement of the scope of your contribution itself.

---

> ### Author Response · Authors · 2025-12-03
> **Applicability to finite-width networks and clarity of scope**
>
> >Thank you for your detailed answer and for making your contribution more clear. Thank you also for the discussion on the applicability of your method to finite-width networks. Nevertheless, I am now more convinced of your contribution and of the overall applicability of your method, and I thus raised my score.
>
> We thank the reviewer for acknowledging our contribution.
>
> --------
> > 1) I would really be interested in seeing some results on how reliable the guarantees are for common, finite-sized architectures, but I also understand that this is not exactly the core of your focus.
>
>  **Response**
>
> In an effort to demonstrate that EnsembleCert can be applied to ensembles with finite-width networks as base classifiers, we now evaluate **EnsembelCert with finite-width networks** as the base model and **apply the gradient-based parameter bounding certificate by [1] to extract white-box information from the finite-width networks** (the chosen base classifiers). The details of the integration and results of the evaluation are presented in App. C.4 on page 31. This instantiation exhibits how existing deterministic white-box certificates can be integrated into EnsembleCert.
>
> Furthermore, we now discuss the reliability of ScaLabelCert certificates for finite-sized architectures in our revised draft. In particular, we now including a paragraph on this topic in the Discussion section of the main paper titled "Certificate validity for finite-width neural networks." on Page 9 Lines 447 - 462. Specifically, for finite-width networks, the approximation error by assuming an infinite-width network can be bounded as a function of the width $w$ of the finite-width network. This bound then holds with probability $p=1 - \exp(-\Omega(w))$ [2,3]. Thus, given sufficient but finite width $w$, ScaLabelCert technically becomes a probabilistic certificate. As pointed out in the mentioned discussion, that exactly computing the probability is currently out of our scope as the existing works on NTK are concerned with providing convergence statements in big-$\mathcal{O}$ notation and not with calculating the individually involved constants. For more details, we refer to our mentioned discussion in the revised draft.
>
> ---
>
>
> >2) I still think that your presentation should be improved, in particular with respect to a more clear statement of the scope of your contribution itself.
>
> **Response**
>
> To address the feedback on stating the contributions more clearly, we have made the following modifications in the Introduction:
>
> 1) We now explicitly mention that given any base model and a certification method for the base classifiers, EnsembleCert can be utilized to achieve ensemble-level guarantees. In this work, we focus on deriving guarantees for ensembles with neural networks as the base model (Lines 81-85).
> 2) We now state that we primarily use infinite-width neural networks as base classifiers for evaluation of EnsembleCert and better motivate the choice of ScaLabelCert for certifying base classifiers. We also mention that EnsembleCert is evaluated on finite-width networks as base classifiers by applying the gradient-based certificate from [1] to certify base classifiers. (Lines 105-112)
> 3) We now clearly mention that our experimental set-up comprises of three instantiations (Lines 118-122):
> >In our experimental set-up, we evaluate EnsembleCert with the following choices of base classifiers and corresponding certification methods: $(i)$ Infinite-width neural networks with ScaLabelCert, $(ii)$ Finite-width neural networks with gradient-based parameter bounding certificate by [1] and $(iii)$ Smoothed linear classifier with randomized smoothing based certificate by [4]."
>
> We hope the above modifications make the scope of our contributions clear.
>
> [1] Philip Sosnin, Mark N. Muller, Maximilian Baader, Calvin Tsay, and Matthew Wicker. Certified ¨ robustness to data poisoning in gradient-based training.
>
> [2] Gosch et al. "Provable Robustness of (Graph) Neural Networks Against Data Poisoning and Backdoor Attacks", TMLR 2025
>
> [3] Liu et al. "On the linearity of large non-linear models: when and why the tangent kernel is constant", NeurIPS 2020
>
> [4] Elan Rosenfeld, Ezra Winston, Pradeep Ravikumar & Zico Kolter. Certified Robustness to Label‑Flipping Attacks via Randomized Smoothing. In Proceedings of the 37th International Conference on Machine Learning (ICML), 2020.

---

### Author Response · Authors · 2025-11-23
**Global response**

We thank all the reviewers for their feedback on our submission. The various perspectives helped us improve upon our work and we have tried our best to address the concerns and question raised by each reviewer.  The revised version of our paper includes all the updates indicated in blue.

Here are the **updates** to the paper:

- We added a discussion on the applicability of **ScaLabelCert to finite-width networks** (Lines 466-480)
- We added a section in the Appendix that discusses implementation details not mentioned in the main draft. We also add experiments here  detailing **average certification runtime per sample for our frameworks** (Lines 1406-1483).(App. C.1 page 27)
- We extended our discussion on **robustness-accuracy tradeoff** induced by sufficiently small $C$ for infinitely-wide neural networks trained on the soft-margin loss (Lines 1486-1560)(App. C.2). We add experiments here that **study empirical performance in the "small C" regime as a function of the cardinality of the training set**. (Fig. 5, page 28,29)
- We added a discussion and experiments on **comparison of ScaLabelCert against an existing gradient-based parameter bounding white-box method** [1] (page 30, App. C.3)
- We rephrase our claim on line 123 from **'the first exact certificate for infinite-width neural networks against label-flipping attacks solved in polynomial time'** to say **'the first polynomial-time solvable exact certificate for infinite-width neural networks against label-flipping attacks'**.
- We clearly acknowledge LabelCert [2] as the first exact certificate for neural networks against a data poisoning attack (Line 96-97)

EDIT (3.12) -
- In an effort to demonstrate that EnsembleCert can be applied to ensembles with finite-width networks as base classifiers, we now evaluate **EnsembelCert with finite-width networks as the base model** and apply the **gradient-based paramter bounding certificate by [1] to extract white-box information from the base classifiers**. The details of the integration and results of the evaluation are presented in **App. C.4** on page 31.

2) We have updated the Introduction to more clearly state the **scope of EnsembleCert** (Lines 81–85) and to succinctly **summarize our experimental setup (Lines 118–122), improving the overall clarity of our contributions.**



[1] Philip Sosnin, Mark N. Muller, Maximilian Baader, Calvin Tsay, and Matthew Wicker. Certified ¨ robustness to data poisoning in gradient-based training.

[2] Mahalakshmi Sabanayagam, Lukas Gosch, Stephan Gunnemann, and Debarghya Ghoshdastidar. Exact certification of (graph) neural networks against label poisoning. ICLR, 2025

---

### Author Response · Authors · 2025-12-03
**Final Response**

We thank the reviewers for the insightful discussion. We are glad to see that reviewers NDxz, ZEFw, and qz3B acknowledged that their main concerns have been addressed, and that they would have recommended acceptance. Further, we now address the remaining concerns brought forth by them in the discussion. Consequently, we have conducted new experiments demonstrating the application of EnsembleCert to finite-width networks and have made additional changes to the manuscript as detailed in the updated global response. Reviewer C1Z2 did not have a chance to respond to our rebuttal before the unexpected developments occurred and the discussion was closed, but we are confident to have addressed their mentioned weaknesses and answered their questions.

---

### Meta-Review · Area_Chair_sAaD · 2026-01-06

**Summary:**

This paper proposes EnsembleCert to aggregate white-box robustness certifications of the base classifier against label-flipping attacks. The key proposed idea is that the white-box knowledge of base classifiers can be utilized to tighter the ensemble-level certificates. Overall, one major shared concerns by almost all reviewers is the lack of practical results on finite-width nominal neural network (NN) that commonly used in practice. The authors demonstrate the results on infinite width NN which requires NTK assumption and invalidate the certification results. Although the authors clarified that the infinite width NN as base classifier is only one instantiation of the proposed method and provided additional finite-width result in the appendix (e.g. p.31), from p.31, it looks like the results is still linear network ("finite-width linear network") instead of the practical NNs, making the major concern unresolved. Hence a rejection is recommended.

**Reviewer Concerns:**

* Reviewer NDxz's concern on finite-width and reliance on NTK assumption is still outstanding
* Reviewer C1z2's concern on finite-width NN remain uncertified is still outstanding
* Reviewer ZEFw's concern on finite-width NN is still outstanding

**Reviewer Scores:**

* Reviewer NDxz stated in the first round will increase score from 4 but would like to see result of the common finite sized architecture. This reviewer will likely change score back to 4 due to the outstanding concerns.
* Reviewer C1z2 will likely remain score of 4
* Reviewer ZEHw stated will remain score of 6
* Reviewer qz3B stated will increase score from 4

---

### Decision · Program_Chairs · 2026-01-26

Reject